# The demise of the giant ape *Gigantopithecus blacki*

Yingqi Zhang[1,2,20 ✉], Kira E. Westaway[1,2,20 ✉], Simon Haberle[3], Juliën K. Lubeek[2], Marian Bailey[4], Russell Ciochon[5], Mike W. Morley[6], Patrick Roberts[7,8,9], Jian-xin Zhao[10], Mathieu Duval[11,12], Anthony Dosseto[13], Yue Pan[1], Sue Rule[3], Wei Liao[14], Grant A. Gully[15], Mary Lucas[8], Jinyou Mo[16], Liyun Yang[17], Yanjun Cai[18], Wei Wang[14 ✉] & Renaud Joannes-Boyau[1,4,19 ✉]

The largest ever primate and one of the largest of the southeast Asian megafauna, *Gigantopithecus blacki*[1], persisted in China from about 2.0 million years until the late middle Pleistocene when it became extinct[2–4]. Its demise is enigmatic considering that it was one of the few Asian great apes to go extinct in the last 2.6 million years, whereas others, including orangutan, survived until the present[5]. The cause of the disappearance of *G. blacki* remains unresolved but could shed light on primate resilience and the fate of megafauna in this region[6]. Here we applied three multidisciplinary analyses—timing, past environments and behaviour—to 22 caves in southern China. We used 157 radiometric ages from six dating techniques to establish a timeline for the demise of *G. blacki*. We show that from 2.3 million years ago the environment was a mosaic of forests and grasses, providing ideal conditions for thriving *G. blacki* populations. However, just before and during the extinction window between 295,000 and 215,000 years ago there was enhanced environmental variability from increased seasonality, which caused changes in plant communities and an increase in open forest environments. Although its close relative *Pongo weidenreichi* managed to adapt its dietary preferences and behaviour to this variability, *G. blacki* showed signs of chronic stress and dwindling populations. Ultimately its struggle to adapt led to the extinction of the greatest primate to ever inhabit the Earth.

Our current understanding of *Gigantopithecus blacki* derives from Early to Middle Pleistocene cave deposits in southern China between the Yangtze River and the South China Sea (Fig. 1 and Supplementary Information section 1). This pongine[7] is considered a key member of the Early to Middle Pleistocene *Gigantopithecus–Sinomastodon* and *Stegodon–Ailuropoda* faunal zones of (sub)tropical oriental Asia, from about 2.0 million years ago (Ma) to 330 thousand years ago (ka)[2,3,8,9]. It is known for its unusually large molars, atypical enamel thickness, estimated body height of about 3 m and mass of 200–300 kg, making it the largest primate ever to have existed on Earth[4]. Despite 85 years of searching, the *G. blacki* fossil record is restricted to four mandibles and almost 2,000 isolated teeth with no postcranial evidence[4]. Its initial discovery in an apothecary shop in Hong Kong as a 'Dragon tooth'[1] initiated a search for the first in situ finds[10] (Extended Data Fig. 1f) and culminated in the discovery of several cave sites in two main areas,

Chongzuo and Bubing Basin, in the Guangxi ZAR province[4]. These sites contain crucial evidence for its survival and eventual demise.

Very few of these *G. blacki* sites have been dated using more than one radiometric technique; thus the timing of extinction remains uncertain[11]. The current timeline for its presence is 2.2 Ma (Baikong cave[12]) to 420–330 ka (Hejiang Cave[9]). During this time, *G. blacki* underwent morphological changes including an increase in tooth size[13] and dental complexity[9], seemingly indicating a dietary change in response to ecological pressure[13]. Reconstructions of *G. blacki* diet based on the dental anatomy indicate a specialized herbivore with adaptations for the consumption of abrasive food[14,15], heavy mastication of fibrous food[16,17] and a fruit-rich diet[6,18]. The diverse forest ecosystem at the time of Baikong had the capacity to support the biomass of several primate communities[4] over a wide area from Guangxi, Guizhou, Hainan and Hubei Provinces[19]. However, by the time of Hejiang,

[1]Key Laboratory of Vertebrate Evolution and Human Origins, Institute of Vertebrate Paleontology and Paleoanthropology, Chinese Academy of Sciences, Beijing, China. [2]School of Natural Sciences, Faculty of Science and Engineering, Macquarie University, Sydney, New South Wales, Australia. [3]School of Culture, History and Languages, ANU College of Asia and the Pacific, Australian National University, Canberra, Australian Capital Territory, Australia. [4]GARG, Southern Cross University, Lismore, New South Wales, Australia. [5]Department of Anthropology and Museum of Natural History, University of Iowa, Iowa City, IA, USA. [6]College of Humanities, Arts and Social Sciences, Flinders University, Adelaide, South Australia, Australia. [7]isoTROPIC Research Group, Max Planck Institute for Geoanthropology, Jena, Germany. [8]Department of Archaeology, Max Planck Institute for Geoanthropology, Jena, Germany. [9]School of Social Sciences, University of Queensland, Brisbane, Queensland, Australia. [10]School of Earth and Environmental Sciences, University of Queensland, Brisbane, Queensland, Australia. [11]National Research Centre on Human Evolution CENIEH, Burgos, Spain. [12]Australian Research Centre for Human Evolution (ARCHE), Griffith University, Brisbane, Queensland, Australia. [13]Wollongong Isotope Geochronology Laboratory, School of Earth, Atmospheric and Life Sciences, University of Wollongong, Wollongong, New South Wales, Australia. [14]Institute of Cultural Heritage, Shandong University, Qingdao, China. [15]College of Science and Engineering, Flinders University, Adelaide, South Australia, Australia. [16]Natural History Museum of Guangxi, Nanning, China. [17]Chongzuo Zhuang Ethnological Musuem, Chongzuo, China. [18]Institute of Global Environmental Change, Xi'an Jiaotong University, Xi'an, China. [19]Palaeo-Research Institute, University of Johannesburg, Johannesburg, South Africa. [20]These authors contributed equally: Yingqi Zhang, Kira E. Westaway. ✉e-mail: zhangyingqi@ivpp.ac.cn; kira.westaway@mq.edu.au; wangw@sdu.edu.cn; renaud.joannes-boyau@scu.edu.au

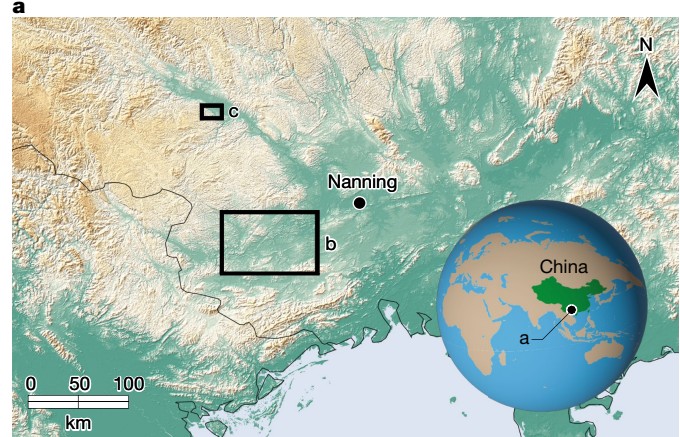

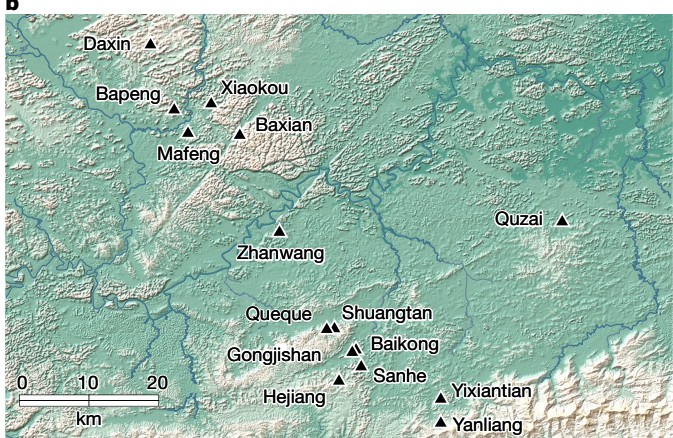

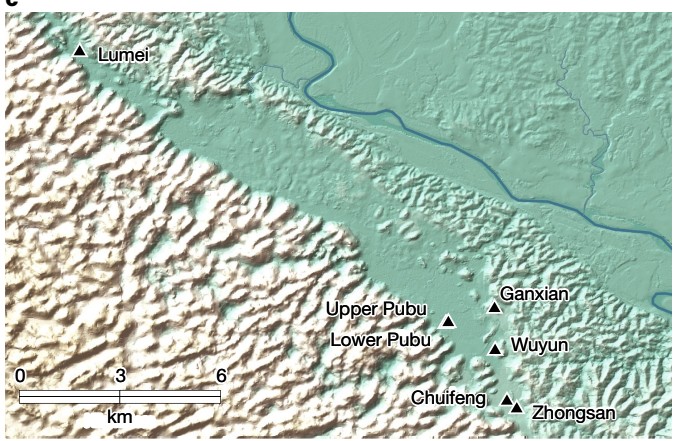

**Fig. 1 | The location of the study sites in this research. a–c**, The location of Southern China, Guangxi ZAR province and the city of Nanning (**a**), with the location of the Chongzuo study area marked by a large box (**b**) and the Bubing Basin study area marked by a smaller box (**c**). **b**, The location of the 16 cave sites analysed in the Chongzuo study area. **c**, The location of the six caves analysed in the Bubing Basin study area including both *G. blacki*-bearing and non-*G. blacki*-bearing caves from both regions.

*G. blacki* had a dramatic range reduction to just Guangxi[9,13]. The reasons for this dramatic reduction and eventual extinction remain hotly disputed[4] because of a lack of a regional approach, a focus on single sites and methods and an absence of behavioural[4] and environmental evidence[20].

To identify the potential causes of *G. blacki* extinction, we applied a regional approach to 22 caves in Chongzuo and Bubing Basin that contained either *G. blacki*-bearing (11) or non-*G. blacki*-bearing (11)

cave deposits (Extended Data Figs. 1 and 2 and Supplementary Information sections 2 and 3). Using a combination of previous excavations (1999–2016) and newly discovered caves (2017–2020) we identified and sampled fossil breccias for dating, palaeoclimate proxies and behavioural analyses. We applied six independent dating techniques to the sediments (post-infrared-infrared stimulated luminescence (pIR-IRSL), optically stimulated luminescence (OSL), electron spin resonance (ESR) on quartz and U-series on speleothem) and fossils (U-series on teeth, coupled US-ESR) to determine a Bayesian modelled age range for each site (Supplementary Information sections 4–8), which were then further modelled to provide a regional extinction window (EW). We applied pollen, charcoal, palaeontological, stable isotope and microstratigraphical analyses to the sediments and fossils to reconstruct the past environments (Supplementary Information sections 3 and 10–12). Finally, we applied trace element, stable isotope and dental microwear textural analysis (DMTA) to the *G. blacki* and closest relative *Pongo weidenreichi* teeth to determine any changes in the diet and behaviour of *G. blacki* before and within the EW that may have related to its demise (Supplementary Information sections 12–14).

According to the 157 radiometric age estimates, the fossil evidence in the 22 caves ranges from 2,300 to 49 ka (Figs. 2a and 3a, Extended Data Figs. 3–6 and Supplementary Information sections 4–8 for all dating tables and discussion of limitations and uncertainties). This study expands the timeline for the presence of *G. blacki* from 2.3 Ma to 255 ka, provides a precise timing for the window of extinction at 295–215 ka (2$\sigma$) (Supplementary Information section 9) and establishes focus points for the palaeoenvironmental and behavioural analysis (pre-EW (2,300–700 ka), transitional phase (700–295 ka), EW (295–215 ka) and post-EW (215 ka to the present)).

Our pollen analysis indicates that during the pre-EW the environment was dominated by arboreal species (*Pinaceae*, *Fagaceae* and *Betulaceae*) with patches of grassland (Figs. 2b and 3b). However, before the EW during the transitional phase there was a change in forest plant communities and an increase in forest disturbance taxa with more open forests dominating. Post-EW about 200 ka, there was a large decrease in arboreal cover, an increase in ferns (for example, *Moraceae* and *Podocarpus*), a large increase in grassland (for example, *Poaceae*) and increased evidence of charcoal in the landscape (Extended Data Fig. 7 and Supplementary Information section 10).

Detailed faunal analysis indicates that the pre-EW sites were characterized by *G. blacki* (in relatively large numbers) (Fig. 3c), *Ailuropoda microta*, *Procynocephalus*, *Sinomastodon*, *Stegodon*, *Hesperotherium* and *Hippopotamodon*, which shifted to *G. blacki* (in relatively small numbers) (Fig. 3c), *Ailuropoda baconi*, *Stegodon* and *Elephas* before the EW and an absence of *G. blacki* post-EW (Supplementary Information section 3). The microstratigraphic analyses of five caves show pre-EW microfacies dominated by fine grains, higher clays and oxides, bioturbation and guano-induced phosphatization. At the EW, grain sizes increased, with lower oxides, bioturbation and bone/tooth alteration enabling better fossil preservation. During the post-EW, this reverted back to pre-EW features (Extended Data Fig. 8c and Supplementary Information section 11).

The stable isotope data indicate that for the pre-EW period the $\delta^{13}$C and $\delta^{18}$O of *G. blacki* range between −16.2 to −13.8‰ and −9.7 to −7.0‰, respectively. During during the EW, this increases slightly to −15.3 to −10.3‰ and −9.3 to −6.3‰, respectively. In the case of *P. weidenreichi*, the pre-EW $\delta^{13}$C and $\delta^{18}$O ranges are similar at −14.7 to −13.7‰ and −7.1 to −6.3‰, extending to −14.7 to −13.3‰ and changing to −4.9 and −4.4‰ during the EW period (Fig. 3d,e, Extended Data Fig. 8b and Supplementary Information section 12).

The trace element analysis of the pre-EW *G. blacki* teeth shows several, distinct and synchronous Sr/Ca and Ba/Ca bandings in the enamel and dentine that change to significantly less visible diffuse banding closer to the EW (Fig. 2d, Extended Data Figs. 9 and 10a and Supplementary Information section 13). In addition, distinct lead banding can be seen

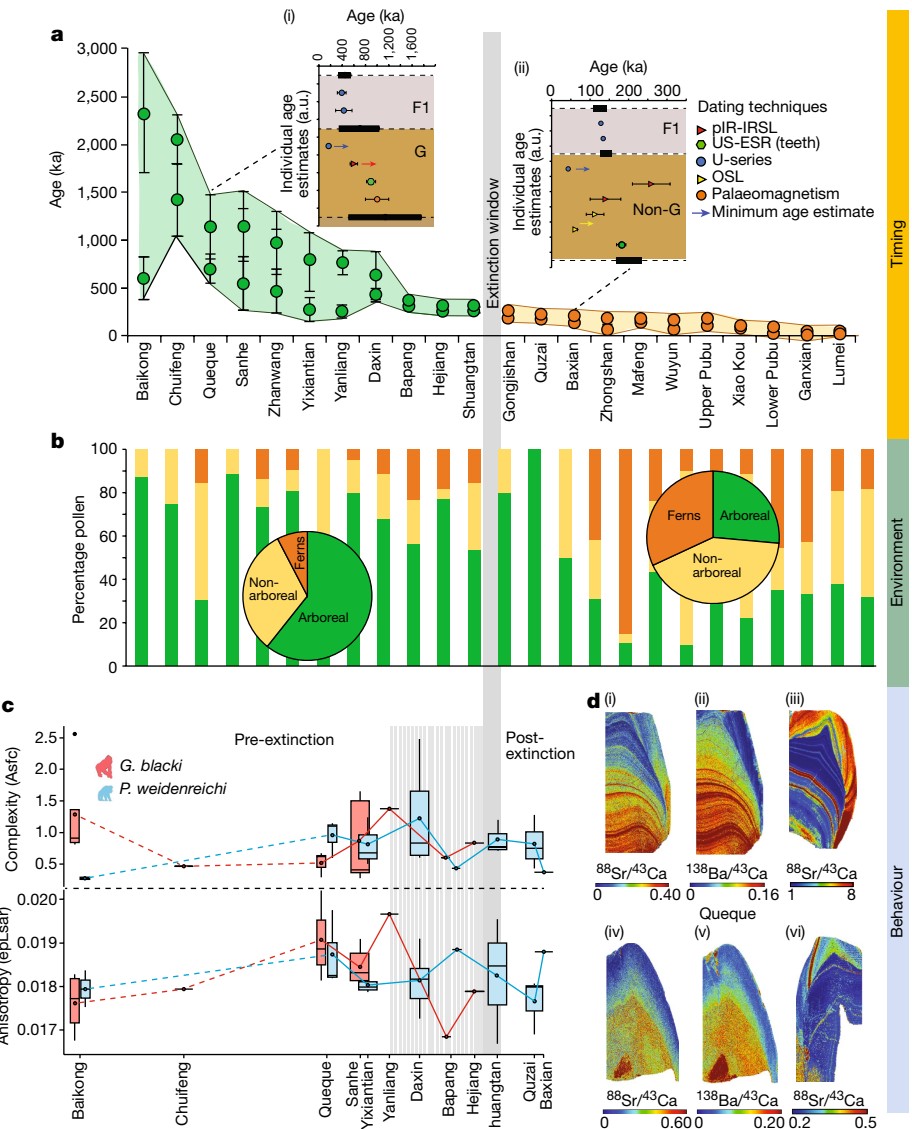

**Fig. 2 | Example datasets to support the extinction events. a–d**, Data relate to timing (**a**), environment (**b**) and behaviour (**c,d**) presented by sites. **a**, Modelled age ranges of each cave (n = 22 caves) using the minimum and maximum age of the fossil-bearing unit (n = 157 samples). The caves (x axis) versus age (y axis), with *G. blacki* (green circles) and non-*G. blacki* (orange circles) breccia. The data points represent mean ages with s.d. at 2σ uncertainties. The insets are modelled breccia from Queque (i) and Baxian (ii). G, *G. blacki*-bearing breccia; F1, overlying flowstone; and Non-G, absence of *G. blacki*. Data points are mean ages with s.d. at 2σ uncertainties. The black horizontal rectangles (with dashed lines) represent the boundary according to the modelling (Supplementary Information section 2 and Supplementary Fig. S1a–v). The modelled EW is the vertical grey line. **b**, Percentage pollen from the sites in **a** representing arboreal (green), non-arboreal (yellow) and ferns (orange). The pie charts provide an average of pollen changes for pre- (left) and post-extinction (right). **c**, DMTA boxplot series according to age of 12 caves (x axis) versus molar microwear complexity (Asfc, top, y axis) and anisotropy (epLsar, bottom, y axis) of *G. blacki* (red, n = 16) and *P. weidenreichi* (blue, n = 22). The boxplots size ranges represent mean complexity and anisotropy values per site. Data are presented as mean values ± interquartile range and whiskers at 95% CI (Supplementary Table S28). **d**, Trace elemental mapping of *G. blacki* and *P. weidenreichi*. Sr/Ca (i) and Ba/Ca (ii) of a right M3 *G. blacki* tooth (CSQSN-44) and Sr/Ca map from a right M2 *P. weidenreichi* tooth (CSQ0811-4) (iii) all from Queque Cave. Below, Sr/Ca (iv) and Ba/Ca (v) from a P4 tooth of *G. blacki* (ST_02_109) compared to Sr/Ca (vi) from a left M3 tooth of *P. weidenreichi* (CLMST0911-118) all from Shuangtan Cave. a.u., arbitrary units.

in the pre-EW, which becomes less distinct during the EW (Extended Data Fig. 10a). The microwear analysis reveals no statistically significant dietary differences between *G. blacki*- and *P. weidenreichi*-bearing sites (Supplementary Information section 14). There are, however, significant dietary differences in four *G. blacki*-bearing sites between the pre-EW and just before the EW. *G. blacki* tends to show slightly higher fluctuations in mean anisotropy and complexity trend lines, whereas those of *P. weidenreichi* seem more stable, especially for anisotropy over and beyond the EW (Figs. 2c and 3f,g, Extended Data Fig. 10b and Supplementary Information section 14).

For the first time, the largest collection of in situ evidence of *G. blacki* spanning its entire range has been robustly dated to provide a precise timeline for the presence and absence of *G. blacki* from the fossil record. Previous dating has mostly focused on the earlier *G. blacki* evidence[2,8] and site-specific chronologies (for example, ref. 9). In contrast, by constraining caves within the entire age range in both Chongzuo and Bubing Basin we have more accurately established a regional window of extinction at 295–215 ka.

The pollen and faunal data indicate that the early mosaic landscapes were interrupted by enhanced environmental variability (Fig. 3b) before

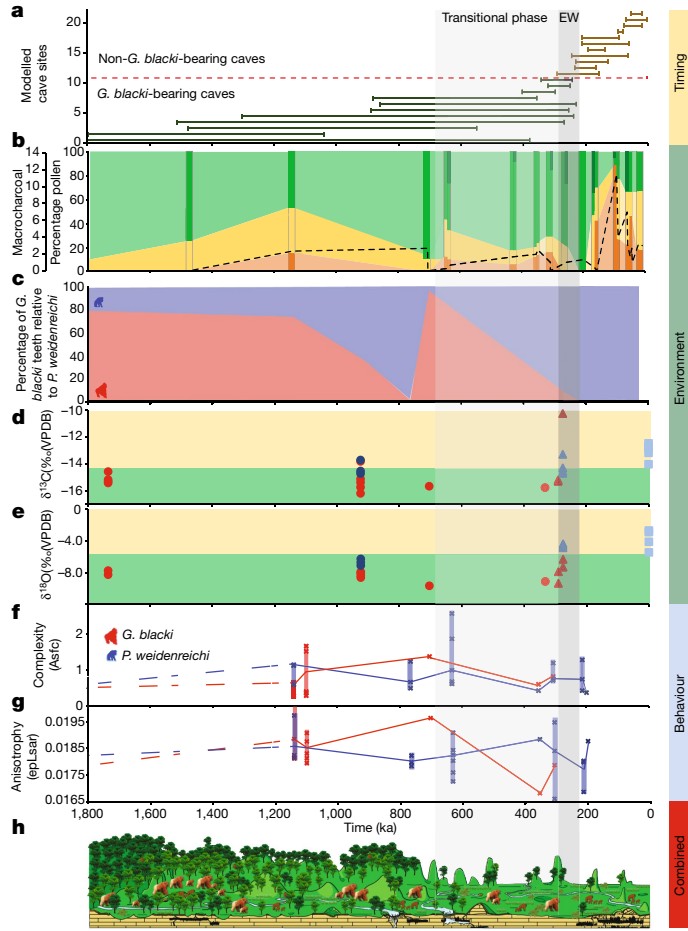

**Fig. 3 | A summary of all datasets plotted against time. a**, Timeline for extinction based on the modelled age ranges for all 22 caves. The numbers on the *y* axis relate to the caves in Fig. 2a. Note the reduced timeline (1,800 ka). The EW (255 ± 40 ka) is a vertical grey box (EW) with a solid lighter grey box (transitional phase) for the start of increased environmental variability. **b**, The percentage pollen plotted on a timeline grouped into arboreal (green), non-arboreal (yellow) and ferns (orange). The darker strips represent sites that contain pollen data, whereas the lighter sections in between represent an estimation of pollen changes. The microcharcoal (black dashed line) correlates with the increase in ferns and decline in arboreal cover. The dark green arboreal sections represent forest disturbance/high turnover taxa such as *Trema*, *Celtis* and *Sapindaceae* are present during the transitional phase and EW. **c**, The percentage of *G. blacki* teeth (red) relative to *P. weidenreichi* teeth (blue) at representative caves as a rough proxy for the relative abundance of *G. blacki* in comparison to *P. weidenreichi* in each site. The relative number of *G. blacki* teeth declines just before the transitional phase representing a change in faunal composition and during the transitional phase representing the extirpation of *G. blacki*. **d,e**, Isotopic changes for fossil *P. weidenreichi* (blue circles and triangles) and *G. blacki* (red circles and triangles) and modern *P. weidenreichi* (blue squares) teeth plotted on a timeline; δ13C (‰) (**d**) and δ18O(‰) (**e**). **f,g**, DMTA boxplot time-series for microwear complexity (**f**) and anisotropy (**g**) of *G. blacki* (red) and *P. weidenreichi* (blue); see Fig. 2c for definitions. **h**, A landscape and environment timeslice demonstrating the change in vegetation and primate species from the pre-EW, through the EW to the post-EW.

the EW in the transitional phase as suggested by the change in forest communities and structures and post-EW as suggested by a decline in arboreal cover and an increase in ferns and grasslands associated with fire. This variability started in a stepwise manner between 1,100 and 350 ka, with dramatic increases from about 200 ka (Fig. 3b). We have interpreted this variability as shifts towards increased seasonality and drier environments, which caused a shift to seasonal

subtropical/tropical moist lowland forests and an increase in shrubs and open grassland environments before and during the EW (Supplementary Information section 10). This environmental variability is also seen in the sedimentary record as the stable low-energy environments of pre-EW were replaced by unstable high-energy environments of the EW with water availability restricted to the wet seasons (Extended Data Fig. 8c and Supplementary Information section 11).

The decline in forest cover during this period is documented in China[21], Southeast Asia[22] and Australasia[23]. However, our pollen study demonstrates that the key to *G. blacki* extinction is not the deterioration in arboreal cover but rather the influence of environmental variability in changing the composition of forest communities, particularly the increase in disturbance taxa. Our stable isotope and trace element data provide new insights into the extent of this variability and the impacts on *G. blacki* (Supplementary Information sections 12 and 13). Pre-EW, *G. blacki* and *P. weidenreichi* both lived in closed canopy forested environments (Fig. 3b and Extended Data Fig. 8b), with stronger biogenic banding (Fig. 2d(i)–(iii)), probably reflecting a larger diversity of food sources, including seasonal fruits and flowers and periodic water consumption, as indicated by the clear lead banding (Extended Data Fig. 10a,b). The most likely food sources would have been in greater availability all year round causing only discrete stress in the population (Fig. 2d(i)–(iii)). With the exception of one individual, throughout the EW period *G. blacki* seems to have maintained a more specialized closed canopy niche, reliant on perhaps a mixture of forest plants (Extended Data Fig. 8b). This specialization during an environmental shift may have caused a more diffused biogenic signal in individuals' dental tissue (Fig. 2d(iv)–(v)), thus suggesting a greatly reduced dietary diversity, less regular water consumption (Extended Data Fig. 10c,d) and increased chronic stress in the population (Fig. 2d(iv)–(v)). This is the first insight into the behaviour of *G. blacki* as a species on the brink of extinction, which is in stark contrast to *P. weidenreichi* (Fig. 2d(vi)) that shows much less stress at this time. Beyond the EW, *P. weidenreichi* seems to have shifted to exploit the more open, seasonal habitats (Extended Data Fig. 8b), perhaps continuing to exploit the seasonal masting of fruit as modern *Pongo* does in Borneo today[24].

The changes in microwear values in *G. blacki* and *P. weidenreichi* teeth may also be linked to periods of fruit scarcity. *G. blacki* tends to show more specific dietary preferences (in both fruits and fibrous foods) indicating greater reliance on fibrous fall-back foods (Fig. 2c), such as over the EW when the climate became more seasonal and less fruits were available. This might have forced *G. blacki* to adapt its diet from higher nutritionally preferred components in lower supply to less nutritional fall-back foods in plentiful supply. The increased consumption of fibrous foods in *P. weidenreichi* over the EW may indicate a better switch to fall-back foods and an overall more flexible and balanced diet (Fig. 2c and Extended Data Fig. 10). This first DMTA analysis on the entire range of *G. blacki* material provides a unique insight into its inability to adapt and its potentially poor choice in fall-back foods.

Our study presents a precise timeline for *G. blacki* presence and extinction. During the pre-EW period, *G. blacki* flourished alongside other primates as a successful specialist (Fig. 3c), enjoying a large diversity of food in a rich evergreen-deciduous forest (Fig. 2d(i)–(ii)) and plentiful water sources (Extended Data Fig. 10a–d) within stable environmental conditions (Fig. 2b). Around 700–600 ka in the transitional phase, there was a shift towards increased seasonality causing a change in forest communities (Fig. 3b), less diversity in food sources (Fig. 2d(iii)–(iv)), unstable high-energy environments (Extended Data Fig. 8c), changes in the composition of the fauna and widespread faunal turnovers (Fig. 3c and Supplementary Information section 2), a shift towards seasonal habitats by *P. weidenreichi* (Extended Data Fig. 8b) and a shift in the dietary diversity and behaviour of *G. blacki* (Figs. 2d and 3f,g).

Despite sharing similar environments with *P. weidenreichi* pre-EW, from 600 to 300 ka there is evidence of the inability of *G. blacki* to

adapt to this transitional period, which had a greater impact on its resilience to the changing ecology. The reliance of *G. blacki* on fruits and lower nutritious fall-back foods (Fig. 2c) created a higher-risk foraging strategy and, combined with its much larger, less mobile body size made it more vulnerable to changes in forest structures[25] (Fig. 2c). Moreover, *G. blacki* was exclusively terrestrial, possibly with a small geographic range[20] but periodically travelled down the valley for water consumption (Extended Data Fig. 10a–d), whereas *P. weidenreichi* was more arboreal, mobile and semisolitary collecting water in the leaf canopy. Furthermore, the unique dentognathic features[13,14] and giant body size[4,5] of *G. blacki* suggest a higher demand in food uptake and slower and more delayed growth pattern, which may imply a lower reproduction rate[26]. Although *G. blacki* increased in tooth size over the Pleistocene, implying an increase in body size also, *P. weidenreichi* decreased[27] making it a more agile adaptor. *P. weidenreichi* also demonstrated a flexibility towards the open habitats (Extended Data Fig. 8b) potentially moving in smaller groups and was able to adjust its behaviour in response to the environmental variability, causing a less stressed population (Fig. 3d).

By about 300 ka there is evidence of a struggling *G. blacki* population as the number of *G. blacki* caves and teeth reduced (Fig. 3c), indicating a dwindling population. The stark change in the teeth banding of *G. blacki* indicates chronic stress in the population (Fig. 2d(iv)–(v)) and changes from its preferred dietary behaviour (Fig. 2c and Extended Data Fig. 10f,g) indicate that *G. blacki* was struggling to respond to the environmental changes on a potentially shrinking territory[20]. It would seem that its forest refugia changed its structure and became too open and disturbed for this species to sustain itself. When compared to other well-known extinction events in North America and Australia influenced by *Homo sapiens*[28–30], there is no evidence to suggest that archaic hominins played a role in this earlier megafaunal extinction event in southern China.

Presenting a defined cause for extinction is a feat that has seldom been achieved for many extinct species as it requires a genus- and species-specific approach[28]. Although determining the exact drivers of megafaunal extirpation and extinction can be highly challenging[29,30], our multiproxy record of *G. blacki* timing, environment and behaviour provides robust regional insights into the ecological context of this species. *G. blacki* was the ultimate specialist and, when the arboreal environments changed, its struggle to adapt sealed its fate. In comparison, the generalist *Homo* extended and diversified across Southeast Asia during this period and seemed to have flexibly exploited the new mosaic environments that posed such a problem to *G. blacki*. Overall, our dataset provides important context for the changing fortunes of different primate species in Southeast Asia, shedding new light onto the demise of the largest primate ever to have roamed the planet.

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

## Methods

### Speleology and excavation techniques

Caves were discovered using a combination of local knowledge, field survey, drone mapping surveys and targeted reconnaissance with a dedicated caving team. Excavation grids were set up on the basis of the shape of cave passages and distribution of fossil-containing deposits. Jackhammers were used to break blocks of fossil-breccia and the fossils were extracted using geological hammers. Fine cleaning, identification and cataloguing were conducted in the Institute of Vertebrate Paleontology and Paleoanthropology (IVPP) laboratories.

### Luminescence dating with pIR-IRSL/SG quartz

Large bulk samples of the fossil-bearing breccia were sampled in situ under subdued red-light conditions from each cave site (Supplementary Fig. S1a–v) and processed using the standard sample purification procedures for quartz and feldspar separation including a 40% and 10% wash in hydrofluoric acid for 45 and 10 min, respectively[31]. All luminescence analysis was conducted at the 'Traps' luminescence dating facility at Macquarie University in Sydney, Australia. Single aliquots of 90–125 µm feldspar and 180–212 µm single grains (SGs) of quartz or feldspar were processed in a Riso TL-DA-20 containing an automated DASH set up with a dual laser single-grain attachment and a blue/UV sensitive photomultiplier tube (PDM9107Q-AP-TTL-03) using either the blue filter pack (Schott BG-39 (2 mm) and Corning 7-59 (4 mm) filters for feldspar or U340s (2× Hoya U340 3.5 mm) for quartz. Feldspar equivalent doses were corrected according to the results of the anomalous fading tests (using a weighted mean fading rate of 2.0 ± 0.2% per decade)[32] but no residual corrections were undertaken and both feldspar and quartz $D_e$s were then run through a minimum age model[33] to identify the population that had the most bleaching before burial.

Measurements of $^{238}$U, $^{235}$U, $^{232}$Th and $^{40}$K were estimated using Geiger-Muller multicounter beta counting and thick source alpha counting of dried and powdered sediment samples in the laboratory, combined with in situ gamma spectrometry in the field. The corresponding (dry) beta and gamma dose rates were obtained using the conversion factors of ref. 34 and the beta-dose attenuation factors of ref. 35. Cosmic-ray dose rates were estimated from published relationships[36], making allowance for the sediment overburden at the sample locality (ranging from 3.99 to 0.20 m), the altitude (ranging from 2,000 to 176 m above sea level) and geomagnetic latitude and longitude of the sampling sites.

### U-series dating of teeth

A total of 22 *G. blacki* and 9 *P. weidenreichi* sp. fossil teeth were analysed for U-series dating. Uranium series measurements were undertaken by laser ablation combined with multicollector inductively coupled plasma mass spectrometer (MC-ICP-MS) at GARG-SCU according to protocols in ref. 37. Laser ablation was performed with a New Wave Research 213 nm laser and thorium ($^{230}$Th, $^{232}$Th) and uranium ($^{234}$U, $^{235}$U, $^{238}$U) isotopes were measured on a Thermo Neptune XT MC-ICP-MS. Teeth were ablated using rasters of 5 min each and were measured with standards before and after.

### Coupled US-ESR dating of teeth

Enamel fragments from each tooth dated by coupled US-ESR techniques were separated using a hand-held diamond saw following the protocol developed by ref. 38. Fragments were then measured at room temperature on a Freiberg MS5000 ESR X-band spectrometer at a 0.1 mT modulation amplitude, ten scans, 2 mW power, 100 G sweep and 100 KHz modulation frequency. Each fragment was irradiated, following exponentially increasing irradiation times. Sediment elemental concentrations, external beta and gamma dose rate contributions and water content were obtained from in situ measurements. The external beta-dose rates have been extrapolated from the U, Th and K contents measured on a portion of sediment subsample (about 8 g). The external gamma dose rates were determined using a portable gamma spectrometer at each site.

### ESR dating of quartz

A total of seven samples of purified quartz (previously prepared at Macquarie University) were analysed for ESR dating purpose. For a couple of them (CBAK10 and CZW2), two grain-size fractions were measured. Quartz grains were dated by means of the multiple aliquots additive dose method and following the multiple ventre approach initially proposed by ref. 39. In each sample, the ESR signals of both the aluminium (Al) and titanium (Ti) centres were either acquired in separate spectra using specifically optimized parameters (standard CENIEH procedure, for example, ref. 40) or in a single spectrum (for example, ref. 41). Gamma irradiations and ESR measurements were performed at the National Research Centre on Human Evolution (CENIEH), Spain, using a Gammacell-1000 and an EMXmicro 6/1Bruker X-band ESR spectrometer, respectively.

### U-series dating of carbonates

Separate subsamples were drilled from the fresh cross-section of a hand specimen of the in situ flowstone using a hand-drill. The powdered subsamples were subjected to chemical treatment and isotopic measurements by mass spectrometry[42]. U-series dating of most speleothem samples was conducted in the Radiogenic Isotope Facility (The University of Queensland) using a Nu Plasma MC-ICP-MS. Analytical procedures followed previous publications for MC-ICP-MS[43–45]. $^{230}$Th/$^{234}$U ages were calculated using Isoplot EX 3.75 (ref. 41) and half-lives of 75,690 years ($^{230}$Th) and 245,250 years ($^{234}$U)[46]. Analyses were also undertaken by laser ablation MC-ICP-MS at the Wollongong Isotope Geochronology Laboratory, University of Wollongong[47]. Laser ablation was performed with a New Wave Research ArF 193 nm Excimer laser, equipped with a TV2 cell.

### Modelling

To evaluate the uncertainties of the integrated dating approach to the site (Supplementary Tables 1–4), Bayesian modelling was performed on all independent age estimates using the OxCal (v.4.4) software 52 (https://c14.arch.ox.ac.uk/oxcal.html)[48]. The analyses incorporated the probability distributions of individual ages, constraints imposed by stratigraphic relationships and the reported minimum or maximum nature of some of the individual age estimates. Each individual age was included as a Gaussian distribution (with mean and s.d. defined by the age estimate and their associated uncertainties) and the resulting age ranges for each unit were presented at 1$\sigma$. The code used for each site is publicly available in Zenodo (10.5281/zenodo.10077255).

### Pollen analysis

Pollen analysis followed a modified standard methodology described by ref. 49, in which sediment was dispersed in Calgon (3%) treated with HCl (10%) and sieved at >125 µm, allowed to settle in HL (heavy liquid/LST-lithium heteropolytungstates) at a density of 2.01 SG and centrifuged, then acetolysis which removes cellulose and stains the pollen followed. The remaining sample was then mounted on slides with glycerol. Pollen identification was aided by the Australasian Pollen and Spore Atlas (online resource[50]) and a handbook of quaternary pollen and spores in China[51]. Macrocharcoal analysis followed the methodology outlined by ref. 52.

### Microstratigraphy and spectroscopy

Five intact cave blocks were sampled for the purposes of a range of synergistic microcontextualized analyses. First, a microstratigraphic study was undertaken using petrographic microscopy (for

example, refs. 53–55). Sediment blocks were prepared at the Flinders University Microarchaeology Laboratory and ten glass thin sections (76 mm × 50 mm × 30 μm) were made by Adelaide Petrographics. Thin sections were analysed using a Leica DM2700 P (Wetzlar) polarizing microscope following the terminology of ref. 56. Alkalinity (pH), X-ray diffraction (XRD)[57,58] in an Aeris Malvern Panalytical benchtop X-ray diffractometer (2018, The Netherlands) and X-ray fluorescence (XRF)[59,60] in an Axios Malvern Panalytical WD-XRF spectrometer tests were applied to the microsampled bulk sediments subsamples at Macquarie University.

### Stable isotope analysis

A total of 27 teeth (15 fossil *G. blacki* and 7 fossil and 5 modern *P. weidenreichi* teeth) were cleaned using an air abrasion system. Enamel powder for bulk analysis was obtained using a diamond-tipped drill. All enamel powder was pretreated following established protocols[23,61]. Following reaction with 100% phosphoric acid, gases evolved from the samples were analysed for their stable carbon and oxygen isotopic measurements using a Thermo Gas Bench 2 connected to a Thermo Delta V Advantage Mass Spectrometer at the Max Planck Institute for Geoanthropology (formerly for the Science of Human History). The δ13C and δ18O values were compared against International Standards. Overall measurement precision was studied through the measurement of repeat extracts from a bovid tooth enamel standard ($n = 30$, ±0.2‰ for both δ13C and δ18O values).

### Trace element analysis of teeth

Fossil teeth were sectioned with a high-precision diamond saw and polished to more than 10 μm smoothness. Laser ablation ICP-MS was used for trace elemental mapping analyses of the samples according to the published protocol from ref. 62. The GARG facility at Southern Cross University uses an ESI NW213 coupled to an Agilent 7700 ICP-MS, using rastered laser beams run along the sample surface in a straight line. A laser spot size of 40 μm, a scan speed of 80 μm s⁻¹, laser intensity of 80% and a total integration time of 0.50 s were used to produce data points.

### DMTA

DMTA was applied to facet 9, as close as possible to the (ante mortem) tip crushing point of moderately worn (wear stages 2 to 4; ref. 63) first (m1), second (m2) and third (m3) lower molars of extinct *G. blacki* ($n = 16$), extinct *P. weidenreichi* ($n = 22$) (IVPP) and extant *P. pygmaeus* ($n = 3$) (South Australian Museum). Sample size was restricted by fossil availability. Cleaning, moulding with polyvinylsiloxane and casting with epoxy resin followed standard DMTA procedures[64–68]. Scanning of 242 × 181 μm² areas was conducted on a Sensofar PLμ neox confocal profiler at the Flinders University Palaeontology Microscopy facility. Axonometric digital elevation models were fabricated in SensoMAP Premium 8.2.9564 following the 'soft filter procedure'[68] and analysed with the embedded scale-sensitive fractal analysis module. Statistical analyses and data visualization were carried out in Minitab 19.2020.1 and R Studio 1.4.1717.

### Reporting summary

Further information on research design is available in the Nature Portfolio Reporting Summary linked to this article.

### Data availability

The data that support the findings of this study are included in the Supplementary Information. More raw data are available from publicly available Zenodo data repositories: dating https://doi.org/10.5281/zenodo.10080908, and environment and behaviour https://doi.org/10.5281/zenodo.10080973. Source data are provided with this paper.

### Code availability

Custom codes for the OxCal program used for the Bayesian modelling in this study are publicly available in Zenodo: https://doi.org/10.5281/zenodo.10077255. An example of this code is included in Supplementary Fig. 11.

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

**Acknowledgements** This research was funded by the Australian Research Council with a Discovery grant to K.E.W., Y.Z., S.H. and R.C. (DP170101597), Future Fellowship (FT180100309) to M.W.M., LIEF (LE130100115) to G.G. and G.P. and by the Chinese Academy of Sciences Strategic Priority Research grant to Y.Z., K.E.W. and R.J.-B. (XDB26030303). Aspects of the ESR dating study on quartz have been covered by the Ramón y Cajal Fellowship RYC2018-025221-I granted to M.D. and funded by MCIN/AEI/10.13039/501100011033 and 'ESF Investing in your future'. P.R. would like to thank the Max Planck Society for funding and support. The DMTA and microstratigraphical analyses was supported by W. V. Scott Estate Charitable Trust Fund (123402902) and a Ruggles-Gates Scholarship (125201851) by Royal Anthropological Institute (United Kingdom) grants to J.K.L. The research in the Bubing Basin was funded by the National Social Science Foundation of China (20&ZD246) and the BaGui Scholars Project of the Guangxi. We wish to acknowledge the IVPP for providing support and access to the excavated fossil material, the Chongzuo Zhuang Ethological Museum and the Natural History Museum of Guangxi for providing logistical and access support, C. Jin for previous excavation work and for inspiring this research, Q. Cui and the Beijing Cavers for their tireless cave reconnaissance, the local excavators at both Chongzuo and Bubing Basin, particularly X. Qi, X. Liu, F. Tian, C. Huang and Luo brothers. K.E.W. is grateful to K. Tewari for assistance with sample processing in the luminescence dating facility at Macquarie University and G. Prideaux at Flinders University for his supervisory and facilitatory role in DMTA analysis. M.D. is grateful to M. J. A. Escarza, CENIEH, for technical support throughout the analytical procedure. A. A. Gallo, CENIEH, performed the bulk XRD analyses of the quartz samples. J.K.L. is grateful to V. Hernandez and C. McAdams for microstratigraphic data interpretation, S. Löhr and P. Wieland for help with XRD and XRF, S. Murray for particle size analysis, C. Burke for dental casting, X. Liu for data visualization and P. Petocz for statistical aspects. Permission for excavation and sampling were obtained by the IVPP through a permit issued by the National Bureau of Cultural Relics in Beijing no. 3312044 (2010 to the present) and through a collaborative agreement in conjunction with the Chongzuo Museum (signed 8 October 2018). Sediment samples were exported to Australia using biosecurity permit nos. IP0000522151 and IP0002282095. *G. blacki* teeth were imported to Australia by Y.Z. with permission from IVPP and the Chinese Academy of Sciences permit no. [2018]9335 for the purposes of dating and analysis.

**Author contributions** Y.Z., K.E.W., R.C., R.J.-B., J.K.L. and S.H. explored and excavated the sites and collected faunal and dating samples. K.E.W. conducted the OSL and pIR-IRSL dating. M.D. conducted ESR dating on the sediment. R.J.-B. conducted the US and coupled US-ESR dating of the teeth. J.-x.Z., A.D. and Y.C. conducted the U-series measurements on the speleothem. M.B. and R.J.-B. conducted the trace element analysis. P.R. and M.L. conducted stable isotope analysis and J.K.L. and G.A.G. conducted the DMTA analysis all on the fossil teeth. S.H. and S.R. conducted the palynological analysis and J.K.L. and M.W.M. conducted the microsedimentological analyses, both on the cave breccia. Y.Z. and Y.P. analysed the fauna and W.W., L.W. and L.Y. helped to find access and sample the sites. K.E.W. designed the dating approach and conducted the Bayesian modelling. K.E.W. and Y.Z. wrote the paper with contributions from all co-authors.

**Competing interests** The authors declare no competing interests.

**Additional information**
**Correspondence and requests for materials** should be addressed to Yingqi Zhang, Kira E. Westaway, Wei Wang or Renaud Joannes-Boyau.

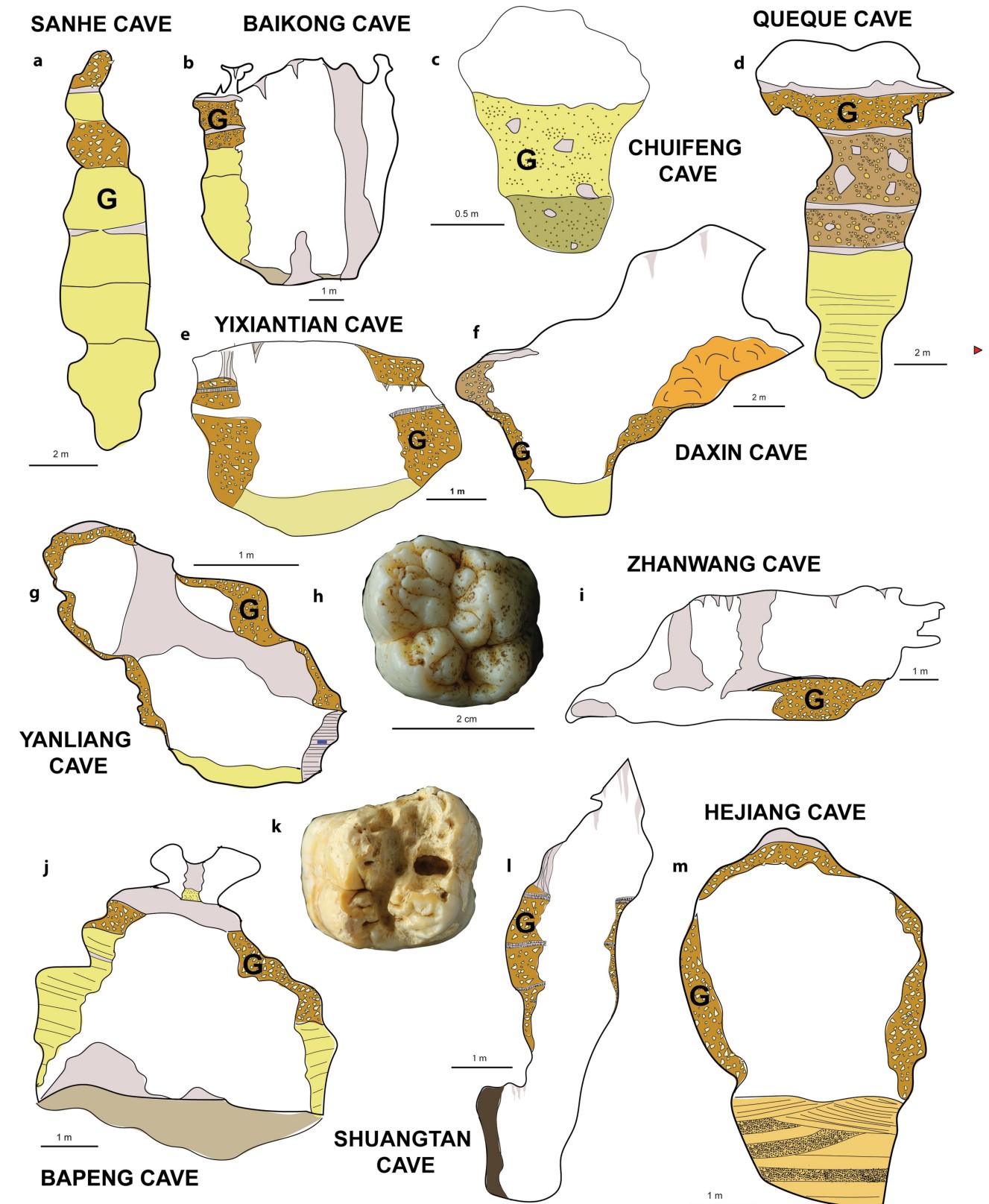

**Extended Data Fig. 1 | The *G. blacki* caves in Chongzuo and Bubing Basin.**
**a**, Sanhe; **b**, Baikong; **c**, Chuifeng; **d**, Queque; **e**, Yixiantian; **f**, Daxin; **g**, Yanliang;
**h**, a typical *G. blacki* teeth from older evidence from Sanhe cave; **i**, Zhanwang;
**j**, Bapeng; **k**, a typical *G. blacki* teeth from younger evidence from Hejiang cave;
**l**, Shuangtan; and **m,** Hejiang. All profiles display the main stratigraphic
relationships and the units containing *G. blacki* evidence. See the individual
cave plans, profiles and sampling locations plus the keys for the symbols in
Fig. S1a–v.

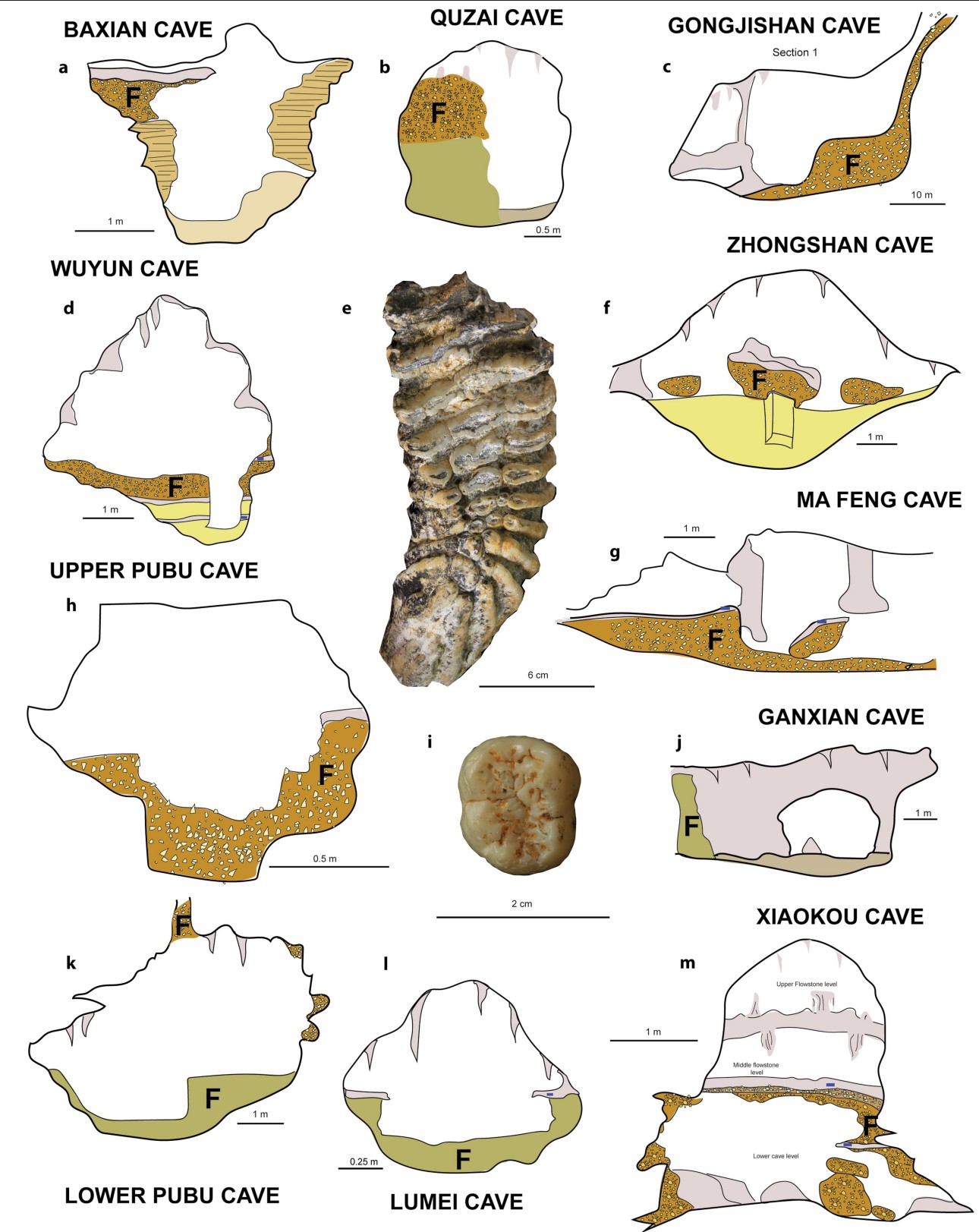

**Extended Data Fig. 2 | The non-*G. blacki*-bearing caves in Chongzuo and Bubing Basin. a**, Baxian; **b**, Quzai; **c**, Gongjishan; **d**, Wuyun; **e**, typical fauna from Baxian Cave – *Elephas maximus*, left M3 (CFLSY201104-2-1); **f**, Zhongshan; **g**, Ma Feng; **h**, Upper Pubu; **i**, typical fauna from Baxian Cave – *Pongo weidenreichi*, left M2, (CFLSY201011-23-871); **j**, Ganxian; **k**, Lower Pubu; **l**, Lumei; **m**, Xiao Kou. All profiles display the main stratigraphic relationships and the units containing fossil evidence but no evidence of *G. blacki*. See the individual cave plans, profiles and sampling locations plus the keys for the symbols in Fig. S1a–v.

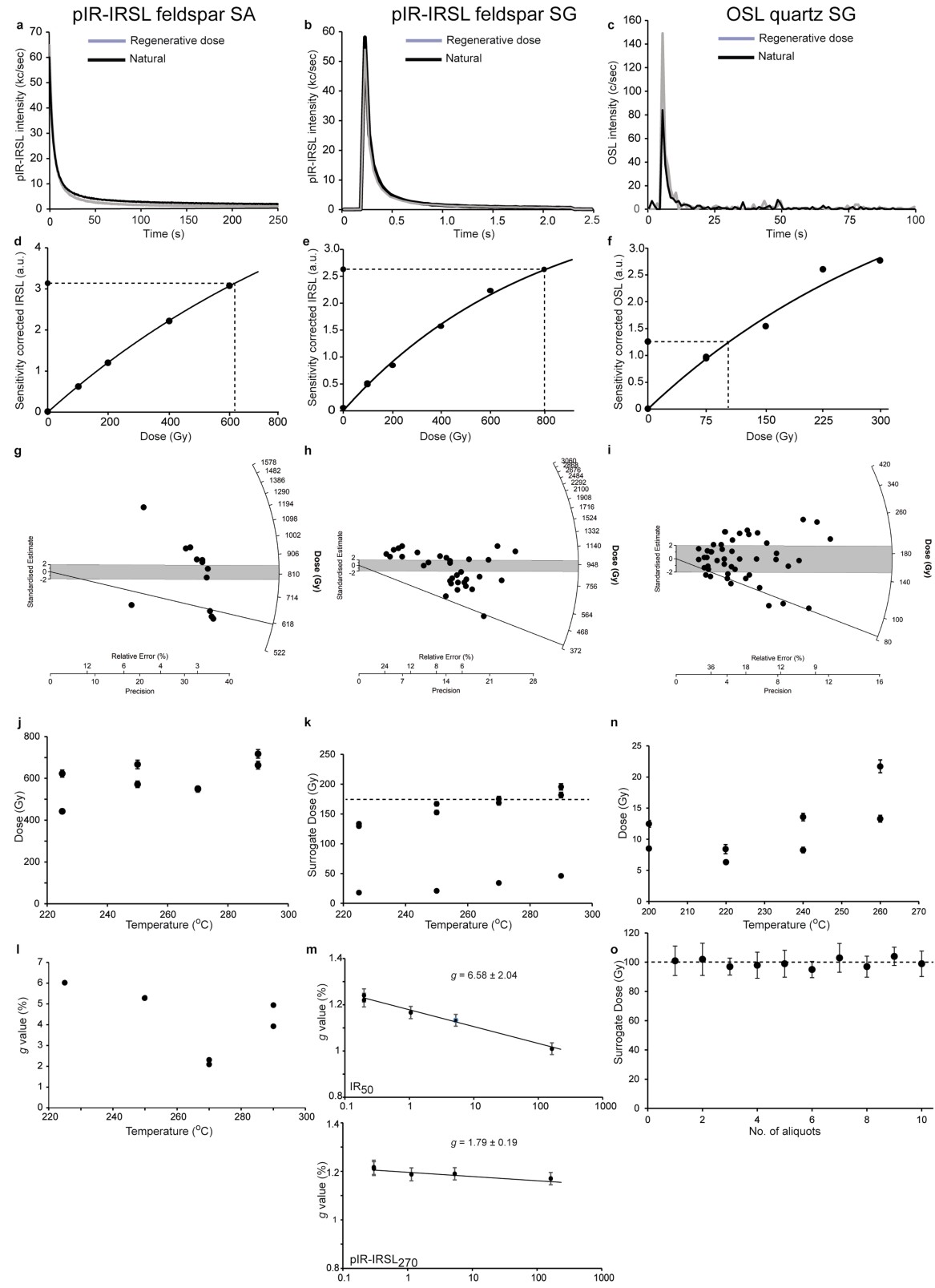

**Extended Data Fig. 3 |** See next page for caption.

**Extended Data Fig. 3 | The luminescence characteristics of the quartz and feldspar sampled from caves in the Guangxi region of southern China.** The resulting data from the application of three luminescence dating techniques; pIR-IRSL SA feldspars, pIR-IRSL SG feldspars and OSL SG quartz to typical samples from three different caves have been displayed. **a**, **d**, **g**, pIR-IRSL data for a single-aliquot from sample HEJ1 from Hejiang cave (a *G. blacki* bearing cave): **a**, a shine down curve including both the natural and regenerative dose; **d**, the dose response of the sample aliquot with a $D_e$ of 631 ± 18 Gy, a recycling ratio of 1.04 ± 0.03 and a D0 of 1136 ± 139 Gy; **g**, the corresponding radial plot for this sample with an uncorrected MAM of 615 ± 11 Gy (n = 12 aliquots). **b**, **e**, **h**, pIR-IRSL single-grain data for sample GONG-16 from Gongjishan cave (a non-*G. blacki* bearing cave): **b**, single-grain shine down including both the natural and regenerative dose; **e**, the dose response of the sample aliquot with a $D_e$ of 801 ± 51 Gy, a recycling ratio of 1.06 ± 0.05 and a D0 of 787 ± 124 Gy; **h**, the corresponding radial plot for this sample with an uncorrected MAM of 388 ± 23 Gy (n = 32 grains). **c**, **f**, **i**, OSL quartz single-grain data for sample CLPB-5 from Lower Pubu cave (a non-Giganto bearing cave): **c**, a single-grain shine down curve including both the natural and regenerative dose; **f,** the dose response of the sample aliquot with a $D_e$ of 108 ± 14 Gy, a recycling ratio of 1.03 ± 0.17 and a D0 of 378 ± 188 Gy; **i**, the corresponding radial plot for this sample with a MAM of 64 ± 7 Gy (n = 45 grains). **j-m**, Preheat, dose recovery, bleaching and fading tests for the pIR signal at the four preheat and pIR-IRSL stimulation temperature combinations (outlined in the supp) using sample CSH1 (n = 8). **j**, A preheat plateau test with the 270 °C pIR-IRSL stimulation and preheat temperature of 300 °C producing the least scatter between aliquots (n = 12). **k**, A dose recovery test with a surrogate dose of 175 Gy, with the 270/300 °C combination recovering the dose within error margins, also shown are the residual values (in Gy) after a 4 hours bleach using a solar simulator (n = 8). **l**, The difference in *g* values as a measure of fading at the 4 different combinations. **m**, An example of the difference in fading between the $IR_{50}$ measurement and $pIR_{270}$ measurement (n = 8). **n**, **o**, procedural tests for the OSL signal conducted on small aliquots: **n**, preheat plateau test using 2 aliquots per temperature point (n = 8), 220 °C was the chosen temperature; **o**, a dose recovery test using a surrogate dose of 100 Gy (n = 10) and the chosen preheat of 220 °C on 10 small aliquots – all aliquots returned the surrogate dose within standard errors. All data points from j-o are mean values with s.d. at 1σ uncertainties.

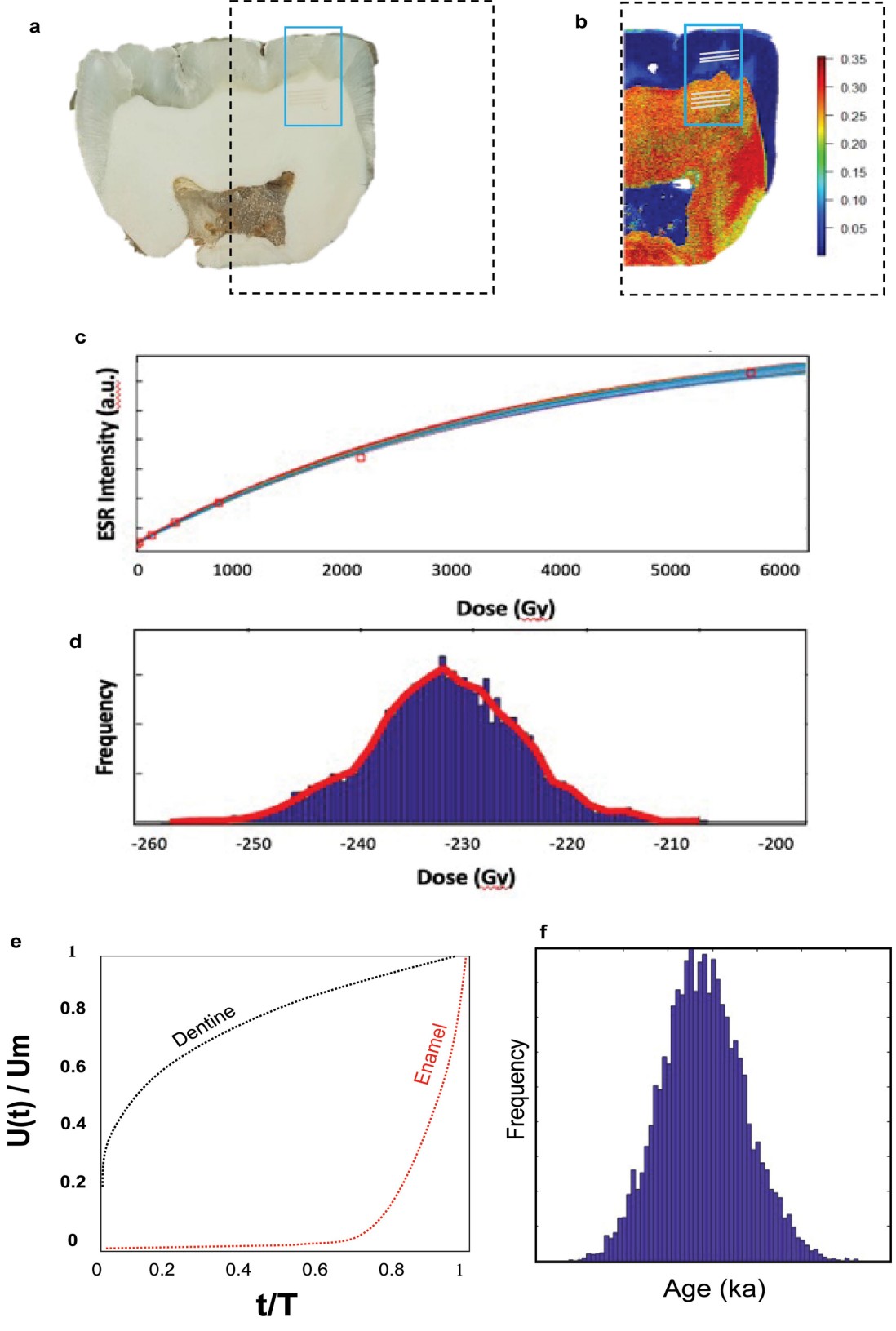

**Extended Data Fig. 4 | Coupled US-ESR dating protocol. a**, Section of *Gigantopithecus blacki* tooth (Baikong 1096), with dotted black rectangle describes the area analysed for uranium distribution, while the blue rectangle highlights the uranium series rasters position. Uranium rasters are placed parallel to each other with a 200 micron spacing to avoid contaminations. Surface prior to analyses are cleaned with a preablation run at high-speed (200microns/s) and lower energy (-10% of ablation energy). **b**, LA-ICP-MS elemental mapping of the uranium distribution and diffusion pattern into the dental tissues of Baikong 1096; the pattern observed is typical of a diagenetic diffusion signal with heterogenous distribution. **c**, Dose Response Curve (DRC) modelling using Monte Carlo algorithm (Baxian RTK201306-77). **d**, Dose equivalent distribution frequency of Baxian RTK201306-77 according to the DRC simulation, using the McDoseE 2.0 software from ref. 69. **e**, Model of uranium diffusion over time for both dentine and enamel of Zhanwang O1Z1 sample. **f**, Coupled US-ESR Age distribution frequency using[70] programme of Zhanwang O1Z1 sample.

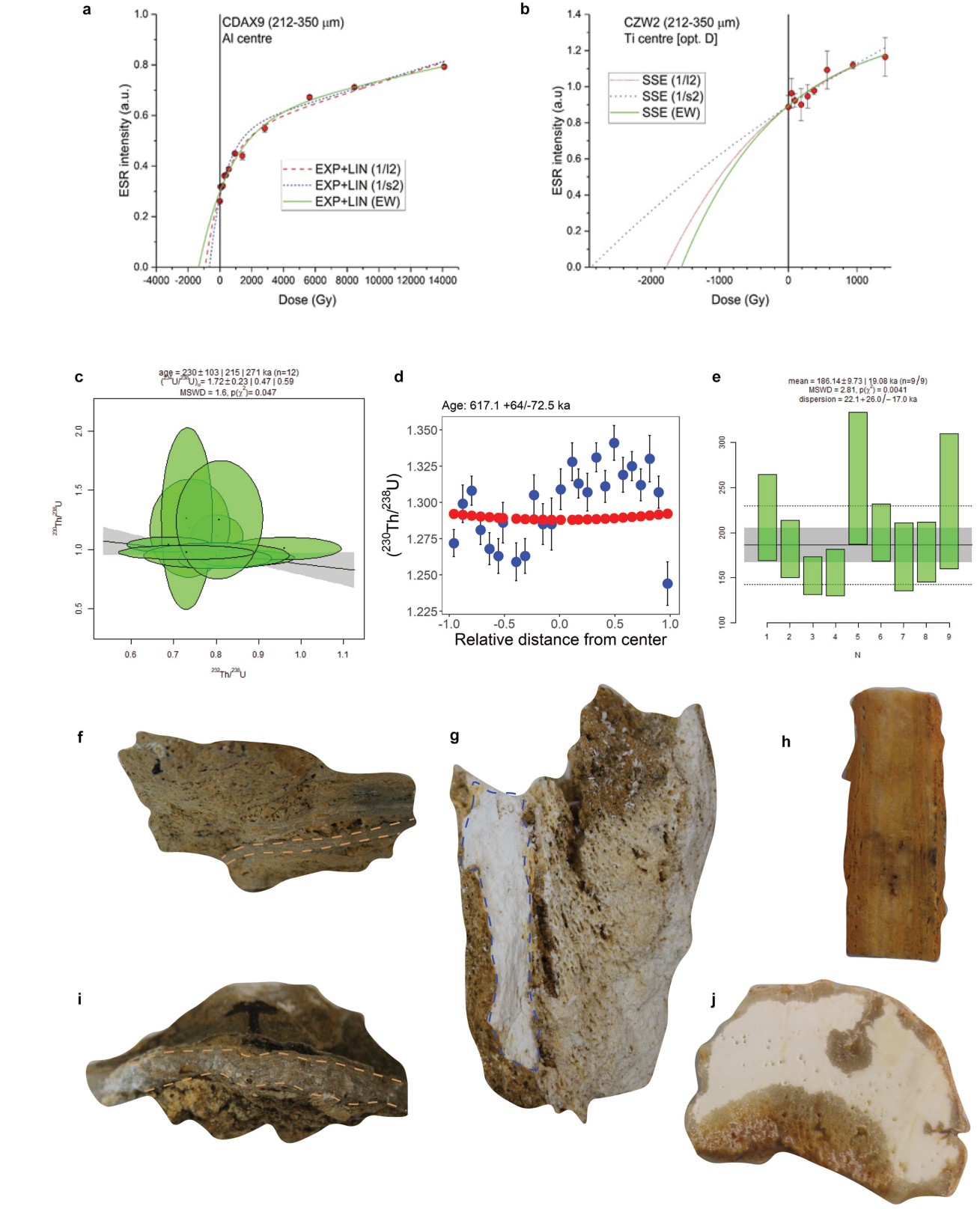

**Extended Data Fig. 5** | See next page for caption.

**Extended Data Fig. 5 | ESR quartz and U-series on carbonates and bone.**
**a**, ESR dose response curve for CDAX9 (212-350 µm) obtained from the measurement of the Al centre. Experimental data points represent mean ESR intensities and associated 1 standard deviation (vertical error bars), derived from the repeated measurements (n = 3; see Table S18). **b**, ESR dose response curve for CZW2 (212-350 µm) obtained from the measurement of the Ti centre. Experimental data points represent mean ESR intensities and associated 1 standard deviation (vertical error bars) derived from the repeated measurements (n = 3; see Table S19). **c**, U-series series isochron plot for sample WUY-F1 providing an age estimate of 230 ± 103 ka. **d**, Diffusion absorption density (DAD) modelling[197] of the U-series ages for a bone from Yanliang Cave

(image in **g**) providing an age estimate of 617 + 64/-73 ka, n = 10, with each point representing the mean age with s.d. at 2 σ uncertainties. **e**, A weighted mean plot of U-series ages for sample CMF-F3 providing an age estimate of 186 ± 10 ka. **f**, A flowstone sample from Shuangtan Cave (SHTC-F1C) with the purest layer depicted by a brown dashed line providing an age estimate of 330 ± 33 ka. **g**, A bone encased in breccia deposits sampled from Yanliang Cave (data in **b**). **h**, A flowstone sample from Ma Feng cave (CMF-F3) providing an age estimate of 186 ± 9 ka. **i**, A flowstone sample from Baxian cave (BAX-F2a) with the purest layer depicted by a brown dashed line providing an age estimate of 128 ± 3 ka. **j**, The bone sampled from Yanliang (in g) after being cleaned and cut prior to sampling.

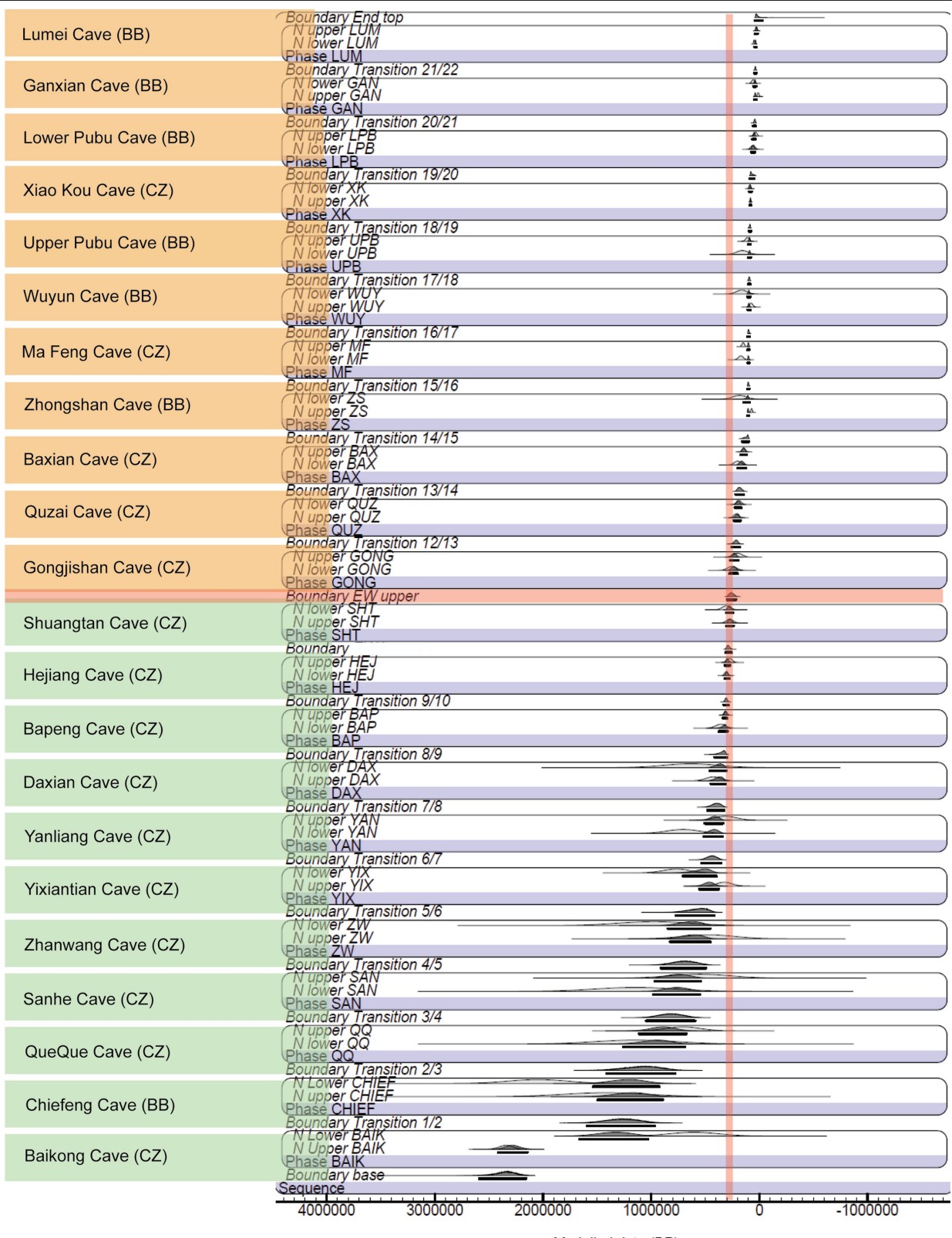

**Extended Data Fig. 6 | Modelling of all 22 sites to establish the timing of the Extinction Window.** The master Bayesian plot composed of the modelled age ranges for all 22 caves. The names of the caves have been included on the left – colour coded to signify the *G. blacki*-bearing caves (green) and the non-*G. blacki*-bearing caves (brown). Following each cave name its location is signified in brackets with either CZ – Chongzuo area or BB – Bubing Basin. See Fig. 1 for the locations of these areas. The master model has been presented at 2 σ error margin with the age range of each cave representing the upper and lower boundary of the fossil-bearing layer. The extinction window of *G. blacki* lies in the boundary between Shuangtan and Gongjisgan Caves (red horizontal line) and has been modelled between 295–215,000 yrs ago (255 + −40 ka) (red vertical line).

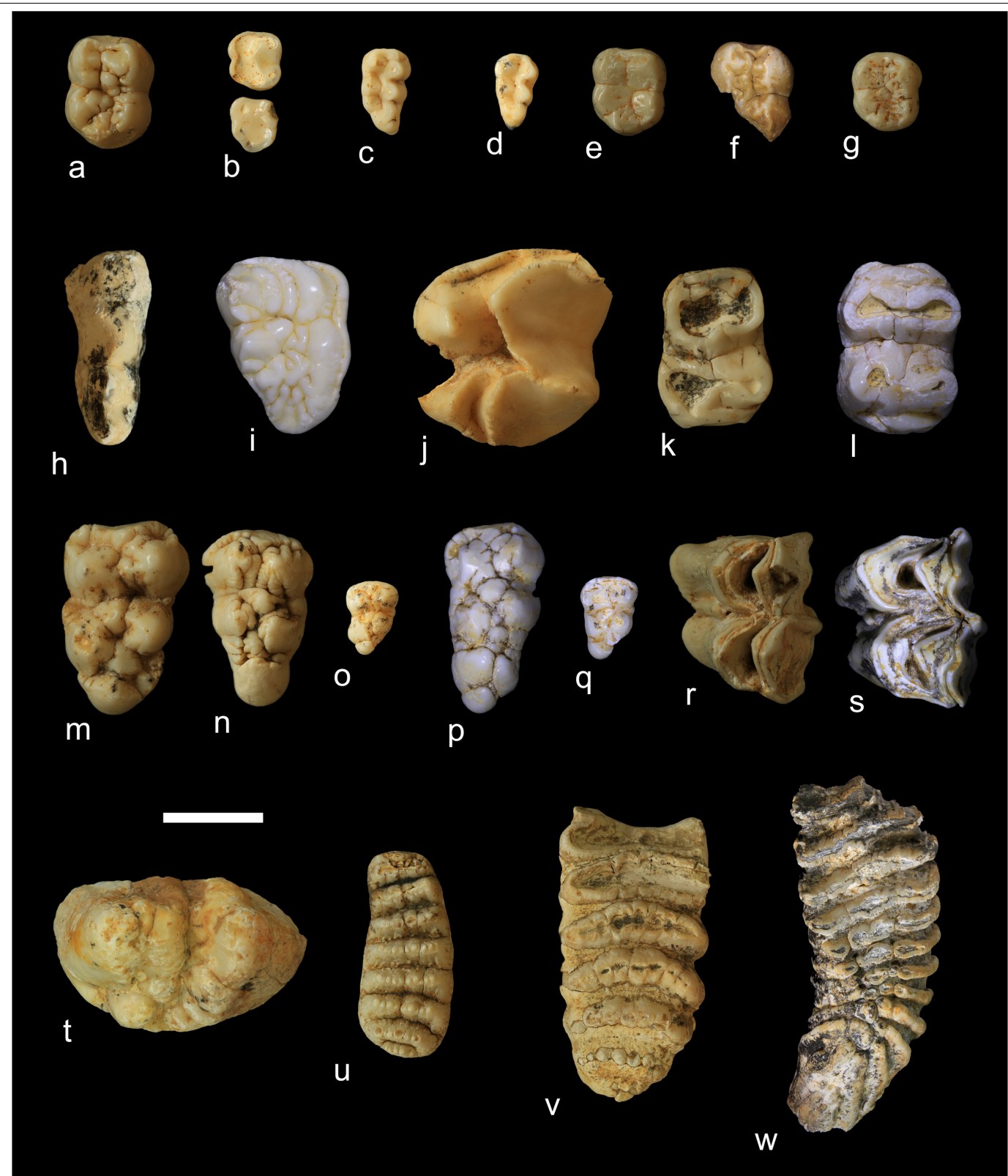

**Extended Data Fig. 7** | See next page for caption.

Extended Data Fig. 7 | Typical fossils of the selected Chongzuo Pleistocene faunas. Baikong: **a**, *Gigantopithecus blacki*, left M2, CLBBD201011-31-MT33; **b**, Mystery ape, left M2-3, CLBBD201011-32-867, 932; **c**, Procynocephalus cf. wimani, right M3, CLBBD201011-32-976; **d**, Macaca sp., right M3, CLBBD201011-32-977; **e**, *Pongo weidenreichi*, left M2, CLBBD201011-32-875; *H. Pachycrocuta brevirostris licenti*, left P4, CLBBD201011-24-2026; **j**, Hesperotherium cf. sinense, left M2, CLBBD201011-7-523; **k**, Tapirus cf. sanyuanensis, left M1, CLBBD201011-27-2442; **m**, Hippopotamodon ultimus, left M3, CLBBD201011−1-31; **n**, Sus peii, right M3, CLBBD201011-1-62; **o**, Sus xiaozhu, left M3, CLBBD201011-27-2442; **r**, Megalovis guangxiensis, left M2, CLBBD201011-19-1808; **t**, Sinomastodon yangziensis, molar fragment, CLBBD201011-23-1913; **u**, Stegodon huananensis, right M3, CLBBD201011-22-1903. **Yixiantian**: **f**, *Gigantopithecus blacki*, right M2, CFLSY201104-2-17; **g**, *Pongo weidenreichi*, left M2, CFLSY201011-23-871; **i**, Ailuropoda baconi, left M2, CFLSY201011-20-2166; **l**, Megatapirus augustus, left M1, CFLSY201104-2-3; **p**, Sus peii, left M3, CFLSY201011-1-9; **q**, Sus xiaozhu, left M3, CFLSY201011-6-385; **s**, Megalovis guangxiensis, left M2, CFLSY201011-9-667; **v**, Stegodon orientalis, left dM4, CFLSY201011-22-2278; **w**, Elephas maximus, left M3 CFLSY201104-2-1. Scale bar = 2 cm for **a**–**t**, 6 cm for **u**–**w**.

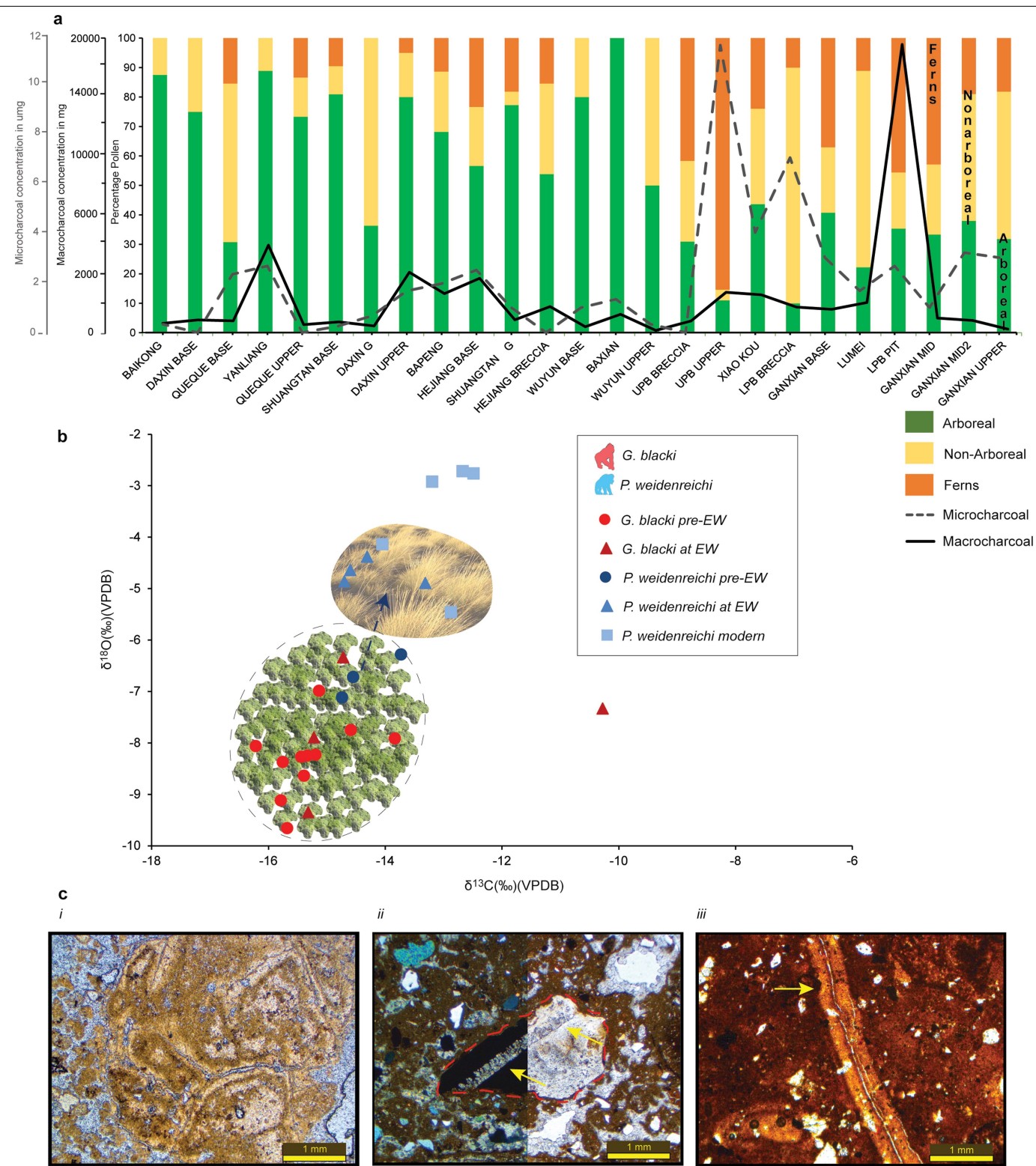

**Extended Data Fig. 8 | Palaeoenvironmental proxies used in this study.**
**a**, Pollen and charcoal analysis plotted as percentage pollen divided into
arboreal, non-arboreal and ferns. Micro (grey dashed line) and macrocharcoal
(black solid line) have been plotted over the top to aid comparison. **b**, Stable
isotope analysis of fossil and modern *P. weidenreichi* and *G. blacki* teeth. The
data points have been divided into arboreal dominated and more open
grasslands dominated isotopic values and display a shift in ecological
preference by *P. weidenreichi* but not *G. blacki*. **c**, A selection of micrographs
from the pre-EW (Queque Cave), EW (Shuangtan Cave) and post-EW (Mafeng
Cave). ***i***: Weathered coprolite in calcite rich-clay matrix, indicating faunal
habitation at Queque Cave during stable, lower energetic, environmental

conditions (2.5×, PPL). We used the point-counting method to assess the
relative abundance of a particular feature and test for reproducibility between
point-counts. Micrograph from thin section CQQ-A. ***ii***: Phosphatized bone
fragment with calcite inclusion (arrows) in a sandy matrix at Shuangtan Cave,
indicating unstable, higher energetic, environmental conditions (2.5×, XPL left
and PPL right). Micrograph from thin section CSHT-A. ***iii***: Lighter coloured
sandy-silty banded remnant feature with (Fe, Ti and/or Al) oxide stains in
clayish matrix at Mafeng Cave, indicating a recovery from unstable, higher
energetic to stable, lower energetic, environmental conditions (2.5×, PPL).
Micrograph from thin section CMF-B.

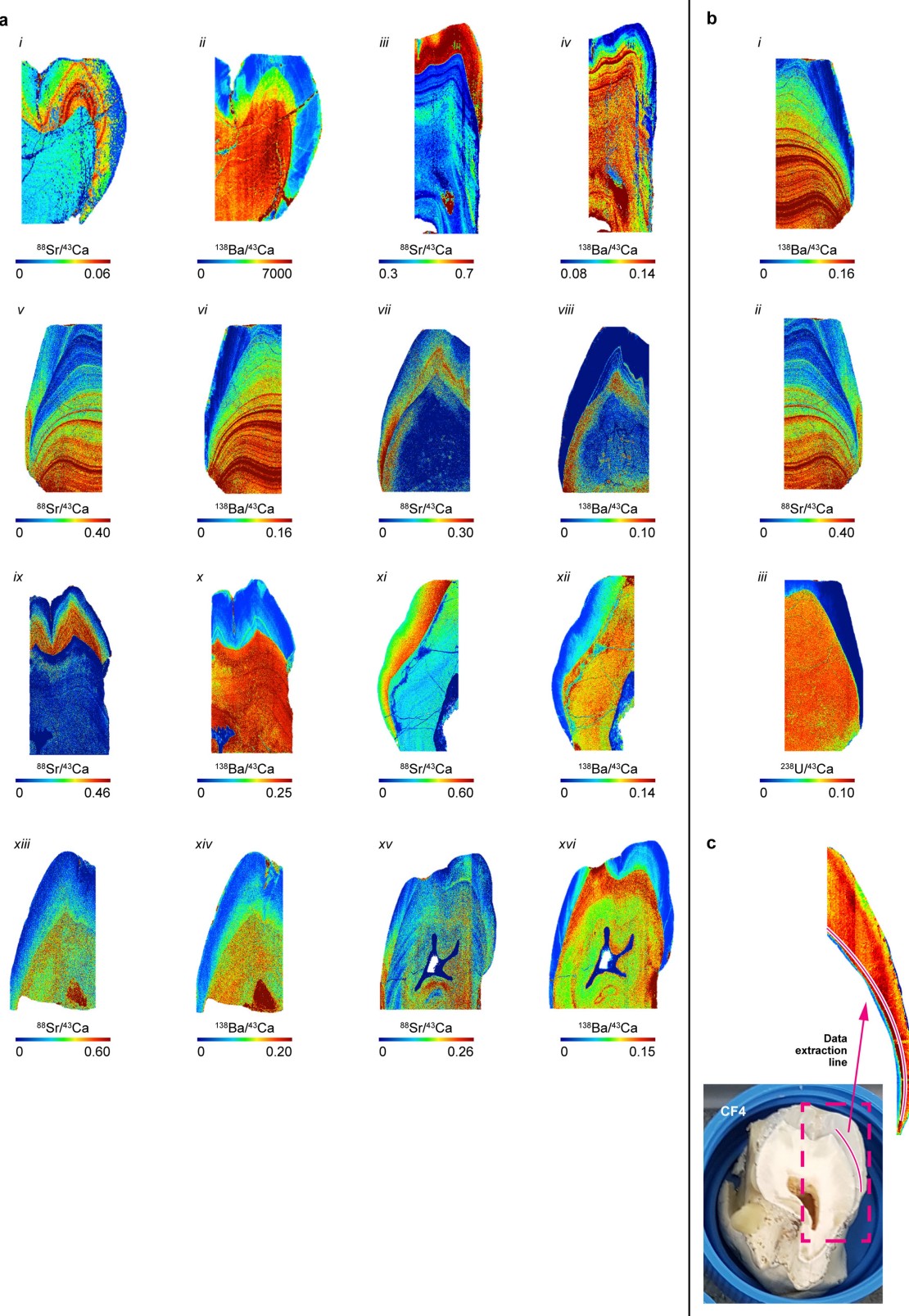

**Extended Data Fig. 9 | A selection of strontium and barium elemental maps.** Produced from *Gigantopithecus blacki*. Fossil teeth from various cave sites. **a**, *i-ii*: Chuifeng 1; *iii-iv* Chuifeng 2; v-vi Queque 44; *vii-viii*: Queque 5; *xi-xii*: Shuangtan 50; *xiii-xiv*: Shuangtan 109; *xiv-xvi*: Hejiang 1573.8. **b**, Uranium as a marker of diagenesis. Queque 44 shows the expected distribution of uranium, mirroring the barium and strontium. **c**, Data extraction location of each sample, taken along the enamel, parallel to the enamel-dentine junction.

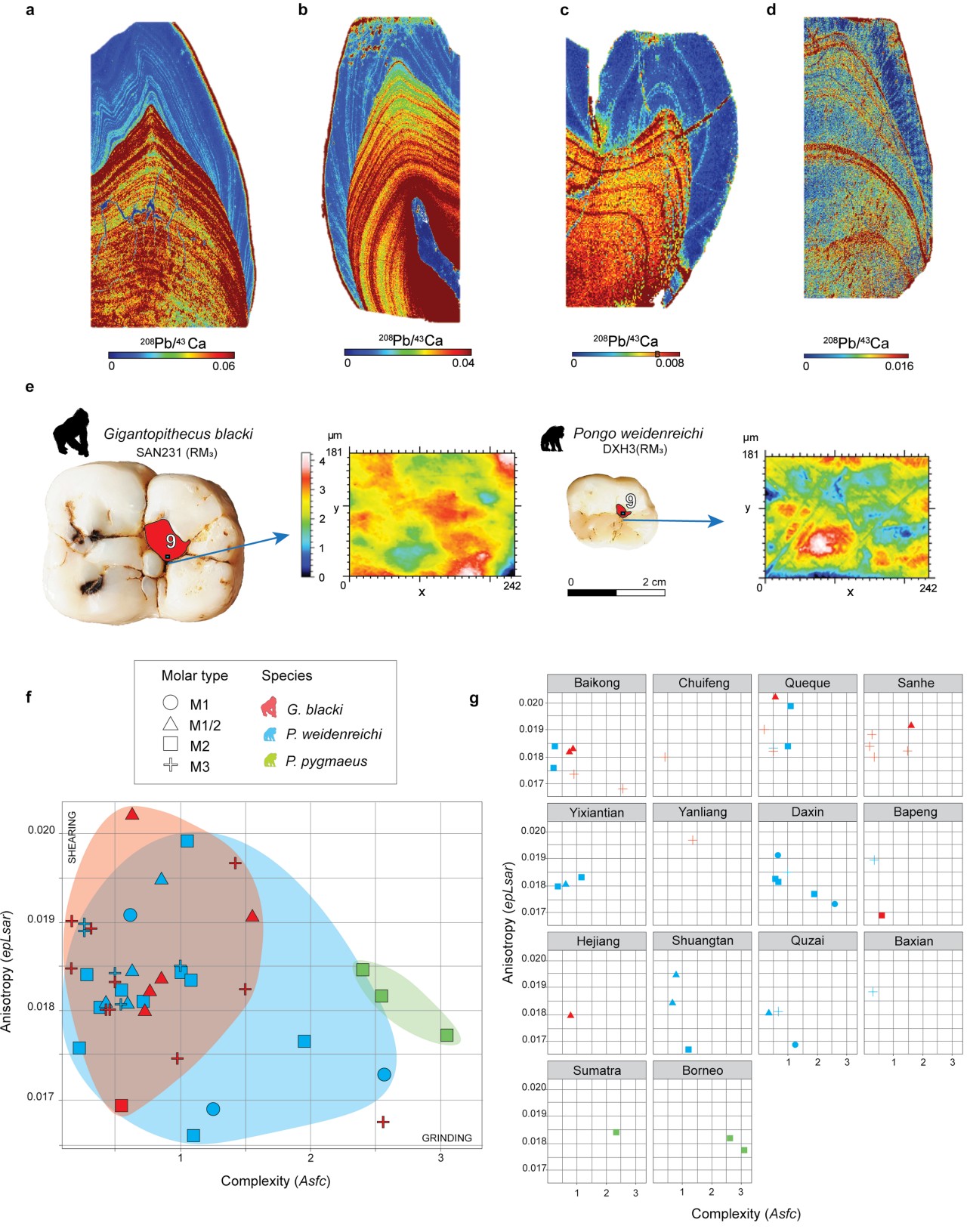

**Extended Data Fig. 10 | Behavioural proxies used in this study. a**, A $^{208}$Pb/$^{43}$Ca elemental map of a left M3 *G. blacki* tooth (CSQ0811-5) from the early site of Queque showing distinct lead banding. **b**, The same scan of a male right P3 (CSQSN-31) also showing the distinct banding. **c**, Teeth from Shuangtan and **d**, Hejiang, showing the banding has lessened and become more blurred. **e**, Lower right M$_3$ of *G. blacki* (left, SAN231) and *P. weidenreichi* (right, DXH3) teeth showing facet 9 (red shade), close the tip crushing point. 2D occlusal maps of the scanned areas are indicated by the blue arrows. **f**, Scatterplots with

DMTA results for complexity (Asfc, x axis) versus anisotropy (epLsar, y axis). Symbols indicate the molar type and colour shades represent the three different species (*G. blacki, P. weidenreichi* and *P. pygmaeus*). All individual molars (n = 41) are plotted to visualize the dietary differences at species level. **g**, Individual molars are plotted per site (n = 14 of which 12 from China), following the geochronology from Early Pleistocene (top left) to Holocene (bottom right). Data visualization was performed in R Studio 1.4.1717.

# Reporting Summary

## Statistics

For all statistical analyses, confirm that the following items are present in the figure legend, table legend, main text, or Methods section.

| n/a | Confirmed | |
|---|---|---|
| ☐ | ☒ | The exact sample size (*n*) for each experimental group/condition, given as a discrete number and unit of measurement |
| ☐ | ☒ | A statement on whether measurements were taken from distinct samples or whether the same sample was measured repeatedly |
| ☐ | ☒ | The statistical test(s) used AND whether they are one- or two-sided<br>*Only common tests should be described solely by name; describe more complex techniques in the Methods section.* |
| ☒ | ☐ | A description of all covariates tested |
| ☐ | ☒ | A description of any assumptions or corrections, such as tests of normality and adjustment for multiple comparisons |
| ☐ | ☒ | A full description of the statistical parameters including central tendency (e.g. means) or other basic estimates (e.g. regression coefficient) AND variation (e.g. standard deviation) or associated estimates of uncertainty (e.g. confidence intervals) |
| ☐ | ☒ | For null hypothesis testing, the test statistic (e.g. *F*, *t*, *r*) with confidence intervals, effect sizes, degrees of freedom and *P* value noted<br>*Give P values as exact values whenever suitable.* |
| ☐ | ☒ | For Bayesian analysis, information on the choice of priors and Markov chain Monte Carlo settings |
| ☒ | ☐ | For hierarchical and complex designs, identification of the appropriate level for tests and full reporting of outcomes |
| ☐ | ☒ | Estimates of effect sizes (e.g. Cohen's *d*, Pearson's *r*), indicating how they were calculated |

*Our web collection on statistics for biologists contains articles on many of the points above.*

## Software and code

Policy information about availability of computer code

| Data collection | All software used for data collection is commercially available or described in the published literature (citations included in supplementary section). Software used includes: MCDoseE 2.0 (for ESR dating), Riso Sequence Pro (4.59) and DRAC v.1.2 (for ESR and OSL dating) |
|---|---|
| Data analysis | All software used for data collection is commercially available or described in the published literature with citations provided in the methods and supplementary information. Software used includes: Isoplot EX 3.75 (U-series), MATHLAB USESR (US-ESR), OxCal (version 4.2) (Bayesian modeling), Riso Analyst (4.57), Australasian Pollen and Spore Atlas (ANU), SensoMAP Premium 8.2.9564, Minitab (19.2020.1) R Studio (1.4.1717), IUCN habitats classification scheme (Ver. 3.1), Metropolis-Hastings algorithm,coupled US-ESR, DATA programs, Microcal OriginPro 8.5 software using a Levenberg-Marquardt algorithm, R package UThwigl, shinyImaging (for elemental mapping). |

For manuscripts utilizing custom algorithms or software that are central to the research but not yet described in published literature, software must be made available to editors and reviewers. We strongly encourage code deposition in a community repository (e.g. GitHub). See the Nature Portfolio guidelines for submitting code & software for further information.

## Data

Policy information about availability of data

All manuscripts must include a data availability statement. This statement should provide the following information, where applicable:
- Accession codes, unique identifiers, or web links for publicly available datasets
- A description of any restrictions on data availability
- For clinical datasets or third party data, please ensure that the statement adheres to our policy

The data that support the findings of this study are included in the Supplementary Information. Additional data, including data sets and raw data have been placed in two Zenobo data repositories and are publicly available; Dating; 10.5281/zenodo.10080908, Environment and behaviour; 10.5281/zenodo.10080973. In addition, Custom code for Ox-cal program have been deposited in Zenodo DOI: 10.5281/zenodo.10077255. There are no restrictions on data availability

## Research involving human participants, their data, or biological material

Policy information about studies with human participants or human data. See also policy information about sex, gender (identity/presentation), and sexual orientation and race, ethnicity and racism.

| | |
|---|---|
| Reporting on sex and gender | N/A |
| Reporting on race, ethnicity, or other socially relevant groupings | N/A |
| Population characteristics | N/A |
| Recruitment | N/A |
| Ethics oversight | N/A |

Note that full information on the approval of the study protocol must also be provided in the manuscript.

# Field-specific reporting

Please select the one below that is the best fit for your research. If you are not sure, read the appropriate sections before making your selection.

☐ Life sciences    ☐ Behavioural & social sciences    ☒ Ecological, evolutionary & environmental sciences

For a reference copy of the document with all sections, see nature.com/documents/nr-reporting-summary-flat.pdf

# Ecological, evolutionary & environmental sciences study design

All studies must disclose on these points even when the disclosure is negative.

| | |
|---|---|
| Study description | To determine the timing and potential cause/s of extinction of Gigantopithecus blacki we incorporated three interrelated components; chronology, past environments and behaviour using six dating techniques and eight proxy lines of evidence |
| Research sample | Primarily we sampled and analysed G. Blacki fossil teeth as evidence of their presence for dating, past environment reconstruction and behavioural analysis. In addition, as we were limited by the number of G. blacki fossils we also identified and analysed faunal teeth and bones from all 22 excavated sites for comparative purposes particularly Pongo as a useful extant species. We also used materials for dating including sediments, cave breccias, speleothem, teeth and bone. |
| Sampling strategy | To establish a statistically significant sample size we selected 22 caves across two known G. blacki regions in southern China; Chongzuo and Bubing Basin. The caves were selected based on the presence of fossil bearing breccia and the relative height on the karst plain. We sampled all 11 known G. blacki caves as this represents the extent of our knowledge of G blacki material in these areas. We sampled widely across these caves and collected material for six different dating techniques resulting in 157 radiometric ages, including the dating and analysis of 22 G. blacki teeth. Sediment sample locations were chosen based on proximity to the fossils. To constrain the fossil assemblage we conducted 52 direct dating age estimates of the bone and teeth from all 22 caves. Samples were chosen according to the weathering state of the enamel/dentine or collagen. Particular emphasis was placed on dense, thick bones without cortical sections. Such bones come closest to conform to the general assumption of the diffusion adsorption model. |
| Data collection | The 22 caves were excavated by the IVPP (Chongzuo) and NHMG (Bubing Basin) from 2000-2019, who recorded all data pertaining to the excavation and catalogued all fossils obtained. All data pertaining to the dating techniques were collected by the appropriate dating specialists during fieldwork between 2017-19 and during laboratory analysis from 2017-2019. |
| Timing and spatial scale | Sampling of the caves for dating was conducted after excavation (see above) but occurred in collaboration with the original excavating team to ensure the association between the fossils and datable material. |

| | |
|---|---|
| Data exclusions | Samples were excluded if they failed tests for each dating method. These tests and exclusion criteria are outlined for each dating method in the Methods Summary and Supplementary Information. Fragmentary fossils that could not be identified were excluded from analysis for the excavations. For the US dating of two teeth samples were excluded from the dating as no enamel layer offered a uranium concentration <5 ppm, which has been described as the maximum acceptable concentration within the enamel layer to obtain a reliable equivalent dose. For the luminescence dating single-aliquots were rejected according to a rejection criteria devised by cited reference. |
| Reproducibility | Multiple samples (157) and dating methods (six) were used to determine a chronology for the G. blacki evidence. For both the sedimentological and fossil context independent ages estimates were employed to guarantee that the results are reliable and reproducible. Age estimates are consistent and stratigraphically correct between samples. |
| Randomization | Randomization was used in the Monte Carlo simulations to estimate the fit of the OSL data in Analyst |
| Blinding | Each dating sample was processed independently with no exchange of data or results until final age estimates had been produced to ensure that the results were generated in isolation. |

Did the study involve field work?   ☒ Yes   ☐ No

## Field work, collection and transport

| | |
|---|---|
| Field conditions | Fieldwork was conducted between July-August and occasionally in December in the Guangxi ZAR region of southern China. The southern hemisphere summer months were hot with occasional rain, while the December trips were cooler and drier. Excavations and sampling were conducted on rugged steep karst terrain, often involving climbing and abseiling activities. |
| Location | The two study areas are located in a radius of ~100-40 km from Nanning – first around Chongzuo city (22°37'N, 107°37'E), southwest of Nanning and the second close to Bubingzhen town in Bubing Basin (23°35'N, 106°59'E) to the north west of Nanning both in Guangxi ZAR in southern China. |
| Access & import/export | Permission for excavation and sampling were obtained by the Institute of Vertebrate Paleontology and Paleoanthropology, Chinese Academy of Sciences, Beijing, via a permit issued by The National Bureau of Cultural Relics in Beijing #3312044 (2010-present) and via a collaborative agreement in conjunction with the Chongzuo Museum (signed 8/10/2018). Sediment samples were exported to Australia using Bio-security permits #IP0000522151 and IP0002282095. G. blacki teeth were imported to Australian by Zhang with permission from IVPP and the Chinese Academy of Sciences permit #[2018]9335 for the purpose of dating and analysis. |
| Disturbance | Mostly this study sampled caves that were already excavated by local institutions so the disturbance to the landscape was minimal. For the two caves excavated during the 2017-19 efforts were made to minimise the impact of the local communities and excavated sediments were backfilled. All excavation procedures and techniques followed national and local regulations. |

# Reporting for specific materials, systems and methods

We require information from authors about some types of materials, experimental systems and methods used in many studies. Here, indicate whether each material, system or method listed is relevant to your study. If you are not sure if a list item applies to your research, read the appropriate section before selecting a response.

## Materials & experimental systems

| n/a | Involved in the study |
|---|---|
| ☒ | ☐ Antibodies |
| ☒ | ☐ Eukaryotic cell lines |
| ☐ | ☒ Palaeontology and archaeology |
| ☒ | ☐ Animals and other organisms |
| ☒ | ☐ Clinical data |
| ☒ | ☐ Dual use research of concern |
| ☐ | ☒ Plants |

## Methods

| n/a | Involved in the study |
|---|---|
| ☒ | ☐ ChIP-seq |
| ☒ | ☐ Flow cytometry |
| ☒ | ☐ MRI-based neuroimaging |

## Palaeontology and Archaeology

| | |
|---|---|
| Specimen provenance | Fossils were recovered from excavations at 22 caves in the Guangxi ZAR region of southern China. More details about the specimen numbers and provenance of these fossils is included in the Supplementary Information. A full list of permits for the excavations can be found in the "Access and import/export" section above. |
| Specimen deposition | Fossils recovered from the 2000-2019 excavations at Chongzuo are housed at the Institute of Vertebrate Paleontology and Paleoanthropology, Chinese Academy of Sciences, Beijing, while those collected in Bubing Basin (1999-2016) are housed at Natural History Museum of Guangxi, Nanning, China. Access to these fossil can be sought via these institutions. |

| Dating methods | The Methods Summary and Supplementary Information provides detailed descriptions of each dating method. Uranium-series dating of the speleothem samples was conducted in the Radiogenic Isotope Facility of The University of Queensland and the Wollongong Isotope Geochronology Laboratory, at University of Wollongong, Wollongong using VG Sector 54 thermal ionisation mass spectrometer (TIMS) and a Nu Plasma multi-collector inductively coupled mass spectrometer (MC-ICP-MS). All luminescence analysis, OSL and pIR-IRSL, was conducted at the "Traps" luminescence dating facility at Macquarie University in Sydney, Australia using a TL-DA-20 Luminescence reader. Laser ablation mass spectrometry to measure U-series isotopes along the teeth were conducted at Southern Cross University, Australia. Additional U-series measurements and ESR measurements were undertaken at GARG Southern Cross University. ESR dating of quartz was undertaken at National Research Centre on Human Evolution CENIEH, Burgos, Spain. |
|---|---|

☒ Tick this box to confirm that the raw and calibrated dates are available in the paper or in Supplementary Information.

| Ethics oversight | No ethical approval was required for this study as the specimens have been fossilized for many thousands of years |
|---|---|

Note that full information on the approval of the study protocol must also be provided in the manuscript.

# Dual use research of concern

Policy information about dual use research of concern

## Hazards

Could the accidental, deliberate or reckless misuse of agents or technologies generated in the work, or the application of information presented in the manuscript, pose a threat to:

No | Yes
☒ | ☐ Public health
☒ | ☐ National security
☒ | ☐ Crops and/or livestock
☒ | ☐ Ecosystems
☒ | ☐ Any other significant area

## Experiments of concern

Does the work involve any of these experiments of concern:

No | Yes
☒ | ☐ Demonstrate how to render a vaccine ineffective
☒ | ☐ Confer resistance to therapeutically useful antibiotics or antiviral agents
☒ | ☐ Enhance the virulence of a pathogen or render a nonpathogen virulent
☒ | ☐ Increase transmissibility of a pathogen
☒ | ☐ Alter the host range of a pathogen
☒ | ☐ Enable evasion of diagnostic/detection modalities
☒ | ☐ Enable the weaponization of a biological agent or toxin
☒ | ☐ Any other potentially harmful combination of experiments and agents

# Plants

| Seed stocks | Fossilized pollen samples were collected from the cave sediments directly at the ANU palaeoecology laboratory |
|---|---|
| Novel plant genotypes | N/A |
| Authentication | N/A |

