## [Peer Review File · Nature]

Manuscript Title: The demise of the greatest ape; Gigantopithecus blacki

Reviewer Comments & Author Rebuttals

Reviewer Reports on the Initial Version:

Referees' comments:

Referee #1 (Remarks to the Author):

My expertise is in luminescence dating. I do not have particular expertise in any of the other topics so I will confine my detailed comments to the luminescence dating. I do have some general comments on the presentation, however.

This paper provides dating and dietary reconstruction to determine the time of extinction of a giant ape and the cause of its extinction. When I first read the main text, I thought this was a pretty good paper, but when I delved into the details contained in the supplementary material, I had second thoughts. The main text is mainly a summary of results, and repetitious to some degree. To find out what the authors really did, the reader has to look at the supplementary. I do not think the role of the supplementary material is to provide the meat of the argument.

This is particularly true for the discussion on chronology. The main text says that six chronological methods are employed but it does not list what those are. Five are listed under methods, although some are just variations of the same method (U-series dating is involved in three of them, for example). The supplementary information says that luminescence dating is the backbone of the dating approach. Although I appreciate that these were difficult samples for luminescence dating, the lack of sensitivity of the feldspar grains is a major drawback, making the luminescence dating somewhat problematic. This is particularly so given the evidence of partial bleaching. No one reading just the main text would get this impression.

I think the main text should be rewritten to include some of the difficulties in the methods and temper the conclusions accordingly. The supplementary material should just include the full data, and not critical discussions that belong in the main text. Some other comments on the general presentation:

1. In general, I found the paper hard to read. Figures are located in three different places: main text, supplementary and something called extended data figures. References are in two different places. The references for the supplementary material are split between the main text and the supplementary.
2. Figure 2a. I had a hard time understanding what I was looking at in the insets. I assume the y-axis refers to different dating techniques, but I think the symbols should be explained in the caption rather than having the reader search through extensive supplementary material to understand what they are. I see from the supplementary that F1 refers to flowstone but this should be mentioned in the caption. In short, I could not understand the figure until I looked at the supplementary.
3. Figure 2c and 3. I do not see the trend in complexity that different between G and P. The anisotropy for G seems highly variable across time.
4. Figure 2d. This part of the figure would be more convincing if examples of P were also given.

Comments on luminescence section of Supplementary

1. In general, I found the luminescence section somewhat sloppy. I had a hard time figuring out what actually was done. Many of my subsequent comments are examples of this sloppiness.
2. Lines 13-14. Incomplete sentence.
3. Lines 16. What is meant by a “low number of luminescence decays”? Do the authors mean a low number of grains with a measurable luminescence signal?
4. Line 36. I assume the same curve fitting and error determination as used in quartz were applied here.
5. Line 62. Determining the gamma dose rates and the beta dose rates (to compare with beta counting) from alpha counting assumes secular equilibrium in the decay chains, does it not? You demonstrate disequilibrium later on, bringing into doubt the beta counting and alpha counting results? Also I do not think you mean gamma dose rate when you talk about conversion factors from Guerin et al. because beta dose rates are also involved. Reword.
6. Lines 75-78. Confusing the way this is written. I also wonder about the 12.5 +/- 0.5 % estimate for internal K in feldspars. Smedley et al (2012) (Radiat Meas) suggested using 10 +/- 2 %, and given that some of your samples are fairly insensitive, the internal K might even be less.
7. Lines 87-89. Awkward wording. Multi-grain aliquots are used to test suitability for single-grain analysis, but then single-grains are used anyway because of low sensitivity? I would think the procedure would be reversed. Test single-grains first and then use multi-grain aliquots if acceptance rates for single-grains are low.
8. Line 104. I am somewhat confused. I thought the multi-grain aliquots were to be used for these procedural tests, but here single-grains are used.
9. Lines 105-110. I am not sure I follow. Different grains on the same disk were used for different preheat/stimulation combinations?
10. Line 112 – Is this the Huntley-Lamothe procedure?
11. Table S9. I am not sure what preheat range means. Are you talking about the range of equivalent dose values obtained on single grains using that particular preheat? I find the value of 3 for the 50,270 combination rather unbelievable.
12. Line 128. Is the lower fading rate from the multi-grain aliquots due to averaging?
13. Line 130. You do not say what the lowest rate is.
14. Lines 137-149. I found this paragraph incomprehensible. You start off talking about provenience, then switch to discussing growth curves (although nothing about the 50,270 combination you are using) and saturation, and then end with discussion of fading correction. The last sentence about using the Lamothe technique even though it is designed for the linear section of the growth curve did not make any sense.
15. Lines 152-157. I am confused about what you did here. You measured what I take to be the over-dispersion typical of a single-aged sample by conducting a bleaching test on a modern sample. Then I am not sure what you did. If you measured the equivalent dose of these grains and then applied the MAM using 10-15% over-dispersion, then you should have ended up with a single-aged sample. The argument seems circular.
16. Line 179. What is a plateau of 260°C? Do you mean the widest plateau was at 260°C.
17. Lines 187-190. Do you mean the sensitivity of the quartz was better than feldspar? Under-estimation of age for quartz near saturation is a common phenomenon.
18. Lines 13-14 under discussion. If you applied in your MAM analysis 10-15% over-dispersion, typical for single-aged samples, I am surprised that any of your samples that say were in the 20-25%

over-dispersion range produced any difference between MAM and the central age model. If there was a difference, then application of the MAM would have been as justified for these as those with the 60% over-dispersion. This assumes partial bleaching is the cause of the over-dispersion and not mixing (you might comment on this).

19. Tables S4-S6. I appreciate the difficulty you had in getting a sufficient number of sensitive grains, but I notice in some of the radial graphs, the minimum age is based on only 1 or 2 grains. You might comment on the reliability of those ages. I think the graphs would be improved, also, if you added the 2-sigma error shading around the minimum age line.

20. Line 73. Do you mean by the fossil age that obtained by ESR or U-series?

21. Lines 73-89. The higher age for the single (multi-grain) aliquots is mainly due to averaging effects.

22. Lines 123-124. I do not understand how higher dose rates correlate with less disequilibrium. Do you mean disequilibrium has less effect with higher dose rates? This would particularly be the case if the higher dose rate was because of K.

23. Lines 75-76 under modelling discussion. I cannot understand the justification for using the multi-grain aliquot IRSL results in the modeling, since they cannot be particularly accurate given the problem of averaging.

Referee #2 (Remarks to the Author):

The huge charismatic extinct ape *Gigantopithecus blacki* (estimated standing height around 3 meters), one of the largest of the megafauna from southeast Asia and by far the largest known primate, was first described in 1935, on the basis of 'dragon bones' (fossil teeth) from Chinese apothecaries by G.H.R. von Koenigswald. It is known from numerous caves in southern China (also tentatively elsewhere in southeast Asia) from thousands of teeth, just four mandibles, but no postcranial material. Other great apes - orangutans *Pongo* species - survive to the present day elsewhere in southern Asia.

G. blacki has previously not been subject to a major study. The extensive and innovative research, presented by Westaway et al., based on 22 caves in two regions of southern China includes a detailed chronology, reconstruction of changing climates and vegetation, and inferred behaviour of *G. blacki*. On the basis of 157 radiometric ages from six dating techniques (including OSL and other luminescence data, U-series – on tooth enamel, electron spin resonance) they record its occurrence in southern China from about 2 million years ago to the late middle Pleistocene between about 295 and 215 thousand years ago (extinction window: EW) when it is inferred to have gone extinct. Notably, it was one of the only primates to go extinct in the last 2.6 million years whereas other great apes (e.g., gorilla, chimpanzee, orangutan) survive until the present day.

The submitted paper constitutes an exceptionally detailed and thorough study of what has hitherto been a relatively poorly known species by a team of specialists in a wide range of disciplines. The main text is lavishly augmented by extensive supplementary information which includes detailed descriptions of the caves, sediments and stratigraphy, dating methods and illustrations of faunal material (both *G. blacki* and associated taxa).

Based on pollen and charcoal analyses the authors show that from 2.3 Ma the environment in

southern China was a mosaic of forests and grasses, which supported both *G. blacki* and its taxonomic relative *Pongo weidenreichi* (an ancestral orangutan). *G. blacki*'s disappearance can be linked to the marked changes in climate and vegetation cover just prior to about 295-215 thousand years ago (EW). This period experienced enhanced environmental variability due to an increase in seasonality, which resulted in a decline in tree cover and increase in open environments. The paper convincingly demonstrates that the EW corresponds with the disappearance of *G. blacki*, indicating that environmental changes caused "dwindling populations and ultimately this led to its demise," whereas the related *Pongo. weidenreichi* (ancestral orangutan) was able to adapt its dietary preferences and behaviour to this variability. Older cave sites have *G. blacki* and *P. weidenreichi*, whereas the cave faunas post 415 ka have only *P. weidenreichi*. Later sites also have associated charcoal suggesting increased frequency of fire.

The main findings are nicely summarized in figure 3h.

Added observations: it is notable that *G. blacki* went extinct in the late middle Pleistocene, substantially earlier than the well-known late Pleistocene event in North America and elsewhere. Along with SubSaharan Africa, southern Asia had very few late Quaternary megafaunal extinctions, in marked contrast to other regions such as North and South America. It is notable that unlike the late Quaternary extinctions in other parts of the world, this megafaunal extinction occurred in the absence of *H. sapiens*, but is clearly attributable to climatic/vegetational change. *G. blacki* extinction occurred well before the appearance of *Homo sapiens*, regionally, but it links convincingly to strong evidence of climatic/vegetational changes.

Referee #3 (Remarks to the Author):

The article of Westaway and co-authors is an important contribution for the understanding of the demise in the late Middle Pleistocene of *Gigantopithecus blacki*, the largest great ape of the southeast Asian past megafauna. The authors used 6 dating techniques and multiple proxy indicators of ecology (diet and behavior) to identify the timing and causes of its demise. The bulk of the work – radiometric dating of 22 sites in southern China – allows to determine more precisely this event, outlining three periods, pre-Extinction Window, EW, and post-EW.

Lines 70-92: First, I would suggest to the authors to add some information in the main Text to better describe the ecosystems at the time of *Gigantopithecus*. In the Introduction, it is essential to place *Gigantopithecus* as an element of a diverse primate communities. The diversity of primates was much higher in the Early to Middle Pleistocene than during the Late Pleistocene with genera now extinct (*Gigantopithecus*, *Procynocephalus*, "Mystery Ape") along with modern genera (*Pongo*, *Macaca*, *Trachypithecus*, *Rhinopithecus*, *Nomascus*). That suggests that the forests had a great capacity to support this biomass of primates, along with other large ground-dwelling mammals. From when do we have the co-occurrence of *Pongo weidenreichi* and *Gigantopithecus blacki* in the studied fossil records? From Baikong?

I have also some comments on the interpretation given by the authors lines 113-116 and 150-151. The authors describe a decrease in arboreal cover combined with an increase in ferns and a large increase in grassland just prior and during the Extinction window, based on their pollen analysis presented in Figures 2b and 3c. However, the results seem not to support these major changes in

the vegetation prior and during EW. In both Figures, we can note that the arboreal cover remained predominant and extended into the post-Extinction Window period. Furthermore, this is in accordance with the stable isotope data of *P. weidenreichi* and *G. blacki* which both relied on canopy forests through the period. What the data suggest is a major shift of forest cover along with signs of increase of charcoal in the landscape in the post-EW period, around 200 ka, after the extinction of *G. blacki*. Therefore, could the instable period from ~700 to ~250 ka be due to changes in forest plant communities and forest structures with more open forests (rather than a decrease in arboreal cover), along with other climatic changes such as increase in seasonality, precipitation, overall lower temperatures, etc. Could the authors be clearer on this aspect?

Nevertheless, the data of the other analyses (stable isotope, dental microwear texture, trace element...) are rather convincing and undeniably show that this transitional period had a greater impact on the ecology of *Gigantopithecus* than on that of *Pongo*, most likely due to differences in their life histories (weight, locomotion, lifespan, reproduction rate, etc) and flexibility to dietary changes.

Figure 3c: I have also a comment on the use of the number of teeth to estimate the population size of *Pongo* and *Gigantopithecus* by site. The number of specimens of a given taxon depends on the size of assemblages (ex: 2592 isolated teeth at Baikong vs 11796 teeth at Baxian) and on the depositional and taphonomic conditions within each site (which are not described in SI). That means that the number of specimens (here teeth) is not a good parameter to estimate the size of a population. Moreover, one individual can be represented by one tooth (in case of highly fragmented remains) or by 32 teeth (in case of the deposition of complete jaws in the site). That is why the minimum number of individuals (MNI) is a better parameter to estimate the size of a population.

Author Rebuttals to Initial Comments:

Responses to referees comments 2022-12-19733A

We thank the reviewers for their detailed comments, which we found to be insightful and helpful and have greatly improved the quality of the manuscript. These are our detailed responses;

The main text is mainly a summary of results, and repetitious to some degree. To find out what the authors really did, the reader has to look at the supplementary. I do not think the role of the supplementary material is to provide the meat of the argument.

The word limit restrictions of the main paper combined with the amount of data to discuss mean that we have no choice but to put the detailed discussions of each technique in the supplementary. The main text provides a summary and further details can be sought within the supplementary as is typical for a Nature structure. However, we have included more details to the main text on the techniques applied – especially the dating techniques.

The main text says that six chronological methods are employed but it does not list what those are

We have added further details to the main text such as listing all the dating techniques employed and provided the locations of where to find the discussion of the limitations and uncertainties associated with the dating. L100 “(Fig. 2a, 3a, ED3-6 and Supp. 4-8 for all dating tables and discussion of limitations and uncertainties)”

the lack of sensitivity of the feldspar grains is a major drawback, making the luminescence dating somewhat problematic.

The lack of sensitivity in a sample could be a problem but closer analysis of the data and XRF analysis of the single-aliquot discs reveals that the majority of signal comes from one or two feldspar grains. This means that although most of the feldspars don't luminesce the ones that do are quite bright and have a reasonable K content. We also know this from the samples that were run for single-grain. Some samples were only run using single-aliquots because the number of grains that luminesce were small making it inefficient to run a large number of single grain discs before we ran out of grains. However, when both the SG and SA are compared with the independent age estimates it can be seen that the comparison is favourable and both results do not represent large overestimates of the burial age – a graph to demonstrate this comparison has been added to the luminescence supplementary section (Fig. S6)

I think the main text should be rewritten to include some of the difficulties in the methods and temper the conclusions accordingly.

A brief reference to these uncertainties has been added to the main text with the location of a more detailed discussion in the supplementary sections. Due to the space requirements of Nature and the large number of techniques applied to this evidence means it is not possible to

include all the difficulties and uncertainties in the main text – this is the role of the supplementary. However, they can certainly be eluded to in the main text with a reference to the supplementary for further details.

The supplementary material should just include the full data, and not critical discussions that belong in the main text.

Unfortunately, there just isn't the space in the main text to include all of these critical discussions but they will be referred to so the reader can follow up with the details in the supplementary should they require further explanation or justification.

Figures are located in three different places: main text, supplementary and something called extended data figures. References are in two different places. The references for the supplementary material are split between the main text and the supplementary. This is the structure determined by Nature

Figure 2a. I had a hard time understanding what I was looking at in the insets. I assume the y-axis refers to different dating techniques, but I think the symbols should be explained in the caption rather than having the reader search through extensive supplementary material to understand what they are. I see from the supplementary that F1 refers to flowstone but this should be mentioned in the caption. In short, I could not understand the figure until I looked at the supplementary.

y-axis labels have been added to the insets in this figure, symbols and abbreviations have been added to the caption

3. Figure 2c and 3. I do not see the trend in complexity that different between G and P. The anisotropy for G seems highly variable across time.

We appreciate this observation, however, we kindly wish to point out that this has already been explained (in slightly different wording) in lines 142-147 of the main text. It is correct that the complexity between G and P is not that different, because there are no statistically significant dietary differences between G and P-bearing sites. However, there are significant dietary differences between four G-bearing sites (Baikong, Bapeng, Yanliang and Queque) and G tends to show slightly higher fluctuations in trend lines, especially for anisotropy (while those of P seem more stable, in particular anisotropy over and beyond the EW).

4. Figure 2d. This part of the figure would be more convincing if examples of P were also given.

We agree. The trace element scans for *P. weidenreichi* have been added for both Queque (pre-EW) and Shuangtan (EW) caves to enable a direct comparison with the *G. blacki* population. As we have limited space in this figure we have only added the strontium scan for *P. weidenreichi* (although the barium scan tells the same story). This direct comparison clearly shows that although the *P. weidenreichi* population shows some changes since Queque their

scan displays much less stress when compared to the *G. blacki* scans from the same cave. We thank the reviewer for this helpful suggestion. We have added to the text L183 “This is the first insight into *G. blacki*’s behaviour as a species on the brink of extinction in comparison to *P. weidenreichi* (Fig. 2d vi) which in stark contrast displays much less stress at this time”.

2. Lines 13-14. Incomplete sentence.

Typo corrected

3. Lines 16. What is meant by a “low number of luminescence decays”? Do the authors mean a low number of grains with a measurable luminescence signal?

Yes we have changed it to a low number of grains with a measurable luminescence signal

4. Line 36. I assume the same curve fitting and error determination as used in quartz were applied here.

Curve fitting and error determination details have been added

5. Line 62. Determining the gamma dose rates and the beta dose rates (to compare with beta counting) from alpha counting assumes secular equilibrium in the decay chains, does it not? You demonstrate disequilibrium later on, bringing into doubt the beta counting and alpha counting results?

We undertook three methods of dose rate determination to test the reliability of each method 1) Alpha and beta counting, 2) in-situ gamma and beta counting and 3) high resolution gamma spectrometry to ensure that the estimated dose rates were as accurate as possible and that we were not relying on just one method. We did detect a small amount of disequilibrium in certain samples and this was accounted for in the error margin as explained in the text. The three different methods were coeval within error margins suggesting that we are close to the true environmental dose rate – especially when we accounted for the potential impacts of disequilibrium.

Also I do not think you mean gamma dose rate when you talk about conversion factors from Guerin et al. because beta dose rates are also involved. Reword.

Changed text to converted to gamma and beta dose rates

6. Lines 75-78. Confusing the way this is written.

This sentence has been simplified

I also wonder about the 12.5 +/- 0.5 % estimate for internal K in feldspars. Smedley et al (2012) (Radiat Meas) suggested using 10 +/- 2 %, and given that some of your samples are fairly insensitive, the internal K might even be less.

The feldspar grain that did luminescence, although few in number were quite bright – indicating a high internal K and as these dominated the signal we thought that the 12.5 +/- 0.5 % was a better estimate. To confirm this, we tested the K content using the XRF and it returned values of between 10-12 %.

7. Lines 87-89. Awkward wording. Multi-grain aliquots are used to test suitability for single-grain analysis, but then single-grains are used anyway because of low sensitivity? I would think the procedure would be reversed. Test single-grains first and then use multi-grain aliquots if acceptance rates for single-grains are low.

This section has been changed to L89 “Due to low yields in potassium feldspar grains at these sites, we decided to only apply single-grain pIR–IRSL techniques to a select number of suitable samples” As the procedural test were conducted mostly on single-aliquots it made sense to keep use SA for the SAR runs – any samples that produced decays for all SA and that had enough feldspars were also run for SG. We determined that using SA was a more efficient way of identifying the grains that luminesce than running a large number of SG discs.

8. Line 104. I am somewhat confused. I thought the multi-grain aliquots were to be used for these procedural tests, but here single-grains are used.

The text states that initial feldspar procedural tests used single-aliquots to conduct the preheat plateau, fading, bleaching and dose recovery tests. It also states that we also modified these tests for single-grains, but only for certain samples. We have now added L96 “...single-grains of feldspar for certain samples that yielded a slightly larger amount of the 180-212 μm size fraction”

9. Lines 105-110. I am not sure I follow. Different grains on the same disk were used for different preheat/stimulation combinations?

As stated in the text four single-grain discs were used for the single-grain tests allowing one disc for each combination – the text has been updated to make this clearer L102 “we used four single grains discs with a total of 24 accepted grains and recycled these discs for all the tests, with one disc per preheat/stimulation combination”.

10. Line 112 – Is this the Huntley-Lamothe procedure?

The anomalous fading tests for single-grains follows the procedures of Rhodes et al. (2015) as stated, but are based on the Huntley-Lamothe procedure but with some slight modifications to make it applicable to single-grains.

11. Table S9. I am not sure what preheat range means. Are you talking about the range of equivalent dose values obtained on single grains using that particular preheat? I find the value of 3 for the 50,270 combination rather unbelievable.

This is a typo and has been corrected to 23 Gy

12. Line 128. Is the lower fading rate from the multi-grain aliquots due to averaging?

The lower fading rate seen for single-aliquots could be attributed to the averaging effect but considering that the single-aliquots have very few grains that luminescence then the fading effects are averaged over only a few grains – in comparison the SG discs actually have more grains that luminescence (as the discs were hand selected according to this criteria) so this effectively improves the average fading value.

13. Line 130. You do not say what the lowest rate is.

The lowest rates have been added to the sentence so it now reads "The fading results from the single aliquot tests (2-6%) were lower than from the single-grain tests (3-7%), which revealed a range of fading from the highest IR₅₀ (~12 %) to the lowest pIR-IRSL_{50,270} (2-3%), pIR-IRSL_{50,290} (3-4%)_ and pIR-IRSL_{200,290} signals (4-6%).

14. Lines 137-149. I found this paragraph incomprehensible. You start off talking about provenience, then switch to discussing growth curves (although nothing about the 50,270 combination you are using) and saturation, and then end with discussion of fading correction. The last sentence about using the Lamothe technique even though it is designed for the linear section of the growth curve did not make any sense.

This paragraph is explaining our procedures to determine the field saturation of the feldspar grains to better understand the inherent fading in the samples. We have now added the following sentences to make it clearer "In an effort to understand the extent of fading in these samples we wanted to investigate the field saturation of the feldspar grains. To do this we had to find an outcrop of rock that contained feldspars. As the dominant geology in this region is Triassic and Carboniferous cherty limestone, the feldspars in these samples must have been transported via fluvial process from the late Palaeozoic granite lithology found far to the south of these caves. Thus, to represent the local granite geology from the region a small weathered granitic clast found in the cave breccia was used to estimate the field saturation as the closest representation of the provenance of the feldspar grains".

'15. Lines 152-157. I am confused about what you did here. You measured what I take to be the over-dispersion typical of a single-aged sample by conducting a bleaching test on a modern sample. Then I am not sure what you did. If you measured the equivalent dose of these grains and then applied the MAM using 10-15% over-dispersion, then you should have ended up with a single-aged sample. The argument seems circular.

We estimated the intrinsic sources of uncertainty, including instrument reproducibility, by conducting dose recovery tests on a modern sample. This was estimated to be between 7-15%. Prior to the application of the MAM model to the accepted equivalent doses this additional uncertainty of 7% for single aliquots and 12% for single grains was added to the uncertainty of all individual dose estimates (calculated from counting statistics and curve fitting errors). This additional uncertainty was added as a percentage of the individual dose estimates to account for the variability in measured dose distributions arising from additional intrinsic factors. We consider this as the minimum uncertainties with which the dose in these samples can be measured using a single aliquot or single grains. This is following the suggestions of Galbraith et al. (1999) and Medialdea et al. (2014). These details have been added to the text, including the Galbraith reference.

16. Line 179. What is a plateau of 260°C? Do you mean the widest plateau was at 260°C.

This has been changed to ‘revealing the widest plateau at 260 °C

17. Lines 187-190. Do you mean the sensitivity of the quartz was better than feldspar? Under-estimation of age for quartz near saturation is a common phenomenon.

No, the sensitivity of quartz was also low as stated, but as the sample yielded a greater proportion of quartz grains we had the opportunity to run more discs and therefore improve the number of accepted grains compared to feldspars.

18. Lines 13-14 under discussion. If you applied in your MAM analysis 10-15% over-dispersion, typical for single-aged samples, I am surprised that any of your samples that say were in the 20-25% over-dispersion range produced any difference between MAM and the central age model. If there was a difference, then application of the MAM would have been as justified for these as those with the 60% over-dispersion. This assumes partial bleaching is the cause of the over-dispersion and not mixing (you might comment on this).

We have attributed the over dispersion in these samples to partial bleaching as stated at the beginning of this paragraph and in line 82 (“This could indicate a significant averaging effect from the number of grains on the disc, however as the signal is dominated by only 1-2 grains

a more likely explanation is a partial bleaching effect, which is also to be expected in cave environments”) but for clarity we have also added this at the end of this paragraph.

19. Tables S4-S6. I appreciate the difficulty you had in getting a sufficient number of sensitive grains, but I notice in some of the radial graphs, the minimum age is based on only 1 or 2 grains. You might comment on the reliability of those ages.

A discussion of the reliability of some of the age estimates was added to the discussion section. For the single aliquot results we added L20 “We were able to analyse most of the samples using the MAM but some of the result relied on only 1-2 aliquots for D_e determination, such as CSHT1,11 CHEJ2,3 CYAN13 and CGONG16. However, statistically significant data was obtained from pIR-IRSL single-grain analysis for the CSHT, CHEJ and CGONG samples, and OSL single-grain analysis for the younger CBAX samples. For some samples such as CYIX and CYAN we were unable to proceed with a single-grain investigation and thus these single-aliquot age estimates are viewed as maximum ages. For the single-grain results we added L32 “For some samples, such as HEJ1,2 and CMF3 the MAM is based on only 1-2 grains, but when used in conjunction with the coeval single-aliquot results and the favourable comparison with independent age estimates the reliability of these results is improved”.

I think the graphs would be improved, also, if you added the 2-sigma error shading around the minimum age line.

We have added a 2 sigma shading around the MAM line (in blue) for all radial plots in Fig. S4 and 5 – this helps the reader to see which grains are dominating the MAM. We appreciate this useful suggestion.

20. Line 73. Do you mean by the fossil age that obtained by ESR or U-series?

The fossil age obtained by coupled US-ESR, the fossil age obtained by just U-series represents a minimum age and tended to be younger than the luminescence ages.

21. Lines 73-89. The higher age for the single (multi-grain) aliquots is mainly due to averaging effects.

Normally this would be the case in single-aliquots that contain many grains that luminesce, however in these aliquots the signal is dominated by only 1-2 grains that luminesce so the averaging effect would be minimal.

22. Lines 123-124. I do not understand how higher dose rates correlate with less

disequilibrium. Do you mean disequilibrium has less effect with higher dose rates? This would particularly be the case if the higher dose rate was because of K.

This comment about dose rates in the caves was part of a wider discussion of disequilibrium. The paragraph above had discussed that L115 “Shuangtan cave (CSHT) is the most cemented of all the breccias reducing its capacity for water seepage and leaching and subsequently has the lowest deficiencies in the ^{238}U and ^{232}Th decay chains”. If we assume a similar initial dose rates for the caves in this area, then the caves that contain a higher dose rate such as Shuangtan (2.6-2.7 Gy/ka), could be due to a lower amount of leaching and disequilibrium in the cave environment so the total dose rate is higher than the caves that experienced more leaching. However, we then go on to add that L131 “Shuangtan dose contain the highest values of ^{40}K in all the samples (double that of Hejiang cave), which could also explain the higher dose rate”. So in the case of Shuangtan cave, the higher dose rates relate to the cemented composition of the breccia and the presence of a higher K content.

23. Lines 75-76 under modelling discussion. I cannot understand the justification for using the multi-grain aliquot IRSL results in the modeling, since they cannot be particularly accurate given the problem of averaging.

In the same way that U-series dating of bone and teeth are included as minimum ages in the model we have included the pIR-IRSL single aliquot results, albeit as maximum ages. This is in keeping with the nature of the dating methods and the quality of the results. Therefore, they are not presented as true ages but presented with an appropriate uncertainty and caveat. We have added L79 “However, as each single-aliquot contains only 1-2 grains that luminescence the averaging effect is present, but minimal. Thus, we believe that the inclusion of the single-aliquots results, albeit as maximum ages, is still justifiable. The inclusion of both U-series dating on teeth and bone and the pIR-IRSL single-aliquot age estimates into the model provides supporting data that help to sandwich the true burial age of the breccia”.

Referee #2 (Remarks to the Author):

Added observations: it is notable that *G. blacki* went extinct in the late middle Pleistocene, substantially earlier than the well-known late Pleistocene event in North America and elsewhere. Along with SubSaharan Africa, southern Asia had very few late Quaternary megafaunal extinctions, in marked contrast to other regions such as North and South America.

This is an interesting point that we have incorporated into the main text L226 “When compared to other well-known extinction events in North America and Australia²⁸⁻³⁰, this megafaunal extinction occurred much earlier and in the absence of *H. sapiens*”

It is notable that unlike the late Quaternary extinctions in other parts of the world, this megafaunal extinction occurred in the absence of *H. sapiens*, but is clearly attributable to

climatic/vegetational change. *G. blacki* extinction occurred well before the appearance of *Homo sapiens*, regionally, but it links convincingly to strong evidence of climatic/vegetational changes.

Another valid point that we have incorporated into the main text – see above

Referee #3 (Remarks to the Author):

Lines 70-92: First, I would suggest to the authors to add some information in the main Text to better describe the ecosystems at the time of *Gigantopithecus*.

This is a valid point and has been added to the main text within the limitations of word count to provide a better context for *G. blacki* during the pre-EW period. We have added L88 “The diverse forest ecosystem at the time of Baikong had the capacity to support the biomass of multiple primate communities⁴ over a wide area. But by the time of Hejiang *G. blacki* had a dramatic range reduction” so the readers have a sense of how rich and diverse the ecosystems were at this time. This has also been added to the summary of the pre-EW in L198 “During the pre-EW period, *G. blacki* flourished alongside other primates as a successful specialist (Fig. 3c), enjoying a large diversity of food in a rich evergreen-deciduous forest (Fig. 2di-ii)”

In the Introduction, it is essential to place *Gigantopithecus* as an element of a diverse primate communities. The diversity of primates was much higher in the Early to Middle Pleistocene than during the Late Pleistocene with genera now extinct (*Gigantopithecus*, *Procynocephalus*, “Mystery Ape”) along with modern genera (*Pongo*, *Macaca*, *Trachypithecus*, *Rhinopithecus*, *Nomascus*). That suggests that the forests had a great capacity to support this biomass of primates, along with other large ground-dwelling mammals.

We agree and have incorporated this into the sentence on the ecosystems – see above

From when do we have the co-occurrence of *Pongo weidenreichi* and *Gigantopithecus blacki* in the studied fossil records? From Baikong?

Yes we have the co-occurrence of *P. weidenreichi* and *G. blacki* in Baikong, Chuifung, Sanhe, Queque, Yixiantian, Yanliang, Hejiang and Shuangtan. This data has been added to Fig 3 – where the previous faunal counts has been updated to % counts relative to *P. weidenreichi* so the data is comparable.

I have also some comments on the interpretation given by the authors lines 113-116 and 150-151. The authors describe a decrease in arboreal cover combined with an increase in ferns and a large increase in grassland just prior and during the Extinction window, based on their pollen analysis presented in Figures 2b and 3c. However, the results seem not to support these major changes in the vegetation prior and during EW. What the

data suggest is a major shift of forest cover along with signs of increase of charcoal in the landscape in the post-EW period, around 200 ka, after the extinction of *G. blacki*.

We agree with the reviewers point and have made this distinction much clearer. The real decline in arboreal cover did not start until ~200 ka. We have changed this in the text (see below)

Therefore, could the instable period from ~700 to ~250 ka be due to changes in forest plant communities and forest structures with more open forests (rather than a decrease in arboreal cover), along with other climatic changes such as increase in seasonality, precipitation, overall lower temperatures, etc. Could the authors be clearer on this aspect?

We agree with this helpful suggestion and have modified the interpretation and associated text so that the unstable period (which we have now named the transitional period) signifies a change in plant communities and forest structures leading to more open forest environments L117 “However, just prior to the EW there was a change in forest plant communities and an increase in forest disturbance taxa with more open forests dominating. Post EW ~200 ka, there was a large decrease in arboreal cover”. To further support this argument, we have added in an additional taxon into the arboreal cover in Fig 3b – the darker shading denotes the percentage taxa that represent forest disturbance, such as *Celtis*, *Trema* and *Sapindaceae*. This disturbance taxa can be seen to increase during the transitional phase, EW and continues to increase into the post-EW period.

We have also added L156 “The pollen and faunal data indicate that the early mosaic landscapes were interrupted by enhanced environmental variability (Fig. 3b) prior to the EW in the transitional phase; as suggested by the change in forest communities and structures, and post EW; as suggested by a decline in arboreal cover and an increase in ferns and grasslands associated with fire”. This change in pollen interpretation has helped us to identify the novel outcomes of this research and to develop our argument for the drivers of extinction. L168 we have added “The decline in forest cover during this period is documented in China²¹, Southeast Asian²² and Australasia²³. However, our pollen study demonstrates that the key to *G. blacki* extinction is not the deterioration in arboreal cover but rather the influence of environmental variability in changing the composition of forest communities, particularly the increase in disturbance taxa”.

this transitional period had a greater impact on the ecology of *Gigantopithecus* than on that of *Pongo*, most likely due to differences in their life histories (weight, locomotion, lifespan, reproduction rate, etc) and flexibility to dietary changes.

We agree and have added this to the text L208 “Despite sharing similar environments with *P. weidenreichi* pre-EW, from 600-300 ka there is evidence of *G. blacki*'s inability to adapt to

this transitional period, which had a greater impact on its resilience to the changing ecology”

Figure 3c: I have also a comment on the use of the number of teeth to estimate the population size of *Pongo* and *Gigantopithecus* by site. The number of specimens of a given taxon depends on the size of assemblages (ex: 2592 isolated teeth at Baikong vs 11796 teeth at Baxian) and on the depositional and taphonomic conditions within each site (which are not described in SI).

The sedimentological and depositional details of selected sites have been provided in the microstratigraphy section (SI section 11)

That means that the number of specimens (here teeth) is not a good parameter to estimate the size of a population. Moreover, one individual can be represented by one tooth (in case of highly fragmented remains) or by 32 teeth (in case of the deposition of complete jaws in the site). That is why the minimum number of individuals (MNI) is a better parameter to estimate the size of a population.

We agree with this comment, the MNI (minimal number of individuals) is usually used in faunal analyses as an indicator of number of individuals and is a better indicator of population size than raw faunal counts. However, the authors think MNI is arbitrary in this research because it will put more weight on taxa with fewer representative specimens and therefore only takes into consideration partial data sets rather than the full data set. To avoid this, in Fig 3c – we have removed the raw faunal counts and replaced it with a new indicator to represent the relative population size of *G. blacki* and *P. weidenreichi* in a faunal assemblage. We have used the percentage of *G. blacki* teeth relative to *P. weidenreichi* teeth as a ratio of teeth in a fauna. The rationale behind this indicator is that, theoretically, each tooth of an animal has equal chance to be fossilized, and the animal with more teeth will have more chance to be fossilized as isolated dental remains. As *G. blacki* and *P. weidenreichi* have the same dental formula, they have the same number of teeth in their mouth, so a relative measure will not bias either species. These relative percentages can then be used as a rough proxy for the relative population size of *G. blacki* in comparison to *P. weidenreichi*. The *G. blacki* percentages are then representative of the size of fossil assemblage in each cave because they are scaled to the number of *P. weidenreichi* finds.

In the faunal analysis, we wanted to avoid putting more weight on taxa with fewer representative specimens, so we introduce a new indicator to represent the relative population size of a certain taxon in a fauna, i.e. the ratio of Individual Number Index for a Genus (INIG) and Individual Number Index for a Fauna (INIF). The former is calculated from the specimen number of a genus divided by the number of teeth of the genus, and the latter is calculated from the sum of INIGs of all the genera in a fauna. The rationale behind this is that, theoretically, each tooth of an animal has equal chance to be fossilized, and the animal with more teeth will have more chance to be fossilized as isolated dental remains. Of course, this indicator only works for isolated dental remains, which is the right scenario for this research. This indicator

is also used to finally calculate the Habitat Predominance Index for a Fauna (HPIF) in this research.

Reviewer Reports on the First Revision:

Referees' comments:

Referee #1 (Remarks to the Author):

I was mainly satisfied with the author responses to my previous comments. I have just a few additional minor comments.

Main text

Lines 131-135 – You don't explain the significance of the differences in stable isotopes. You do in the supplementary, but you need to have a sentence here.

Supplement

Lines 40-41 – I do not understand why the disc locating procedure was done before any heating. All grains would get the same heating regardless. Can you explain.

line 78 – In response to reviewer comment about using the 12% value for internal K, you mentioned some measurements on some grains. Could you include that procedure here?

line 123 – I agree the 270°C stimulation provided the best results from the tests, but it did not have the lowest residual value.

line 151 – Could you explain how you determined the field saturation. I understand that you showed the old sample was in saturation, but could you explain how the value of saturation was obtained – so that it could be compared with the Do values in the table. I was not clear.

lines 161-163 – It probably does not make difference with the low fading rates you used, but shouldn't the fading correction be done on the age and not the equivalent dose? That is what Huntley-Lamothe did.

line 170 – The table says 5% but here you say 10%.

lines 9-17 under discussion – Just because some samples might be partially bleached does not mean all of them are. I do not see the logic in using an argument of consistency to apply MAM to all. In fact, you argue later that the likelihood of bleaching differs with different caves. Does using the central age model for the samples with low over-dispersion change the results appreciatively? Also, could the high over-dispersion for some samples be because of mixing? Maybe not, because of the hard breccias?

line 11 – How can you have large deficiencies in Pb-210 but not in the U-238 decay chain? Isn't Pb-210 part of the U-238 decay chain.

Referee #2 (Remarks to the Author):

The majority of referees comments concern issues with radiometric dating, which is not my area of expertise.

I am pleased to see that the authors have responded favourably to my suggestion that the extinction of *G. blacki* can be usefully viewed in the context of later megafaunal extinctions in North America

and elsewhere.

Referee #3 (Remarks to the Author):

I fully recommend the publication of this work on the “demise of *Gigantopithecus blacki*” that integrates – to my knowledge – the largest range of analyses (chronology, environment, behaviour) on the subject. The authors improved the presentation of their results in the main Text.

I still have few comments :

Lines 52-53 in the abstract: “Its demise is enigmatic considering that it was one of the only primates to go extinct in the last 2.6 million years when other great apes, including orangutan, survived up to the present”.

This statement is not exact, since in the last 2.6 million years, some other Asian primates went extinct like *Procynocephalus* (see line 123) and, moreover, the Order of Primates also includes hominins (in continental Asia *Homo erectus*, Denisovans, *Homo sapiens*). So, I suggest to modify this sentence to: “Its demise is enigmatic considering that, among known pongines, it was the only taxon to go extinct in the last 2.6 million years when orangutans survived up to the present”.

Line 84: “...an increase in tooth size...”. Is there also a change in the size of the mandibles, that would suggest an allometric relationship? See also line 213 “While *G. blacki* increased in size...”.

Which parameter this increase in body size is based (tooth crown surface, mandible size?). Add just this precision.

Line 90: “But by the time of Hejiang, *G. blacki* had a dramatic range reduction to just Guangxi”. Some publications also mentioned the possible presence of *Gigantopithecus* at that time in Vietnam and Thailand. So, I suggest “... *G. blacki* had a dramatic range reduction to Guangxi, considering the region studied (Southern China).”

Line 119: *Podocarpus* ?

Line 122: ... (in relative large numbers)

Line 124: ... (in relative small numbers)

Line 122-125: The Figure 3c shows in fact two drops in the relative number of the *G. blacki* teeth (vs *P. weidenreichi*), the first one just before the EW transitional phase, and the second one during the EW. But these results have probably a different meaning, the first one is probably related to the composition of the faunal assemblage and the second one is linked to the extirpation of *Gigantopithecus*. If it is correct, a small precision is needed in the Text.

Line 224-226: “When compared to other well-known extinction events in North America and Australia, this megafaunal extinction occurred much earlier and in the absence of *H. sapiens*.”

This statement seems rather odd to me. First, the extinction of species (and turnover of faunas) was a common phenomenon in the evolution (the present analysis which demonstrates that the major causes for the demise of *Gigantopithecus* were environmental, is a good example) and, secondly, at the time of the demise of *Gigantopithecus*, *Homo sapiens* was not present in Asia. This is different from the extinction of megafauna aggravated by the activities of modern humans from ~ 50,000 years ago in Australia, and later in North America.

But that raises an interesting question not mentioned by the authors in the Text: Among the 11 Chinese sites analyzed, are there some faunal assemblages which contain *Gigantopithecus* associated with archaic hominins prior to the EW? And what about the 11 faunal assemblages after the EW?

I suggest this sentence (if I well understood what the authors want to explain): “When compared to other well-known extinction events in North America and Australia influenced (or accelerated) by Homo sapiens, no evidence support here that archaic hominins might have played a role in this earlier extinction event of megafauna in southern China”.

Figure 3: The authors should add in the Fig3 (d and f), the image of G. blacki (red) and P. weidenreichi (blue) as in Figure 2 c (the small silhouettes on the left).

Legend of the Figure 3 c: The authors chose to keep the NISP, but I suggest this change in the legend: These relative percentages are used as rough proxy for the relative abundance of G. blacki in comparison with P. weidenreichi in each site”.

Referee #4 (Remarks to the Author):

I will comment only on ESR and U-series results. As the topic of the paper lies on a chronological framework it is important to have reliable dates. The work performed in this paper is impressive and many valuable information is provided by the results. However I stay very skeptical about some results which preclude the publication in this state.

1/The relation with the dated samples, particularly teeth dated, and the stratigraphic sketch for each site, is not clear at all. It is well known that breccias are not very suitable formations to be dated by paleodosimetric methods for obvious reasons such as karstic racking, colluvium deposit, water circulation, sediment mixing and secondary carbonatation phenomena.

For instance in Yanliang site I have many difficulties to establish a relation with breccia containing Gigantopithecus and the speleothem which underlies. How can you claim that the underlying speleothem corresponds to F2 layer ? For each site, photos do not help, or very occasionally, to understand the stratigraphic succession.

In Shuangtan site, photos are unclear and we can not observe any difference between G, F2 and F3 layers. For F1 layer how can you say that this is the same layer in each part of the walls ? Where do come from the teeth analysed by LA ICPMS ? No detail is given for this part excepting that the authors consider the ages as minima. I am aware of the difficulties to make relationship with different layers in a cave and it seems very risky to make a Bayesian model which is by definition based on the stratigraphic location of the samples. For instance, is it relevant to include the F1 sample which yields a corrected age of 347 ± 346 ka ?

For Yixiantian cave, I have the clear impression that you make stratigraphy using dates !

In Baikong cave, why did not you take a sample in the lower speleothem included in the G breccia ? I did not find the result obtained on the F1 layer.

2/ ESR/U-Series ages on teeth

The authors decided to determine Equivalent doses (D_e) by using SSE function. Though no information is given (growth curves, adjusted r^2). I wonder wether the SSE does not overestimate the values, particularly for the greater equivalent doses. Would it be possible to have short explanation for this choice ? and to present a comparative table with results obtained by SSE and ExpLin functions ?

The authors assume an equilibrium of the decay chain after ^{230}Th both in enamel and dentine. I made calculations for some samples both by using the parameters proposed in the text and using also a disequilibrium of 0.7 in enamel and 0.5 in dentine with 0% water in enamel. As I did not find

any data on the subtracted enamel layer on each side, I used 100 ± 20 micrometers.

US-ESR(article) US-ESR recalculated US-ESR disequ
BAPANG 324 ± 44 496 +180/-154 615 +187/-168
QUEQUE 897 ± 78 923 +112/-108 1033 +114/-111
YANLIANG*1214±262(786±262) 2233 +1157/-433 2381 +1040/-470
BAIXIAN 181 ± 12 347 +34 /-33 360 +32/-31
SHUANGTAN357 ± 42 341 +98 /-84 441 +99/-87
271 ± 31 266 +47 /-42 316 +52/-40
ZHANGWANG714 ± 43 766 +64 /-79 776 +65/-67

For Yanliang, there is a large discordance between the age presented in the table S14 of 1214 ka and the age presented in the table S15 of 786 ka. Both ages have the same error range. Moreover, the results that I got are not in agreement with those obtained by the authors.

For Queque, Shuangtan and Zhangwang, there is an agreement between both calculations even though the disequilibrium has a logical tendency in increasing the ages. For the other sites, there is a disagreement that I can not understand and which leaves me skeptical !

3/ ESR dating on quartz

Seven quartz samples from 4 sites have been analysed by ESR. From the author himself the results are not reliable for different reasons well explained in the text. Ti center yields flat growth curves which do not allow any De determination and when it is mathematically possible, the discrepancy with Al center result is high, which is not a good indicator of reliability. For Ti center, I do not understand the difference between option D and E (one for measuring , the other for comparing with Al). I prefer the use of g-value for determining the used peaks. The goal is to get ages so, of course you compare the Ti results with those obtained by Al center ! Would that mean that some results are reliable and not the others ?

All the ages were calculated by taking into account a weight of (1/12) putting the maximum of weight on the natural sample even though the authors used also 1/52 and equal weight (EW).

The natural aliquote is the only « naturally irradiated » sample and remains the most important point in the determination of the growth curve which should cross this point for getting a reliable De value. CQQ looks underestimated as suggested by the results of the other dating methods. One sample of Zhangwang is overestimated while the other can not yield any result. Only Baikong samples are able to give results by using Al center while Ti center can not be used for dating the sites.

For Daxin, a strong interfering signal could be attributed to Fe³⁺ which corresponds to its g-value around $g = 2$. I am not sure that this signal can modify the Al signal height. However, the larger signal shown in fig7E could probably modify the aluminium height. For Zhanwang, why the gamma and cosmic doses are different for both samples (different granulometry) ?

For all these reasons and questions suggested by the text, is it possible to estimate a period during that Gigantopithecus disappeared ? I am not convinced at all and I think the paper should be

reconsidered and completely rearranged in a more appropriate format. As it stands it looks like a book.

Author Rebuttals to First Revision:

Response to Referees comments R2

Referee #1

Main text

Lines 131-135 – You don't explain the significance of the differences in stable isotopes. You do in the supplementary, but you need to have a sentence here.

This particular section is the results not the discussion of the stable isotopes data – we discuss the significance of the differences in the stable isotopes in a later paragraph L168-180. However, in the interests of clarity we have accentuated the similarities and difference between *G. blacki* and *P. weidenreichi* in the stable isotope results data by adding ‘increases slightly’ ‘are similar’ and ‘changing to’ L132-135.

Supplement

Lines 40-41 – I do not understand why the disc locating procedure was done before any heating. All grains would get the same heating regardless. Can you explain.

Just to clarify the heating process referred to is not the preheat but rather the heat up to the stimulation temperature. If the single-grain discs are heated up to their stimulation temperature (270°C in the case of feldspars) prior to single grain discs finding procedures then if there is an issue, such as the disc finding procedure is extended or fails, then the grains will receive an unnecessary or extended heating. This disc then cannot be used again because it has already been held at this temperature for an extended period. This is problematic when a sample returns very few grains and results must be gained from each grain, or if the disc locating procedure is not as efficient on older single-grain discs. This is standard but optional procedure as discussed in Duller et al., 1999. The text has been modified for clarity L40 “The single-grain disc locating process was programmed to occur before any disc heating to stimulation temperature to ensure that differences in the disc locating procedures would not result in grains receiving an extended heating duration within the SAR cycles”

line 78 – In response to reviewer comment about using the 12% value for internal K, you mentioned some measurements on some grains. Could you include that procedure here?

These procedural details can certainly be included – they were not originally included because the results confirmed the results of the reference cited and we were trying to prevent the Supp section from becoming too large. But in the interests of completeness they have now been included. Text has been added “Sample mineralogy was determined using X-ray Diffractometry. Samples were mounted on a silicon crystal low background holder, and diffractograms were collected from 5° to 90° 2θ using a PANalytical X'Pert pro MPD diffractometer, using 45 kV, 40 mA and CuKα radiation at 5° 2θ.min-1. Identification of

minerals was undertaken using PANalytical's High Score Plus v2.2.4 software, with ICDD PDF2 and PAN-ICSD databases. Detection limits depend on crystallinity but are typically around 0.1-0.5 wt%. In addition, the K content of the samples was estimated using a PANalytical Epsilon 3 XL X-ray Fluorescence spectrometer with conditions of 50 kV for 80 seconds with 100 µm Ag filter, 50 kV for 80 seconds with 300 µm Cu filter, 20 kV for 300 seconds with 200 µm Al filter, 12 kV for 120 seconds with 50 µm Al filter and 5 kV for 200 seconds with no filter. The estimated K content of the samples varied from ~10-14% with some samples containing Na feldspars and others containing some quartz (despite rigorous mineral separation procedures)".

line 123 – I agree the 270°C stimulation provided the best results from the tests, but it did not have the lowest residual value.

This is true – we have amended the sentence to read 'one of the lowest residuals'

line 151 – Could you explain how you determined the field saturation. I understand that you showed the old sample was in saturation, but could you explain how the value of saturation was obtained – so that it could be compared with the D_0 values in the table. I was not clear.

Saturation was assumed to be 2 x D_0 value (85% saturation) as expressed by Wintle and Murray (2006). Thus the field saturation data which produced a 2 x D_0 value of 1920 ± 204 Gy provide a comparison to test the D_0 values from the procedural tests – we added the following text to L164 "We assumed that this field saturation must be greater than or equal to the 2 x D_0 , which in this test sample was 1920 ± 204 Gy. This indicates that the field saturation is fairly close to the laboratory saturation using these signals, thus the fading is negligible, but however to be conservative we used a fading rate of 2.0 % per decade. We used the fading correction method by³². This field saturation value also provides a comparison to test the D_0 values obtained from the procedural tests and provides an upper limit for testing the saturation in these sample. All the D_e values obtained are well within this range".

Wintle, A. G., & Murray, A. S. (2006). A review of quartz optically stimulated luminescence characteristics and their relevance in single-aliquot regeneration dating protocols. *Radiation measurements*, 41(4), 369-391.

lines 161-163 – It probably does not make difference with the low fading rates you used, but shouldn't the fading correction be done on the age and not the equivalent dose? That is what Huntley-Lamothe did.

This is correct, for these calculations I used a fading model produced by Sebastian Huot based on the fading correction of Lamothe et al., 2003 in which the calculated D_e and the dose rate are entered into the model and the fading corrections are made at the same time that the age is generated thereby estimating a fading corrected age – this can then be compared to

the non-fading corrected age. Therefore, we have modified the text to reflect this L179
“MAM was applied to the equivalent doses and the MAM De generated age was corrected
according to the results of the anomalous fading tests”

line 170 – The table says 5% but here you say 10%

The footnote should read 5% not 10% - this has been corrected.

lines 9-17 under discussion – Just because some samples might be partially bleached
does not mean all of them are. I do not see the logic in using an argument of
consistency to apply MAM to all. In fact, you argue later that the likelihood of bleaching
differs with different caves. Does using the central age model for the samples with low
over-dispersion change the results appreciatively? Also, could the high over-dispersion
for some samples be because of mixing? Maybe not, because of the hard breccias?

There would have been very little if any post depositional mixing in the breccia deposits
because the majority of the breccias are capped by a hard flowstone that was precipitated
fairly quickly after deposition. It is likely that some of the caves experienced a differing
degree of bleaching but the majority of OD values are consistently above 50%. The most
conservative estimation of the burial dose is to apply the MAM in these depositional
environments. In all the cave sites I have worked on in SEA I have never yet found a truly
well bleached breccia deposit, its nature, usually as a colluvial flow into the cave through
sinkholes, rarely affords full bleaching potential for all of the breccia matrix. Hence the
conservative approach. However, for a few samples the MAM could not be applied (as seen
in the lack of a MAM line in Fig S4 and S5) as the MAM severely underestimated the burial
age as suggested by the results of the independent age estimates. Thus, the CAM was used for
these few samples. Therefore, the text was misleading in stating that the MAM was applied to
all samples it should have said to the majority of samples. This has now been modified “the
majority of the resulting single-aliquots and single-grains have been analysing using a
minimum age model (MAM)”.

line 111 – How can you have large deficiencies in Pb-210 but not in the U-238 decay
chain? Isn't Pb-210 part of the U-238 decay chain.

We describe large deficiencies in ^{210}Pb ‘compared’ to ^{226}Ra and smaller deficiencies in ^{226}Ra
‘compared’ to ^{238}U (CBAX) (L111) indicating that some daughter isotopes in the chain have
experienced more deficiencies than others, as a function of the differing mobility of the
daughter verses the parent isotopes. Therefore, we are discussing the different proportions of
deficiencies within the same ^{238}U chain.

Referee #2 (Remarks to the Author):

I am pleased to see that the authors have responded favourably to my suggestion that the extinction of *G. blacki* can be usefully viewed in the context of later megafaunal extinctions in North America and elsewhere.

Thank you for this helpful suggestion

Referee #3 (Remarks to the Author):

I fully recommend the publication of this work on the "demise of *Gigantopithecus blacki*" that integrates – to my knowledge – the largest range of analyses (chronology, environment, behaviour) on the subject. The authors improved the presentation of their results in the main Text.

I still have few comments :

Lines 52-53 in the abstract: "Its demise is enigmatic considering that it was one of the only primates to go extinct in the last 2.6 million years when other great apes, including orangutan, survived up to the present".

This statement is not exact, since in the last 2.6 million years, some other Asian primates went extinct like *Procynocephalus* (see line 123) and, moreover, the Order of Primates also includes hominins (in continental Asia *Homo erectus*, Denisovans, *Homo sapiens*). So, I suggest to modify this sentence to: "Its demise is enigmatic considering that, among known pongines, it was the only taxon to go extinct in the last 2.6 million years when orangutans survived up to the present".

We did actually use the phrase 'one of the only' not 'the only' but we do agree that the use of primates is misleading and inaccurate considering the number of Asian primates and *Homo* species that went extinct during this period. However, we do feel that 'among known pongines' doesn't really emphasize the significance of this extinction. Instead we have used L52 'it was one of the few Asian great apes to go extinct...'

Line 84: "...an increase in tooth size...". Is there also a change in the size of the mandibles, that would suggest an allometric relationship?

As all four discovered mandibles are incomplete it's hard to interpret any type of allometric relationship, especially considering the issue of sexual dimorphism. Furthermore, with only 4 known mandibles available to research the sample size is not a large enough to see any patterns.

See also line 213 "While *G. blacki* increased in size...". Which parameter this increase in body size is based (tooth crown surface, mandible size?). Add just this precision.

The increase in body size is based on an increase in tooth size, so the text has been changed to "While *G. blacki* increased in tooth size...". We have also added 'implying an increase in body size also'.

Line 90: "But by the time of Hejiang, *G. blacki* had a dramatic range reduction to just Guangxi". Some publications also mentioned the possible presence of *Gigantopithecus* at that time in Vietnam and Thailand. So, I suggest "... *G. blacki* had a dramatic range reduction to Guangxi, considering the region studied (Southern China)."

We discuss each of the proposed evidence for *G. blacki* outside of China in detail in the Supplement 1 (L407) and we conclude that none of the evidence is convincingly *G. blacki*. We do not exclude the possibility that *G. blacki* may have existed in these regions but as of present day we have not seen any robust evidence to persuade us otherwise. Thus, the sentence stands.

Line 119: Podocarpus ?

corrected

Line 122: ... (in relative large numbers) added

Line 124: ... (in relative small numbers) added

Line 122-125: The Figure 3c shows in fact two drops in the relative number of the *G. blacki* teeth (vs *P. weidenreichi*), the first one just before the EW transitional phase, and the second one during the EW. But these results have probably a different meaning, the first one is probably related to the composition of the faunal assemblage and the second one is linked to the extirpation of *Gigantopithecus*. If it is correct, a small precision is needed in the Text.

We agree and have added the following to the main text L201 "changes in the composition of the fauna and widespread faunal turnovers (Fig. 3c and S2), and to the Fig 3 caption "The relative number of *G. blacki* teeth declines just before the transitional phase representing a change in faunal composition and during the transitional phase representing the extirpation of *G. blacki*."

Line 224-226: "When compared to other well-known extinction events in North America

and Australia, this megafaunal extinction occurred much earlier and in the absence of *H. sapiens*.”

This statement seems rather odd to me. First, the extinction of species (and turnover of faunas) was a common phenomenon in the evolution (the present analysis which demonstrates that the major causes for the demise of *Gigantopithecus* were environmental, is a good example) and, secondly, at the time of the demise of *Gigantopithecus*, *Homo sapiens* was not present in Asia. This is different from the extinction of megafauna aggravated by the activities of modern humans from ~ 50,000 years ago in Australia, and later in North America.

But that raises an interesting question not mentioned by the authors in the Text: Among the 11 Chinese sites analyzed, are there some faunal assemblages which contain *Gigantopithecus* associated with archaic hominins prior to the EW?

No, there is no evidence that humans and *G. blacki* overlap either spatially or temporally in this region in the 11 pre-EW caves.

And what about the 11 faunal assemblages after the EW?

Only 2 of the 11 non-*G. blacki* caves (post EW) analysed had any evidence of humans and this evidence was limited to the upper levels of the stratigraphy.

I suggest this sentence (if I well understood what the authors want to explain): “When compared to other well-known extinction events in North America and Australia influenced (or accelerated) by *Homo sapiens*, no evidence support here that archaic hominins might have played a role in this earlier extinction event of megafauna in southern China”.

This sentence was added to the text in the last round at the request of reviewer 2 who wanted it emphasised that this extinction occurred earlier than North America and in the absence of humans. We agree with your rearrangement of the sentence and have added these changes in to the main text.

Figure 3: The authors should add in the Fig3 (d and f), the image of *G. blacki* (red) and *P. weidenreichi* (blue) as in Figure 2 c (the small silhouettes on the left).

These small images have been added to Fig 3

Legend of the Figure 3 c: The authors chose to keep the NISP, but I suggest this change in the legend: These relative percentages are used as rough proxy for the relative abundance of *G. blacki* in comparison with *P. weidenreichi* in each site”.

This has been added

Referee #4 (Remarks to the Author):

I will comment only on ESR and U-series results. As the topic of the paper lies on a chronological framework it is important to have reliable dates.

This is true that the chronology forms the framework for the study hence why we have included data from six different dating techniques to check for consistency and reliability, but it is also one of three very important aspects of this research with environment and behaviour being the other two equally important aspects.

The work performed in this paper is impressive and many valuable information is provided by the results. However I stay very skeptical about some results which preclude the publication in this state.

We would like to express our gratitude to the reviewer for dedicating their time and effort to help us enhance the manuscript. We acknowledge and appreciate the scepticism, as it often serves as a catalyst for scientific improvement. Nevertheless, we respectfully disagree that the concerns raised are sufficient to justify precluding publication.

1/The relation with the dated samples, particularly teeth dated, and the stratigraphic sketch for each site, is not clear at all.

We have provided a detailed plan of each cave to identify the exact location of the fossils and samples for dating all in the supplementary section 2. In addition, we have provided a composite stratigraphy to identify the stratigraphic relationships and the exact location of the samples in relation to these stratigraphic relationships. Finally, we have summarised these relationships in the age models showing the age estimates from each layer and provided a photogrammetry model of the cave and sediments for context – this provides the best angle to view the caves and sediments. We have presented this for all 22 caves which greatly extends the supplementary section by 22 pages but we believe that this level of detail is valuable for providing vital context for the dating. The location of the teeth dated by coupled US-ESR has been indicated by the individual coupled US-ESR symbol (green hexagons), while the location of teeth dated only by U-series dating is indicated by light blue circles. However, as this is obviously not clear enough we have added in this information to each relevant caption. Please note that many of the teeth were measured using both U-series and coupled US-ESR techniques (Table S15) – in these instances only the green hexagon is used to indicate the location of the tooth.

It is well known that breccias are not very suitable formations to be dated by paleodosimetric methods for obvious reasons such as karstic racking, colluvium deposit, water circulation, sediment mixing and secondary carbonatation phenomena.

We find ourselves in partial agreement and disagreement with the reviewer's comments. We concur that breccia deposits may not be the most ideal for dating, due to the reasons mentioned. However, we and many others have still managed to obtain reliable age estimates that are supported by other independent dating techniques (e.g., Westaway et al., 2017; Louys et al., 2022, Duval et al., 2022). Despite the challenges, disregarding an entire and abundant collection of fossils and potentially valuable insights into hominid evolution simply because the situation is not perfect would be ill-advised. As the reviewer possesses a strong background in geochronology they would undoubtedly be aware that ideal dating situations rarely, if ever, exist. This reality is precisely why establishing accurate chronology can be a complex, but not an impossible task.

For instance in Yanliang site I have many difficulties to establish a relation with breccia containing Gigantopithecus and the speleothem which underlies. How can you claim that the underlying speleothem corresponds to F2 layer ?

For all the caves we have tried to simplify the stratigraphy so that the relationships are clear. At Yanliang (as with the other sites) the stratigraphic drawing is a composite stratigraphy which has projected the different stratigraphic units onto one composite image so that the relationships are clear – as is standard practice in a cave environments (e.g., Düringer et al., 2012). This has been added to the caption for clarity. The breccia in this location was overlying the laminated flowstone, which we named F2. We have included a further two images clearly showing the relationship between the overlying F1 flowstone, breccia and underlying F2 flowstone in the Yanliang figure (Fig S1g). The grey section in the middle of the image is the limestone wall rather than a flowstone. This may be confusing the issue, so we have changed the limestone colour in all 22 figures to make this differentiation clearer.

For each site, photos do not help, or very occasionally, to understand the stratigraphic succession.

We have added new stratigraphic photographs to a further eleven cave sites to provide further evidence for the stratigraphic relationships between the layers.

In Shuangtan site, photos are unclear and we can not observe any difference between G, F2 and F3 layers.

Shuangtan represents a fissure cave and is very narrow. This makes photography very difficult, however we have constructed a photogrammetry image that provides a view of the entire cave that is physically impossible to observe in reality. We believe that this provides a

unique perspective on the site. As for the differences between the *G. blacki* bearing breccia and flowstones F1-3 they are very distinct. We have included a more detailed image displaying the relationship between the overlying F1 flowstone and the *G. blacki* bearing breccia to make these differences clearer.

For F1 layer how can you say that this is the same layer in each part of the walls ?

We sampled the F1 on both sides of the cave as depicted by the U-series symbol of either sides of the stratigraphic drawing. Despite one of the results having a large error (CSHT-F1) the resulting ages are in agreement within error margins suggesting that they are indeed the same flowstone unit. They are also precipitated at exactly the same height so the most parsimonious explanation is that they are the same flowstone that capped the breccia the unit. This is also confirmed by the excavators when they first excavated the site.

Where do come from the teeth analysed by LA ICPMS ? No detail is given for this part excepting that the authors consider the ages as minima.

The teeth dated by U-series LA-ICPMS were mostly discovered at the location depicted by the coupled US-ESR symbol (green hexagon) as both measurements were mostly conducted on the same tooth. However, for some teeth only the U-series LA ICPMS was conducted, the location of these teeth have been indicated by light blue circles. This has been added to the caption for clarity.

I am aware of the difficulties to make relationship with different layers in a cave and it seems very risky to make a Bayesian model which is by definition based on the stratigraphic location of the samples.

In most of the caves studied here there are only one or two main layers found, a breccia unit that is overlain and/or underlain by a flowstone unit, as depicted by the age models. In these circumstances the relationships between the layers are very clear. In some of the caves such as Daxin, Queque and Baikong there is an additional silt layer, but this is often sterile and does not contain *G. blacki* evidence. Therefore, we disagree that the application of a Bayesian model to these layers is risky, we have taken over seven years of fieldwork to understand the stratigraphic relationships in these caves and collected samples in accordance with these established relationships.

For instance, is it relevant to include the F1 sample which yields a corrected age of 347 ± 346 ka ?

This sample is one of the only samples to be largely contaminated as indicated by the difference between the uncorrected and corrected values. As such, this result has always been excluded from the age models due to its imprecision, however this wasn't made clear in the modelling section so a justification for this exclusion has been added to the model discussion in Supp section 9. L67 "A small number of data were excluded from the age models based on methodological issues. For U-series dating of flowstones at Shuangtan, the upper F1 proved problematic due to high Th contamination issues resulting in a very imprecise corrected age.

Thus, samples CSHT-F1, and CSHT-F1B-2017 (Table S21) were not used in the model for this reason”.

For Yixiantian cave, I have the clear impression that you make stratigraphy using dates !

This is untrue, when we excavated the site we observed evidence of an older section that was protected by a capping flowstone (later dated to ~500 ka). We interpreted this entire breccia as being deposited in two phases with later fluvial action eroding through the boundary to create the gap and later depositing F2. We identified the source of the fluvial action as a sinkhole to the rear of this section. This interpretation was later confirmed by the dating. This drawing also represents a composite stratigraphy in an effort to include all the stratigraphic layers in one image. This has been added to the caption for clarity.

In Baikong cave, why did not you take a sample in the lower speleothem included in the G breccia ? I did not find the result obtained on the F1 layer.

Previous excavations at Baikong suggested that the *G. blacki* evidence was concentrated in the upper section of the breccia. It is correct that this upper section is underlain by a flowstone but the appearance and morphology of the *G. blacki* material in this cave indicated that the underlying flowstone would be too old for U-series analysis (as confirmed by the dating results) so instead we focussed on dating the teeth using coupled US-ESR.

2/ ESR/U-Series ages on teeth

The authors decided to determine Equivalent doses (D_e) by using SSE function. Though no information is given (growth curves, adjusted r^2). I wonder whether the SSE does not overestimate the values, particularly for the greater equivalent doses. Would it be possible to have short explanation for this choice ? and to present a comparative table with results obtained by SSE and ExpLin functions ?

Indeed, Duval (2015) demonstrated that the SSE method could overestimate D_e values and unconstrained parameters. However, a subsequent study in 2016 showed that the SSE method could reliably estimate D_e values, provided certain conditions were considered and adhered to (see Duval and Grün, 2016). In other words, the SSE can be used as a reasonable approximation of the DSE provided the Max/ D_e ratio value remains within a given range (see Table 4 in Duval and Grün, 2016)

In our calculations of D_e values, we employed the McDoseE 2.0 program, which utilizes a Monte-Carlo iteration approach combined with a Metropolis algorithm (see Joannes-Boyau et al., 2019). Consequently, we generated approximately 100,000 growth curves on average, and we refrained from calculating a mean equation, as it would contradict the principles of our approach, which instead provides a distribution of the solutions. We do not believe that the SSE method overestimated D_e values when used according to Duval et al. (2016). Instead,

we think that the radical distribution within the ESR signal of enamel would contribute more significantly to this issue.

Finally, it is worth noting that the ExpLin functions are rarely (if ever) used for tooth enamel, as they generally represent a more accurate behaviour for quartz saturation curves. There is no physical or empirical basis for using the ExpLin function, unlike the SSE or DSE functions.

The authors assume an equilibrium of the decay chain after ^{230}Th both in enamel and dentine. I made calculations for some samples both by using the parameters proposed in the text and using also a disequilibrium of 0.7 in enamel and 0.5 in dentine with 0% water in enamel. As I did not find any data on the subtracted enamel layer on each side, I used 100 ± 20 micrometers.

US-ESR(article)	US-ESR recalculated	US-ESR disequ		
BAPANG	324 ± 44	$496 +180/-154$	$615 +187/-168$	
QUEQUE	897 ± 78	$923 +112/-108$	$1033 +114/-111$	
YANLIANG*	1214 ± 262	(786 ± 262)	$2233 +1157/-433$	$2381 +1040/-470$
BAIXIAN	181 ± 12	$347 +34 /-33$	$360 +32/-31$	
SHUANGTAN	357 ± 42	$341 +98 /-84$	$441 +99/-87$	
	271 ± 31	$266 +47 /-42$	$316 +52/-40$	
ZHANGWANG	714 ± 43	$766 +64 /-79$	$776 +65/-67$	

We thank the reviewer for such a thorough undertaking. For the enamel layer removal, it is stated on line 2 of the SI section 6: Coupled US-ESR dating of teeth paragraph that we "...stripped the outer $\sim 100 \mu\text{m} \pm 10\%$ on each side [of the enamel]". We agree that by changing the disequilibrium to 0.7 in enamel and 0.5 in dentine with 0% water in enamel, would without a doubt change the calculated ages. There is no scientific reason why the disequilibrium should equal 0.7 in enamel and 0.5 in dentine for all samples and all sites presented. Certainly, measuring the entire U-series decay chain would be more accurate than assuming the value, yet we are not able to do so. Therefore, instead of attributing a random disequilibrium value for each dental tissue, we have to assume that the decay chain is at equilibrium within the sample. It is also impractical to calculate an age for every possible scenario.

As clearly stated in our manuscript, and as the reviewer noticed, the ages we present are calculated with a post ^{230}Th equilibrium both in enamel and dentine.

For Yanliang, there is a large discordance between the age presented in the table S14 of 1214 ka and the age presented in the table S15 of 786 ka. Both ages have the same error

range. Moreover, the results that I got are not in agreement with those obtained by the authors.

For Yanliang, the reviewer is correct and the value in Table S15 is the correct one and we apologize for the mistake. The wrong D_e value was entered for the model the second time, we have now changed the value in Table S14 to the correct one. This has no bearing on the final ages as the results in Table S15 were used in the age model.

For Queque, Shuangtan and Zhangwang, there is an agreement between both calculations even though the disequilibrium has a logical tendency in increasing the ages. For the other sites, there is a disagreement that I can not understand and which leaves me skeptical !

Since, the authors are uncertain of the parameters selected or software used by the reviewer to perform the calculation we are unable to comment on the discrepancy between our ages and the reviewers. But all our parameters can be found in the table S14. We have used both the US-ESR program from Shao et al. (2014) and the DATA program from Grün (2009) to calculate our ages. Both programs have been used and described in many publications. Both independent programs give comparable results.

3/ ESR dating on quartz

Seven quartz samples from 4 sites have been analysed by ESR. From the author himself the results are not reliable for different reasons well explained in the text. Ti center yields flat growth curves which do not allow any D_e determination and when it is mathematically possible, the discrepancy with Al center result is high, which is not a good indicator of reliability.

We agree, this is why the majority of the ESR quartz ages (from sites CQQ and CBAIK) have been excluded from the final age models. This results in only a slight change to the age models for Queque and Baikong as they had previously been entered in a minimum age so had little influence over the age calculation. We have presented all the ages in the supplementary section (Supp section 7) with a justification for their exclusion from the age models in the modelling section (Supp section 9).

For Ti center, I do not understand the difference between option D and E (one for measuring , the other for comparing with Al). I prefer the use of g-value for determining the used peaks.

The text has been modified accordingly. The g values have been provided.

The goal is to get ages so, of course you compare the Ti results with those obtained by Al center ! Would that mean that some results are reliable and not the others ?

Yes, indeed. There is an extensive discussion around the reliability of the ESR data set in Supplementary Material. We also added a short summary at the end of the discussion in order to explain what are, in our opinion, are the more reliable age results for each sample.

All the ages were calculated by taking into account a weight of (1/12) putting the maximum of weight on the natural sample eventhough the authors used also 1/52 and equal weight (EW).

The natural aliquote is the only « naturally irradiated » sample and remains the most important point in the determination of the growth curve which should cross this point for getting a reliable D_e value.

This is the reviewer's opinion. We do not fully agree with this statement. Sometimes the natural point may be totally off, far from the trend given by the other points and the resulting 1/12 data weighing may then bias the fitting results. This is why we carried out this comparison between various fitting options, in order to identify possible fitting artefacts, and evaluate the robustness of the D_e estimates.

CQQ looks underestimated as suggested by the results of the other dating methods. One sample of Zhangwang is overestimated while the other can not yield any result.

Only Baikong samples are able to give results by using Al center while Ti center can not be used for dating the sites.

For Daxin, a strong interfering signal could be attributed to Fe^{3+} which corresponds to its g-value around $g = 2$. I am not sure that this signal can modify the Al signal height. However, the larger signal shown in fig7E could probably modify the aluminium height.

While we agree with the comments above, we must emphasise that they are already included in the dedicated section of the Supplementary Material. The CQQ and CBAIK results have been excluded from the age model as they are maximum age estimates. which are at odds with the independent age estimates for the sites. We have added a justification for their exclusion in the modelling section 9 L71 "In addition, the ESR dating of quartz also proved problematic at some sites (as explained in Supp section 7). Therefore, samples QQ1 and CBAIK10 were excluded from the model due to the low quality of the ESR data with a low goodness-of-fit and D_e precision. Furthermore, they represent maximum ages estimates, which are at odds with the other independent age estimates for the site".

For Zhanwang, why the gamma and cosmic doses are different for both samples (different granulometry) ?

Thanks to the reviewer for spotting the typos in that table. The gamma and cosmic DR of the 2 CZW2 samples should indeed be the same. The values initially indicated for CZW2_90-

212 μ m were wrong. Please note that these corrections do not change any of the age results, since no ages could actually be calculated for sample CZW2_90-212 μ m.

For all these reasons and questions suggested by the text, is it possible to estimate a period during that Gigantopithecus disappeared ?

We believe it is possible based on this large body of data collected, which is unprecedented in the *G. blacki* research. By admission, the ESR quartz data are the weakest of the all the dating techniques but the remaining five dating techniques provide remarkable agreement across the sites that strongly suggest a window of extinction when *G. blacki* drops out of the fossil record (see Fig 2 a).

I am not convinced at all and I think the paper should be reconsidered and completely rearranged in a more appropriate format. As it stands it looks like a book.

As stated in previous R1 review, this research represents a huge amount of data with six dating techniques and eight proxies. To outline the detailed context, methods results and interpretations of all these proxies requires a large supplementary section. We have made sure that this section is well linked to the main text with frequent citing of where in the supplementary sections to find the supporting evidence. Unfortunately, the Supplementary Information does indeed take on the appearance of a 'book' with many chapters, but better that the readers have a chance to explore the raw data and justifications for themselves rather than omit or cut down on this supporting evidence.

Reviewer Reports on the Second Revision:

Referees' comments:

Referee #1 (Remarks to the Author):

I reread the main text and read in detail the supplementary material on luminescence dating, which is my area of expertise. I find that the authors have addressed all my previous concerns to my satisfaction. If the other reviewers agree I see no reason not to publish the paper.

Referee #3 (Remarks to the Author):

I went through the Text and, for my part, the article is ready for publication.

2 small comments in the Abstract:

Line 60: ...and many other primate populations. (no comma)

Line 64:*Pongo weidenreichi*... (full species name in italics)

Referee #4 (Remarks to the Author):

Thanks to the authors for their answers and corrections mainly about the samples location. I know well all the difficulties for taking samples in caves and I am aware that there is no ideal site of course. The explanations concerning the samples location in the stratigraphic sequences can be heard and the multi-method approach is valuable for constraining ages.

But I have still comments because the answers provided by the authors are not satisfactory. About the SSE and ExpLin functions, if I answer like the authors, this is word against word and the discussion does not progress. I have many examples for which SSE calculations are higher than ExpLin data with higher error range and R2 adjustment and specifically for old enamel samples. It would be simple to make a comparative table and explain the reader the reasons of the authors choice and explain why they decided to use SSE function.

For the age calculations, I calculated the ages from the authors' data with the same programs as that used by the authors (Shao et al., 2014 ; Grün, 2009). I presented three ages (the author ages, mines without disequilibrium and the third using a random 0.7 and 0.5 disequilibrium). I just used a random disequilibrium in order to have an idea of the age tendency. Dentine and cement when present, are close to bones constitution and are rarely considered as closed systems versus uranium uptake. Moreover, radon and radium are volatile elements and very few samples are in equilibrium from a gamma point of view. But as you did not perform gamma measurements on different tissues, you considered an equilibrium which can be detrimental for the age calculations particularly when uranium concentration is high in the tissues and when internal dose rate is predominant versus external dose as it is often the case in carbonated environment.

0% water content in enamel, in which water is intracellular, is recommended by Grün and Duval!

It is interesting to observe that for three sites, even when I do not use the disequilibrium, the ages

are different. I am still waiting for the reasons of this disagreement ? For Baixan, a non G. Blacki site, my ages are older than that of the authors and probably modifies the modelled age for this site. For ESR ages on quartz, I do not understand the final comment of this part saying that quartz probably absorbed a massive radiation dose responsible of the flat Ti DRC. Under these conditions, how can we accept the underestimated Al ages ? Radiation dose would have a large effect on Ti centers and another effect on Al center leading to an underestimation of the age ? On this point, I never observed a stable behavior in Ti center when it is saturated. In the major part of samples, we observe a decreasing of the intensity when saturation occurs !

The authors did not take into account my observations about Ti center (option D and E was modified one time and let all over the text line65-67, line100, line136, line141, table S20 etc....). This terminology is useful for the authors who quote themselves but who did not invented anything in the field of Ti center knowledge and its application for dating. Pionner papers should be mentioned (Toyoda, Tissoux et al. etc...). G-values were not replaced in the figures and so, it is difficult to compare with some signals for which the g-value is known in the literature.

The summary at the end of the SI 7 does not bring new highlight on the relevance of the ESR ages on quartz.

My opinion about the natural point as a major point for the building of the DRC will not change. I think that the authors are much more confident into the mathematics description with respect to the natural phenomenon ! We can use also a polynomial function fitting perfectly the points of a growth curve and having a very bad extrapolation leading to a wrong D_e value !

The final comment saying that the Al ages are underestimated lie on the comparison with results obtained by other methods but not when we look at the growth curves. The table S18 in which data weighting are compared is not conclusive for the few usable results. The authors calculated their ages with a $1/12$. So, when I look at the growth curve in front of the natural point which is under the curve, I can not say that the ages are underestimated.

Minor points

Figure 2 shows a vertical extinction window that is not very conclusive. It would be better to show this window as a function of time !

Yixiantian cave : I see only one speleothem dating result in the SI section 8 ?

Referee #5 (Remarks to the Author):

Dear Dr. Zhang et al.,

I find your article a very stimulating and comprehensive approach to the questions surrounding Gigantopithecus' extinction. I find only a series of minor issues with the paper, and fully support its publication.

Minor Issues:

Supplementary Data

Page 2. SI Section 8 is labeled U-Series of Flowstones, but on p. 97 it is called U-series Dating of Carbonates and Bone. Please reconcile this difference.

Table S12 The name of the cave Upper Pubu is not legible.

Page 67. Line 81 and 82. Your statement about why sediment is expected to be younger seems oversimplified to me, and incorrect. From my point of view, fauna and sediments can be coeval, being transported with fluvial sediment during flooding, or dying in caves and then being enclosed by LATER DEPOSITION of aeolian or fluvial sediments. I would expect the time differences to be small in the latter case, effectively making them coeval with respect to the uncertainties in your dating methods. You will need to remove this oversimplified statement and argue from a different point of view for your younger sediment vs. fossil ages.

Table S13, p. 76. Please verify that the last column in Table S13 is +/- 2 sigma and label it as such.

p.78

line 17 replace "thieved" with "sieved"

line 21 replace "nto" with "not"

line 30 replace "steps" with "step"

line 55 replace "close-system" with "closed-system"

line 61 replace "which" with "with"?

Table S14 don't understand text in footnote d : ThO ?

p. 84, line 73 This sentence is very confusing -missing some parentheses and wording is unclear. Please rewrite sentence.

p. 85 first sentence Not clear, please rewrite

line 117: this is unintelligible. Please rewrite

line 157: replace "gain" with "grain"

p.90 This Discussion section has considerable grammatical problems, it is intelligible to a knowledgeable reader, but needs to be carefully rewritten.

Section 3.2 Age Results

line 29: replace "it" with "the pattern"

Table S21

Put in a footnote that explains (If I am correct!) that sample codes with and F after the underscore are flowstones, and those with a B after the underscore are bones.

Table S22, p. 102

How is the age for the 25 bone samples calculated? Please explain.
How is the isochron age for Wuyun calculated?

Main document (merged pdf).

Line 708 replace "place" with "placed"

Line 709 put a space between 200 and micron

Line 709 replace "clean" with "cleaned"

Line 710 replace "preabation" with "preablation"

Line 723 please define DAD

Author Rebuttals to Second Revision:

Referees' comments:

Referee #1 (Remarks to the Author):

I reread the main text and read in detail the supplementary material on luminescence dating, which is my area of expertise. I find that the authors have addressed all my previous concerns to my satisfaction. If the other reviewers agree I see no reason not to publish the paper.

Thank you for all your helpful suggestions and for improving the quality of our manuscript

Referee #3 (Remarks to the Author):

I went through the Text and, for my part, the article is ready for publication.

2 small comments in the Abstract:

Line 60: ...and many other primate populations. (no comma)
removed

Line 64:*Pongo weidenreichi*... (full species name in italics)

Corrected, we thank you for a constructive review that has helped to improve our study

Referee #4 (Remarks to the Author):

Thanks to the authors for their answers and corrections mainly about the samples location. I know well all the difficulties for taking samples in caves and I am aware that there is no ideal site of course. The explanations concerning the samples location in the stratigraphic sequences can be heard and the multi-method approach is valuable for constraining ages. But I have still comments because the answers provided by the authors are not satisfactory.

We regret that the reviewer found our previous answers unsatisfactory. We genuinely hope that with our forthcoming response, we can address any remaining concerns the reviewer may have and provide the necessary clarification.

About the SSE and ExpLin functions, if I answer like the authors, this is word against word and the discussion does not progress. I have many examples for which SSE calculations are higher than ExpLin data with higher error range and R2 adjustment and specifically for old enamel samples. It would be simple to make a comparative table and explain the reader the reasons of the authors choice and explain why they decided to use SSE function.

We understand the reviewer's point, and we agree that our age calculations are based on [clearly stated] deliberate choices of parameters. Yet, we do believe that presenting all potential scenarios for the age calculation will only bring confusion to the debate. If we were

to include multiple situations, we should also calculate an age for Double Saturated Exponential (DSE), on top of ExpLin, perhaps even linear fit in some cases. By doing so we would present a minimum of 3 to 4 potential equivalent doses, and therefore 3 or 4 ages per sample. Quickly, we would end up with many ages per samples, depending on assumptions and parameter changes. Presenting many possible solutions will only be confusing to the reader in our opinion. In addition, we are not aware of any study demonstrating that the radiation-induced ESR signal in fossil enamel follows an EXP+LIN behaviour. Instead, several works showed that the DSE is also an appropriate function (Duval, 2015, Duval and Grun, 2016). However, because the DSE fitting is often quite challenging (there are 5 parameters to fit, and therefore at least 15 dose steps are usually advised), the behaviour of the signal may be approximated to a SSE, provided that Dmax is adjusted.

Moreover, while not faultless, we believe our approach has merits. We have applied consistency and systematic to our age calculations. We base our dating protocol on several published methodologies (cited in the text), which have rationalised our parameter choices for both the calculation of the dose response curves and age models. Ultimately, we present ages that can be discussed, and recalculated, as we present all our parameters. Yet, we believe these ages represent the best age estimations for the different deposits.

For the age calculations, I calculated the ages from the authors' data with the same programs as that used by the authors (Shao et al., 2014 ; Grün, 2009). I presented three ages (the author ages, mines without disequilibrium and the third using a random 0.7 and 0.5 disequilibrium). I just used a random disequilibrium in order to have an idea of the age tendency. Dentine and cement when present, are close to bones constitution and are rarely considered as closed systems versus uranium uptake. Moreover, radon and radium are volatile elements and very few samples are in equilibrium from a gamma point of view. But as you did not perform gamma measurements on different tissues, you considered an equilibrium which can be detrimental for the age calculations particularly when uranium concentration is high in the tissues and when internal dose rate is predominant versus external dose as it is often the case in carbonated environment.

We understand and appreciate the point of view of the reviewer. We agree that a potential disequilibrium in the decay chain would impact ages. We are not aware of studies highlighting a high mobility of radium within dental tissues, on the contrary. It is our understanding that only radon is a volatile element of the decay chain. Additionally, *Gigantopithecus* and *Pongo* teeth, similar to other hominid dental structure, do not have cementum encasing the enamel the way *equus* or certain *bovidae* teeth are organised.

The impact of ^{222}Rn degassing (disequilibrium in the decay chain) would vary greatly depending on the contribution of the internal dose and the actual deviation from equilibrium, meaning that no blank correction can be applied. Without knowing the actual disequilibrium history, the use of a random value seems arbitrary and unwarranted. We therefore believe that our approach that consist of assuming (as clearly stated in the text) equilibrium in the decay chain in the dental tissue remains the most appropriate in these circumstances.

0% water content in enamel, in which water is intracellular, is recommended by Grün and Duval! It is interesting to observe that for three sites, even when I do not use the disequilibrium, the ages are different.

We are not aware (including the author Duval himself) of a paper by Grün and Duval that has systematically analysed water content in hominid teeth. We have “assumed” a 3% +/- 1 interstitial water in the enamel based on our own experience (e.g. heated test of a pongo enamel at 60°C for 24h). Regardless, we have recalculated the ages using a 0% and 3% +/- 3 (see below) All age variations are within errors and therefore undistinguishable from one another.

Here are the results:

Bapeng (3% water) 329 +/- 45 ka; (0% water) 317 +/- 44 ka

Baxian (3% water) 181 +/- 13 ka; (0% water) 179 +/- 13 ka

Yanliang (3% water) 784 +/- 252 ka; (0% water) 773 +/- 251 ka

even when I do not use the disequilibrium, the ages are different.

Ages above for Bapeng, Baxian and Yanliang were recalculated with the same parameters previously described and the ages are identical as the one we presented before in the table S14. We are therefore unable to explain how and why the reviewer, using the same parameters and the same program by Shao et al 2014 does not obtain the same results as us. Perhaps, the reason lies behind the use of the updated dose rate from Guérin et al., 2011? (see footnote of Table S14).

To address diligently the comments from the reviewer we did recalculate some ages (broad range of ages) assuming a 50% ²²²Rn loss in both dental tissues (D: dentine and E: enamel) (although Rn loss is known to be atypical in enamel):

Bapeng

(equilibrium) 329+/-45 ka;
(50% Rn loss D) 341+/-48 ka;
(50% Rn loss D+E) 420+/-47 ka

Baxian

(equilibrium) 181+/-13 ka;
(50% RN loss D) 183+/-12 k;
(50% RN loss D+E) 192+/-12 k;

Yanliang

(equilibrium) 784 \pm 252 ka;
 (50% RN loss D) 793 \pm 252 ka;
 (50% RN loss D+E) 858 \pm 256 ka;

As our re-calculation shows, the impact of water or Rn loss (dentine) would be systematically within the error of our age estimations, making the age results undistinguishable.

Nonetheless, to acknowledge the comments of the reviewer, we have added to the SI the following paragraph to help readers understand the impact of parameters choice (e.g. water and Rn loss in dental tissues) for our age estimates:

“We have also investigated the potential variations of water content within the dental tissues (comparing 0% to 3% in enamel and 3% to 8% in dentine, respectively), and we found that the impact on the age calculation was marginal, with ages statistically indistinguishable from one another. Additionally, we have also considered the impact of disequilibrium in the decay chain after ²³⁰Th. While, we have assumed a ratio of 1 after thorium, the degassing of ²²²Rn is a possibility, which could potentially have shifted some of our results to older ages. We have calculated ages with a potential 50% radon degassing in dentine and then in both dental tissues (although Rn loss is known to be atypical in enamel). We have selected three sites that had a great variation of ages, dosimetry and uranium content, such as Bapeng [equilibrium = 329+/-45 ka; assumed 50% Rn loss in Dentine = 341+/-48 ka; and assumed 50% Rn loss both dental tissues = 420+/-47 ka], Baxian [equilibrium = 181+/-13 ka; assumed 50% Rn loss in Dentine = 183+/-12 ka; and assumed 50% Rn loss in both dental tissues = 192+/-12 ka]; and Yanliang [equilibrium = 784+/-252 ka; assumed 50% Rn loss in Dentine = 793+/-252 ka; and assumed 50% Rn loss both dental tissues = 858+/-256 ka]. As expected, the Rn loss increases slightly the calculated age, especially when considering a degassing in both tissues (unlikely). Regardless, the calculation shows, that impact of Rn loss (dentine) to be consistently within error of our age estimations when assuming equilibrium”.

For ESR ages on quartz, I do not understand the final comment of this part saying that quartz probably absorbed a massive radiation dose responsible of the flat Ti DRC. Under these conditions, how can we accept the underestimated Al ages ? Radiation dose would have a large effect on Ti centers and another effect on Al center leading to an underestimation of the age ? On this point, I never observed a stable behavior in Ti center when it is saturated. In the major part of samples, we observe a decreasing of the intensity when saturation occurs !

We do not disagree with the reviewers' comment. The summary at the end of SI has been

modified accordingly. Our only explanation is that we are possibly dealing with highly heterogeneous samples that may lead to meaningless dose estimates, in addition to the fact that the robustness of the ESR data set is below usual standards, which is why data interpretation should anyway be treated with caution.

The last paragraph of SI 7 now reads as follows:

As a final comment, it seems like most of the ESR results derived from the Al centre may be somewhat underestimated in comparison with the independent age control available at each site. While this most likely result from an underestimated DE estimate, this cannot be explained by a low thermal stability and/or saturation of the ESR signal, since the Al centre is frequently being used to date samples older than 2 Ma-old. Therefore, we suspect that the underestimation might originate from grain averaging effect. Since ESR analyses are based on multi-grain aliquots, the ESR intensities measured are naturally the result of the individual contribution of tens of thousands to hundreds of thousands of grains²²⁰. However, since the homogeneity of the quartz samples cannot be properly evaluated in the absence of single grain ESR analyses, we cannot exclude that if these samples were to include various populations of quartz with different bleaching histories or radiation sensitivity, the grain averaging effects on the ESR signals may lead to totally meaningless dose estimates (e.g., del Val et al., 2022).

As of now, there is a set of evidence suggesting that the quartz samples have absorbed a massive radiation dose, as suggested by the (i) apparently flat DRCs and (ii) high ESR intensities obtained for the Ti centre in many samples. However, on the other side, the ESR intensity of the Ti signal seems to keep increasing at much higher doses above 2 kGy (Fig. S8). In other words, it does not show this non-monotonic behavior that is typically observed at high irradiation doses²¹¹. Moreover, the D_e estimates obtained from the Al signal are <1500 Gy, i.e., not exceptionally high. Consequently, we presently do not have any other explanation for the underestimated dose estimates, and ESR age results, obtained from the Al centre than some grain averaging effects due to highly heterogenous quartz samples, in addition to the fact that the robustness of the ESR data set is below usual standards, which is why data interpretation should anyway be treated with caution.

The authors did not take into account my observations about Ti center (option D and E was modified one time and let all over the text line65-67, line100, line136, line141, table S20 etc...). This terminology is useful for the authors who quote themselves but who did not invented anything in the field of Ti center knowledge and its application for dating. Pionner papers should be mentioned (Toyoda, Tissoux et al. etc...). G-values were not replaced in the figures and so, it is difficult to compare with some signals for which the g-value is known in the literature.

Following the reviewer's comments, we modified Fig. S8 in order to indicate how ESR intensities of the Al and Ti signals were being measured. We hope this will help clarify any possible confusion. Moreover, we have modified the section 1.3 ESR dosimetry of SI section 7 in order to clearly explain the correspondence between the terminology used by Duval and Guilarte (2015) (i.e., Options D and E) and the g values. Once this correspondence has been properly introduced in the Method section, then we assume that there is no need to systematically refer to the two terminologies later on in the manuscript.

Finally, we do acknowledge that the whole relevant literature in the field of ESR dating of quartz might not be cited here, but we feel this goes beyond the scope of the present work. Our work is a dating application study, and there is no room for a proper description of the method. Instead we refer to other works that do include this literature (i.e., Duval and Guilarte, 2015; Duval et al., 2017). That said, we'd like to mention that a couple of papers by Toyoda et al. are actually being cited here.

The summary at the end of the SI 7 does not bring new highlight on the relevance of the ESR ages on quartz.

The summary has been rephrased based on the reviewer's comments (see above).

My opinion about the natural point as a major point for the building of the DRC will not change. I think that the authors are much more confident into the mathematics description with respect to the natural phenomenon ! We can use also a polynomial function fitting perfectly the points of a growth curve and having a very bad extrapolation leading to a wrong D_e value !

The reviewer mentions the natural phenomenon, but he/she may have missed the seminal work by Wagner and Woda (2007) describing the behaviour of the Ti centre. In that work, the authors propose a series of fitting functions together with a physical explanation for the behaviour of the signal at high doses. In Duval and Guilarte (2015) we actually tested these various functions for dating purpose, and concluded that the so-called Ti-2 function was the most appropriate. This function is NOT a polynomial at all, and there are no more issues about the back extrapolation than for a SSE function. As a matter of fact, the Ti-2 function takes into account the fact that the Ti signal behaves as a SSE on the lowest part of the DRC. To sum up, the fitting function that we use here in the present work is based on Wagner and Woda (2007) and further details about its potential and limitations may be found in Duval and Guilarte (2015).

Woda, C., Wagner, G.A., 2007. Non-monotonic dose dependence of the Ge- and Ti centres in quartz. *Radiat. Meas.* 42, 1441e1452.

The final comment saying that the Al ages are underestimated lie on the comparison with results obtained by other methods but not when we look at the growth curves. The table S18 in which data weighting are compared is not conclusive for the few usable results. The authors calculated their ages with a 1/12. So, when I look at the growth curve in front of the natural point which is under the curve, I can not say that the ages are underestimated.

Not sure which sample the reviewer refers to here. Several factors may explain an age underestimation. It could obviously come from the evaluation of the D_e value, and might be

related to the fitting, but not only that. Since ESR analyses are based on multi-grain aliquots, we cannot exclude that a sample showing various populations of quartz with different bleaching histories may lead to totally meaningless dose estimates (e.g., del Val et al., 2022). The truth is we are being quite cautious in our interpretation of the ESR data set. In any case, we'd like to remind that the ESR ages do not play a major role but rather a supporting role in the chronological interpretation of each site.

del Val, M., Alonso, M.J., Duval, M., Arriolabengoa, M., Álvarez, I., Bodego, A., Hai Cheng, Hermoso de Mendoza, A., Aranburu, A., Iriarte, E. (2022). Luminescence and ESR dating of the multi-level cave system of Alkerdi-Zelaieta (Navarre, N Spain) and implications for provenance study. *Quaternary Geochronology* 101380. <https://doi.org/10.1016/j.quageo.2022.101380>.

Minor points

Figure 2 shows a vertical extinction window that is not very conclusive. It would be better to show this window as a function of time !

The extinction window is presented as a function of time in Fig 3, but Fig 2 is based on the data from the each cave – the caves have been placed in chronological order based on their dating results as seen in Fig 2a so the extinction window is in the correct location based on the chronology. But we decided not to use a timeline on the x axis for this figure so that the pollen and DMTA data from each cave could be clearly observed rather than being densely packed around the extinction window and beyond as seen in Fig 3

Yixiantian cave : I see only one speleothem dating result in the SI section 8 ?

The missing result has been added to the table

Referee #5 (Remarks to the Author):

Dear Dr. Zhang et al.,

I find your article a very stimulating and comprehensive approach to the questions surrounding Gigantopithecus' extinction. I find only a series of minor issues with the paper, and fully support its publication.

Minor Issues:

Supplementary Data

Page 2. SI Section 8 is labeled U-Series of Flowstones, but on p. 97 it is called U-series Dating of Carbonates and Bone. Please reconcile this difference.

This has been corrected

Table S12 The name of the cave Upper Pubu is not legible.

Sorry we are confused the Table S12 doesn't mention Upper Pubu – we assume you are referring to Table S11 – we have adjusted the column to make Upper Pubu legible.

Page 67. Line 81 and 82. Your statement about why sediment is expected to be younger seems oversimplified to me, and incorrect. From my point of view, fauna and sediments can be coeval, being transported with fluvial sediment during flooding, or dying in caves and then being enclosed by LATER DEPOSITION of aeolian or fluvial sediments. I would expect the time differences to be small in the latter case, effectively making them coeval with respect to the uncertainties in your dating methods. You will need to remove this oversimplified statement and argue from a different point of view for your younger sediment vs. fossil ages.

We agree that the dating of faunal and sediments can be coeval within error uncertainties if the time between the death of the organism and time of burial is short. Some of our results certainly display this potential. There is also the possibility that certain organisms die in the cave environment and their skeletons are rapidly encased in the burial sediments, however this is not the case for the *G. blacki* fossils as the caves are too small for the sheer size of *G. blacki* and they would have only been present on the landscape outside the cave. However, there is also the possibility that the organism dies on the landscape and the carcass is present on the landscape for a certain period of time before being deposited in the cave – this would make the sediment age younger than the fossil age. We have modified the statement to incorporate these different scenarios.

Table S13, p. 76. Please verify that the last column in Table S13 is +/- 2 sigma and label it as such.

This has been corrected

p.78

line 17 replace "thieved" with "sieved"

line 21 replace "nto" with "not"

line 30 replace "steps" with "step"

line 55 replace "close-system" with "closed-system"

line 61 replace "which" with "with"?

These have all been corrected

Table S14 don't understand text in footnote d : ThO ?

The text was amended accordingly to "U and Th decay from ²⁰⁴".

p. 84, line 73 This sentence is very confusing -missing some parentheses and wording is unclear. Please rewrite sentence.

This sentence has been rewritten "Depending on the samples and ESR signals considered, fitting was performed using either an exponential+linear function (EXP+LIN) (see equation in ²¹²) or a Single Saturating Exponential (SSE) function (see equation in ²¹³), and data were weighted by the inverse of the squared ESR intensity ($1/I^2$) ²¹², the inverse of the squared experimental error ($1/s^2$) and with equal weights (EW)."

p. 85 first sentence Not clear, please rewrite

line 117: this is unintelligible. Please rewrite

These sentences have been rewritten

line 157: replace "gain" with "grain"

This has been corrected

p.90 This Discussion section has considerable grammatical problems, it is intelligible to a knowledgeable reader, but needs to be carefully rewritten.

Section 3.2 Age Results

line 29: replace "it" with "the pattern"

This has been corrected

Table S21

Put in a footnote that explains (If I am correct!) that sample codes with and F after the underscore are flowstones, and those with a B after the underscore are bones.

Actually, all the samples with _F are all flowstones even the ones with -A and -B afterwards – this just signifies different measurements of the same sample. The teeth and bone samples are labelled as tooth and bone after the cave code. This has been added to the table caption.

Table S22, p. 102

How is the age for the 25 bone samples calculated? Please explain.

The 25 bone samples are actually 25 repeated measurements on the same bone. We have amended the SI sentence to “For the bone, 25 single analytical spot measurements were conducted along a transect perpendicular to the bone’s surface, and a single open-system ^{230}Th -U model age was calculated using the R package UThwgl following the procedure described in Dosseto and Marwick (2022), and with 10,000 iterations.” Thus, there is a single open-system age model for the transect of 25 analyses and an age of $680 \pm 217-188$ ka was calculated. While the uncertainty is large, it gives an estimate of the minimum burial age of this bone.

How is the isochron age for Wuyun calculated?

Explanation modified in the SI for clarity, “For sample CWUY_F1, because of the large amount of detrital Th, a single isochron age was calculated from the twelve analyses using a closed system model²²³.” This was performed using the R package IsoPlotR (Vermeesch, 2018)

Main document (merged pdf).

Line 708 replace "place" with "placed"

Line 709 put a space between 200 and micron

Line 709 replace "clean" with "cleaned"

Line 710 replace "preabation" with "preablation"

Line 723 please define DAD

All these have been corrected

Reviewer Reports on the Third Revision:

Referees' comments:

Referee #4 (Remarks to the Author):

I thank the authors for their answers and for the multi method approach which probably represents years of work. However I stay skeptical on some points and particularly on the ESR/U-Th ages calculated on teeth (see below).

SSE and EXPLin functions

According to your answer you do not want to bring confusion because several DE can be obtained. It is a way to recognize that there are differences. I think that the authors should read also the work of other searchers and stop to auto-cite systematically. There are several papers in which DE are calculated by EXPLin because SSE clearly overestimate the values. EXPLin functions work according some constraints that the authors do not use. For instance the first saturation should not be lower than the 5th or 6th point of the growth curve. If Dmax is lower than 2000, SSE is more accurate and if Dma is higher, then ExpLin is more accurate. But It is beyond the topic of the article and I will end here.

Age calculations

When you compare U and Th standards, we can use the 238 keV peak (212lead) or the 583 keV peak (208Thallium) for calibrating Th. But you cannot compare with uranium which has different energies. So you should work with daughters like Rn which must be in equilibrium with uranium (no loss). This is never the case! So for samples with a high uranium content you should take into account this fact. Moreover, the U content in the teeth is relatively high and this has probably an effect on the age. I used a random value just for checking if there are differences in age or not.

Water content in enamel

Ok maybe it was mentioned just by Grün, but anyway a 0% value for water content in enamel is presently preferred even though there is no fundamental changing in the results!

Ages (Shao et al., 2014)

All or quasi p-values are positive suggesting a recent uptake. So, why do not you calculate the ages by using the Grün's Programme?

Age calculation for Bapeng : I used exactly the same parameters with the Grün's programme excepting that In situ dose mentioned in the OSL Table is 330 and not 784 microGy/a. So I have an older age than yours!

Yanliang : You put a De of 1007 and I used a De as mentioned in your Table S14 of 1407 Gy ! The in situ dose represents on the OSL table a value of 245 and you used a value of 796! So it explains the age differences.

Baixian : the gamma dose is different in your calculation sheet and in the OSL table of a factor 2 and I get an older age !!

You wrote that you used the values made for OSL analyses so.

So before the paper be published, the authors should absolutely check all the data used for calculating ages.

ESR ages on quartz Al

If quartz absorbed a massive radiation dose, it should be easy to make a gamma measurement of the quartz and you will have a rapid answer, no?

About Al signal and potential underestimation, I cannot understand from a physical point of view, your sentence saying that a « sample showing various populations of quartz with different bleaching histories may lead to totally meaningless dose estimates »! Do you mean that titane centers are not affected if there are various populations of quartz while Al centers are? It is difficult to imagine that titane center are able to yield better results than Al centers specially for samples older than 400 ka!

Referee #5 (Remarks to the Author):

I have checked the SI and Manuscript against the changes proposed in the Rebuttal to my comments. Most have been made faithfully, but there are a few serious omissions of the proposed changes (I cannot find them in the SI or the Manuscript using searches), which are listed below.

Also there is one new issue raised. They have defined one of their acronyms (DAD) as Diffusion Absorption Density modelling and now that it has been defined I request that they explain the modeling and provide a reference publication to that (see last comment below).

Referee #5 (Remarks to the Author):

Minor Issues:

Supplementary Data

Page 2. SI Section 8 is labeled U-Series of Flowstones, but on p. 97 it is called U-series Dating of Carbonates and Bone. Please reconcile this difference.

This has been corrected

RESPONSE: THIS HAS NOT BEEN CORRECTED, IT IS DIFFERENT ON PAGE 2 AND ON PAGE 99.

Table S12 The name of the cave Upper Pubu is not legible.

Sorry we are confused the Table S12 doesn't mention Upper Pubu – we assume you are referring to Table S11 – we have adjusted the column to make Upper Pubu legible.

RESPONSE: IT HAS BEEN DONE, THANK YOU.

Page 67. Line 81 and 82. Your statement about why sediment is expected to be younger seems oversimplified to me, and incorrect. From my point of view, fauna and sediments can be coeval, being transported with fluvial sediment during flooding, or dying in caves and then being enclosed by LATER DEPOSITION of aeolian or fluvial sediments. I would expect the time differences to be small in the latter case, effectively making them coeval with respect to the uncertainties in your dating methods. You will need to remove this oversimplified statement and argue from a different point of view for your younger

sediment vs. fossil ages.

We agree that the dating of faunal and sediments can be coeval within error uncertainties if the time between the death of the organism and time of burial is short. Some of our results certainly display this potential. There is also the possibility that certain organisms die in the cave environment and their skeletons are rapidly encased in the burial sediments, however this is not the case for the *G. blacki* fossils as the caves are too small for the sheer size of *G. blacki* and they would have only been present on the landscape outside the cave. However, there is also the possibility that the organism dies on the landscape and the carcass is present on the landscape for a certain period of time before being deposited in the cave – this would make the sediment age younger than the fossil age. We have modified the statement to incorporate these different scenarios.

RESPONSE: IT HAS BEEN DONE, THANK YOU.

Table S13, p. 76. Please verify that the last column in Table S13 is +/- 2 sigma and label it as such.

This has been corrected

RESPONSE: IT HAS BEEN DONE, THANK YOU.

p.78

line 17 replace "thieved" with "sieved"

line 21 replace "nto" with "not"

line 30 replace "steps" with "step"

line 55 replace "close-system" with "closed-system"

line 61 replace "which" with "with"?

These have all been corrected

RESPONSE: IT HAS BEEN DONE, THANK YOU.

Table S14 don't understand text in footnote d : ThO ?

The text was amended accordingly to "U and Th decay from 204".

RESPONSE: IT HAS BEEN DONE, THANK YOU.

p. 84, line 73 This sentence is very confusing -missing some parentheses and wording unclear. Please rewrite sentence.

This sentence has been rewritten "Depending on the samples and ESR signals considered, fitting was performed using either an exponential+linear function(EXP+LIN) (see equation in 212) or a Single Saturating Exponential (SSE) function(see equation in 213), and data were

weighted by the inverse of the squared ESR intensity ($1/I^2$) $2I^2$, the inverse of the squared experimental error ($1/s^2$) and with equal weights (EW).”

RESPONSE: IT HAS BEEN DONE, THANK YOU.

p. 85 first sentence Not clear, please rewrite
line 117: this is unintelligible. Please rewrite
These sentences have been rewritten

line 157: replace "gain" with "grain"
This has been corrected

RESPONSE: IT HAS BEEN DONE, THANK YOU.

p.90 This Discussion section has considerable grammatical problems, it is intelligible to a knowledgeable reader, but needs to be carefully rewritten.

Section 3.2 Age Results
line 29: replace "it" with "the pattern"
This has been corrected

RESPONSE: IT HAS BEEN DONE, THANK YOU.

Table S21
Put in a footnote that explains (if I am correct!) that sample codes with and F after the underscore are flowstones, and those with a B after the underscore are bones.
Actually, all the samples with _F are all flowstones even the ones with -A and -B afterwards – this just signifies different measurements of the same sample. The teeth and bone samples are labelled as tooth and bone after the cave code. This has been added to the table caption.

RESPONSE: IT HAS BEEN DONE, THANK YOU.

Table S22, p. 102
How is the age for the 25 bone samples calculated? Please explain.
The 25 bone samples are actually 25 repeated measurements on the same bone. We have amended the SI sentence to “For the bone, 25 single analytical spot measurements were conducted along a transect perpendicular to the bone’s surface, and a single open-system ^{230}Th -U model age was calculated using the R package UThwigl following the procedure described in Dosseto and Marwick (2022), and with 10,000 iterations.” Thus, there is a single open-system age model for the transect of 25 analyses and an age of 680 ± 217 -188 ka was calculated. While the uncertainty is large, it gives an estimate of the minimum burial age of this bone.

RESPONSE: A SEARCH FOR THREE DIFFERENT STRINGS OF WORDS WITHIN THE PROPOSED ADDED SENTENCE "For the bone, 25 single analytical spot measurements were conducted along a transect perpendicular to the bone's surface, and a single open-system ^{230}Th -U model age was calculated using the R package UThwigl following the procedure described in Dosseto and Marwick (2022), and with 10,000 iterations." DID NOT RETURN ANY MATCHES IN THE SI. PLEASE ADD THIS CHANGE. IT IS ACCEPTABLE.

How is the isochron age for Wuyun calculated?

Explanation modified in the SI for clarity, "For sample CWUY_F1, because of the large amount of detrital Th, a single isochron age was calculated from the twelve analyses using a closed system model223." This was performed using the R package IsoPlotR (Vermeesch2018)

RESPONSE: A SEARCH FOR THREE DIFFERENT STRINGS OF WORDS WITHIN THE PROPOSED ADDED SENTENCE "For sample CWUY_F1, because of the large amount of detrital Th, a single isochron age was calculated from the twelve analyses using a closed system model223." DID NOT RETURN ANY MATCHES IN THE SI. PLEASE ADD THIS CHANGE. IT IS ACCEPTABLE.

Main document (merged pdf).

Line 708 replace "place" with "placed"

Line 709 put a space between 200 and micron

Line 709 replace "clean" with "cleaned"

Line 710 replace "preabation" with "preablation"

Line 723 please define DAD

All these have been corrected

RESPONSE: IT HAS BEEN DONE, THANK YOU.

Now that you have defined DAD as Diffusion Absorption Density Modelling, please explain what it is, and provide a reference publication please.

Author Rebuttals to Third Revision:

Referees' comments:

Referee #4 (Remarks to the Author):

I thank the authors for their answers and for the multi method approach which probably represents years of work. However I stay sceptical on some points and particularly on the ESR/U-Th ages calculated on teeth (see below).

SSE and EXPLin functions

According to your answer you do not want to bring confusion because several DE can be obtained. It is a way to recognize that there are differences. I think that the authors should read also the work of other searchers and stop to auto-cite systematically. There are several papers in which DE are calculated by EXPLin because SSE clearly overestimate the values. EXPLin functions work according some constraints that the authors do not use. For instance the first saturation should not be lower than the 5th or 6th point of the growth curve. If Dmax is lower than 2000, SSE is more accurate and if Dma is higher, then ExpLin is more accurate. But It is beyond the topic of the article and I will end here.

We thank the reviewer for the noteworthy debate, and we agree that such discussion is beyond the scope of the current paper.

We would also like to emphasize that we are fully aware of the diligent work carried out by our colleagues. However, it's worth noting that the majority, are primarily focused on powder multi-aliquot DRC and De determination, which might explain the utilisation of EXPLin (although we must admit that we do not recall any study on known-dose samples demonstrating that the EXP+LIN should be preferred over the SSE or DSE; the EXP+LIN is just an approximation of a DSE, but is not recommended for dose estimations on tooth enamel). In contrast, our research is centered around enamel fragments. This unique approach affords us the ability to perform radical separation and angular response modelling. Consequently, we can effectively correct for the presence of unstable radicals, the main source of age underestimation and miscalculation. It's important to recognize that the amount of unstable radicals varies significantly from one tooth to another, impacting both the DRC and the D_e values. These variations cannot be adequately addressed when working with powdered samples.

Age calculations

When you compare U and Th standards, we can use the 238 keV peak (212lead) or the 583 keV peak (208Thallium) for calibrating Th. But you cannot compare with uranium which has different energies. So you should work with daughters like Rn which must be in equilibrium with uranium (no loss). This is never the case! So for samples with a high

uranium content you should take into account this fact. Moreover, the U content in the teeth is relatively high and this has probably an effect on the age. I used a random value just for checking if there are differences in age or not.

We appreciate the reviewer's comment and the provided explanation. We are in agreement with and fully cognizant of the process outlined by the reviewer. Our awareness extends to the methods employed, which involve high-resolution gamma spectrometry. While we acknowledge the numerous merits associated with these measurements, we believe that MC-ICPMS (Multi-Collector Inductively Coupled Plasma Mass Spectrometry) measurements offer a notably higher level of precision, when calculating isotopic ratios spanning from ^{238}U to ^{230}Th . Furthermore, we hold the perspective that addressing isotopic disequilibrium, if present, should not be a one-size-fits-all approach in age calculations. Using a “snapshot” measurements of Rn disequilibrium in laboratory setting and applying the results to the entire burial period might not represent good practice. We acknowledge that more work needs to happen in this space. Meanwhile, we believe our current approach to clearly stipulate a systematic approach in which we assume equilibrium, is warranted.

Water content in enamel

Ok maybe it was mentioned just by Grün, but anyway a 0% value for water content in enamel is presently preferred even though there is no fundamental changing in the results!

We respectfully disagree with the suggestion. Our measurements have indicated a 3% water content within our sample. Therefore, we believe that altering the measured value to a blank value of 0% would not be justified. It's worth noting that this appears to be a somewhat moot argument, as the dating outcomes remain unaffected.

To provide further clarity, we have included an additional sentence in the S14 table caption: "Water content was estimated ... using a heat treatment of 60°C for 24 hours on a *Pongo* tooth and sediment powder, respectively."

Ages (Shao et al., 2014)

All or quasi p-values are positive suggesting a recent uptake. So, why do not you calculate the ages by using the Grün's Programme?

We thank the reviewer for the comments and for diligently taking the time to re-calculate our values. As the reviewer would know, we hold the work of Prof. Grün in high regard, being Joannes-Boyau's mentor for many years. Yet, it's important to note that while the DATA program, which operates exclusively in outdated MS-DOS system, is indeed a useful and respected modelling software, it should not be considered the sole acceptable tool for these calculations. Furthermore, it would be unwise to assume that the DATA program is providing

infallible solution for accurately determining US-ESR ages. Like any model, it has its limitations.

Age calculation for Bapeng : I used exactly the same parameters with the Grun's programme excepting that In situ dose mentioned in the OSL Table is 330 and not 784 microGy/a. So I have an older age than yours!

Yanliang : You put a D_e of 1007 and I used a D_e as mentioned in your Table S14 of 1407 Gy ! The in situ dose represents on the OSL table a value of 245 and you used a value of 796! So it explains the age differences.

Baixian : the gamma dose is different in your calculation sheet and in the OSL table of a factor 2 and I get an older age !!

You wrote that you used the values made for OSL analyses so.

So before the paper be published, the authors should absolutely check all the data used for calculating ages.

We doubled checked the values in the table S14 and found that the correct D_e value was reported for Yanliang (1007Gy).

Regarding the gamma dose, we recognize that our previous sentence, "*The external gamma dose rates were assessed using a portable gamma spectrometer at each site, employing the threshold method (refer to OSL methods in Supplementary Section 3)*" may have been misleading. We did not intend to imply that the same measurements or values used for luminescence calculations were systematically applied but rather that we employed the same threshold methodology and therefore refer the reader to the same method description above.

To eliminate any potential confusion, we have included two additional sentences to provide further clarity to our discussion p80.

"The external gamma dose rates were assessed using a portable gamma spectrometer at each site, employing the threshold method (refer to the similar luminescence methods in Supplementary Section 3). These measurements were conducted as closely as possible to the locations where the teeth were originally excavated when known. Consequently, there may be some variations between gamma dosimetry utilized for ESR modelling and luminescence calculations. For detailed reference, all US-ESR values are comprehensively documented in table S14."

ESR ages on quartz Al

If quartz absorbed a massive radiation dose, it should be easy to make a gamma measurement of the quartz and you will have a rapid answer, no?

We genuinely find it challenging to comprehend the reviewer's intended meaning in this statement. It's important to note that the radiation dose absorbed by quartz is not re-emitted and therefore cannot be measured through "gamma measurements". Consequently, we are regrettably unable to provide a response to this specific comment.

About Al signal and potential underestimation, I cannot understand from a physical point of view, your sentence saying that a « sample showing various populations of quartz with different bleaching histories may lead to totally meaningless dose estimates »! Do you mean that titane centers are not affected if there are various populations of quartz while Al centers are? It is difficult to imagine that titane center are able to yield better results than Al centers specially for samples older than 400 ka!

What we intended to convey here is that multi-grain quartz samples might lack homogeneity, possibly consisting of different grain populations characterized by varying radiation sensitivity and/or D_e values. To enhance the clarity of our discussion, we have added the following sentence to p98: 'Further research will evaluate the relative homogeneity of the quartz luminescence characteristics (e.g., radiation sensitivity, signal brightness) at a single grain level in order to obtain any comparative insights that could be extrapolated to the ESR data.'

As far as whether the Ti would not be affected by this issue (unlike the Al signal), we cannot really say right now based on the existing data set, since the Ti dose estimates are not considered reliable enough for such a comparison (SI, section 3.2.: 'the Ti age results should be treated with caution.'). We therefore prefer to remain cautious in our interpretation of the results. While we do understand (although we do not necessarily agree with it) the reviewer's comment 'It is difficult to imagine that titane (*sic*) center are able to yield better results than Al centers specially for samples older than 400 ka', we would like to remind the reviewer that these samples show unusual characteristics and behaviors, which might result in unexpected/unusual signal pattern.

Referee #5 (Remarks to the Author):

I have checked the SI and Manuscript against the changes proposed in the Rebuttal to my comments. Most have been made faithfully, but there are a few serious omissions of the proposed changes (I cannot find them in the SI or the Manuscript using searches), which are listed below.

Also there is one new issue raised. They have defined one of their acronyms (DAD) as Diffusion Absorption Density modelling and now that it has been defined I request that they explain the modelling and provide a reference publication to that (see last comment

below).

We have explained the modelling in the following text added to the SI on p71:

Measurements were undertaken with rasters parallel to each other but perpendicular to the buccal lingual direction (from pulp cavity to enamel outer surface). For most teeth a Diffusion-adsorption (D-A) model of U uptake for performed D-A is a computational and mathematical approach used to simulate and analyze the behavior of uranium diffusion in porous materials. This modelling technique describe how substances disperse and evolve over time in response to concentration gradients according to, absorption process¹⁹⁷

Referee #5 (Remarks to the Author):

Minor Issues:

Supplementary Data

Page 2. SI Section 8 is labeled U-Series of Flowstones, but on p. 97 it is called U-series Dating of Carbonates and Bone. Please reconcile this difference.

This has been corrected

RESPONSE: THIS HAS NOT BEEN CORRECTED, IT IS DIFFERENT ON PAGE 2 AND ON PAGE 99.

The flowstones had been corrected to carbonates but the 'teeth' instead of bone was indeed a typo. Apologies this has now been properly corrected

Table S12 The name of the cave Upper Pubu is not legible.

Sorry we are confused the Table S12 doesn't mention Upper Pubu – we assume you are referring to Table S11 – we have adjusted the column to make Upper Pubu legible.

RESPONSE: IT HAS BEEN DONE, THANK YOU.

Page 67. Line 81 and 82. Your statement about why sediment is expected to be younger seems oversimplified to me, and incorrect. From my point of view, fauna and sediments can be coeval, being transported with fluvial sediment during flooding, or dying in caves and then being enclosed by LATER DEPOSITION of aeolian or fluvial sediments. I would expect the time differences to be small in the latter case, effectively making them coeval with respect to the uncertainties in your dating methods. You will need to remove this oversimplified statement and argue from a different point of view for your younger sediment vs. fossil ages.

We agree that the dating of faunal and sediments can be coeval within error uncertainties if the time between the death of the organism and time of burial is short.

Some of our results certainly display this potential. There is also the possibility that certain organisms die in the cave environment and their skeletons are rapidly encased in the burial sediments, however this is not the case for the *G. blacki* fossils as the caves are too small for the sheer size of *G. blacki* and they would have only been present on the landscape outside the cave. However, there is also the possibility that the organism dies on the landscape and the carcass is present on the landscape for a certain period of time before being deposited in the cave – this would make the sediment age younger than the fossil age. We have modified the statement to incorporate these different scenarios.

RESPONSE: IT HAS BEEN DONE, THANK YOU.

Table S13, p. 76. Please verify that the last column in Table S13 is +/- 2 sigma and label it as such.

This has been corrected

RESPONSE: IT HAS BEEN DONE, THANK YOU.

p.78

line 17 replace "thieved" with "sieved"

line 21 replace "nto" with "not"

line 30 replace "steps" with "step"

line 55 replace "close-system" with "closed-system"

line 61 replace "which" with "with"?

These have all been corrected

RESPONSE: IT HAS BEEN DONE, THANK YOU.

Table S14 don't understand text in footnote d : ThO ?

The text was amended accordingly to "U and Th decay from 204".

RESPONSE: IT HAS BEEN DONE, THANK YOU.

p. 84, line 73 This sentence is very confusing -missing some parentheses and wording unclear. Please rewrite sentence.

This sentence has been rewritten "Depending on the samples and ESR signals considered, fitting was performed using either an exponential+linear function (EXP+LIN) (see equation in 212) or a Single Saturating Exponential (SSE) function (see equation in 213), and data were weighted by the inverse of the squared ESR intensity ($1/I^2$) 212, the inverse of the squared experimental error ($1/s^2$) and with equal weights (EW)."

RESPONSE: IT HAS BEEN DONE, THANK YOU.

p. 85 first sentence Not clear, please rewrite
line 117: this is unintelligible. Please rewrite
These sentences have been rewritten

line 157: replace "gain" with "grain"
This has been corrected

RESPONSE: IT HAS BEEN DONE, THANK YOU.

p.90 This Discussion section has considerable grammatical problems, it is intelligible to a knowledgeable reader, but needs to be carefully rewritten.

Section 3.2 Age Results
line 29: replace "it" with "the pattern"
This has been corrected

RESPONSE: IT HAS BEEN DONE, THANK YOU.

Table S21

Put in a footnote that explains (If I am correct!) that sample codes with and F after the underscore are flowstones, and those with a B after the underscore are bones. Actually, all the samples with _F are all flowstones even the ones with -A and -B afterwards
– this just signifies different measurements of the same sample. The teeth and bone samples are labelled as tooth and bone after the cave code. This has been added to the table caption.

RESPONSE: IT HAS BEEN DONE, THANK YOU.

Table S22, p. 102

How is the age for the 25 bone samples calculated? Please explain.

The 25 bone samples are actually 25 repeated measurements on the same bone. We have amended the SI sentence to "For the bone, 25 single analytical spot measurements were conducted along a transect perpendicular to the bone's surface, and a single open-system ^{230}Th -U model age was calculated using the R package UThwgl following the procedure described in Dosseto and Marwick (2022), and with 10,000 iterations." Thus, there is a single open-system age model for the transect of 25 analyses and an age of 680 ± 217 -188 ka was calculated. While the uncertainty is large, it gives an estimate of the minimum burial age of this bone.

RESPONSE: A SEARCH FOR THREE DIFFERENT STRINGS OF WORDS WITHIN THE PROPOSED ADDED SENTENCE "For the bone, 25 single analytical spot measurements were conducted along a transect perpendicular to the bone's surface, and a single open-system ^{230}Th -U model age was calculated using the R package UThwgl following the procedure described in Dosseto and Marwick (2022), and with 10,000 iterations." DID NOT RETURN ANY MATCHES IN THE SI. PLEASE ADD THIS CHANGE. IT IS ACCEPTABLE.

The sentence was already included in p98 but the wording was slightly different "25 single analytical spot measurements" was not included and has now been added. This is probably why it wasn't found in a search

How is the isochron age for Wuyun calculated?

Explanation modified in the SI for clarity, "For sample CWUY_F1, because of the large amount of detrital Th, a single isochron age was calculated from the twelve analyses using a closed system model²²³." This was performed using the R package IsoPlotR (Vermeesch2018)

RESPONSE: A SEARCH FOR THREE DIFFERENT STRINGS OF WORDS WITHIN THE PROPOSED ADDED SENTENCE "For sample CWUY_F1, because of the large amount of detrital Th, a single isochron age was calculated from the twelve analyses using a closed system model²²³." DID NOT RETURN ANY MATCHES IN THE SI. PLEASE ADD THIS CHANGE. IT IS ACCEPTABLE.

This exact sentence was added to P105. What was missing was the final part "This was performed using the R package IsoPlotR (Vermeesch 2018)" so this is why it may not have returned a search match. This has now been added.

Main document (merged pdf).

Line 708 replace "place" with "placed"

Line 709 put a space between 200 and micron

Line 709 replace "clean" with "cleaned"

Line 710 replace "preabation" with "preablation"

Line 723 please define DAD

All these have been corrected

RESPONSE: IT HAS BEEN DONE, THANK YOU.

Now that you have defined DAD as Diffusion Absorption Density Modelling, please explain what it is, and provide a reference publication please.

See above

Reviewer Reports on the Third Revision

Referees' comments:

Referee #4 (Remarks to the Author):

SSE and ExpLin functions

There are less than 5 labs in the world performing ESR/U-series dating on teeth enamel. Our lab made at least 10 papers in which we claim that SSE may overestimate the DE and that is sometimes better to use exp Lin in some cases especially for old samples with high De. « The exp lin as an approximation of a DSE » would mean that a linear line would correspond to an exponential curve ! I am not sure of that consideration.

Age calculations

I was talking about a comparison between two models and I don't talk about the Grün's programme as a panacea but it remains really performant especially in the case of the major part of your samples that exhibit p-values comprised between -0.64 and +3.7 (excepted for Heijang sample) ! For gamma measurement in the field I can imagine that there are variations for a same layer of 10-20 %. But in your calculations, the variations change from 330 to 784 microGy/y for Bapeng and from 245 to 796 microGy/y for Yanliang, and reaches a factor 2 in difference for Baixian !! I can not believe such differences in the results which change considerably the ages which become not reliable. So, the conclusions of the paper are affected by this consideration.

ESR ages on quartz

I do not understand the answer of the authors about a massive radiation in quartz which would not be re-emitted ?

The multi-grains quartz samples is the rule in ESR dating because of the relatively weak sensitivity of the method for this mineral, excepting some scarce analyses performed in Q-Band. So, how it could be different not to find heterogeneity in the dose response of at least 3000 quartz grains in a tube ? How can you compare luminescence and ESR on a quartz grain ? What method ? OSL, TL ?

I do not recommend the publication of this paper in this state

Referee #5 (Remarks to the Author):

The only change I suggested in the last round was to remove the last four words in this passage within the SI:

1232 For most teeth a Diffusion-adsorption (D-A) model of U uptake for performed D-A is a computational

1233 and mathematical approach used to simulate and analyze the behavior of uranium diffusion in porous

1234 materials. This modeling technique describe how substances disperse and evolve over time in response

1235 to concentration gradients according to, absorption process197.

The passage lines 1232 through 1235 are still there (I copied them from the existing manuscript version. Thus the changes have not been made in the SI.

This change needs to be made.

I remind you also that I suggested that the reference 197 at the end of the passage be retained.

Dr. W.Jack Rink

Author Rebuttals to Third Revision:

Referee #4 (Remarks to the Author):

SSE and ExpLin functions

There are less than 5 labs in the world performing ESR/U-series dating on teeth enamel. Our lab made at least 10 papers in which we claim that SSE may overestimate the DE and that is sometimes better to use exp Lin in some cases especially for old samples with high De. « The exp lin as an approximation of a DSE » would mean that a linear line would correspond to an exponential curve ! I am not sure of that consideration.

Age calculations

I was talking about a comparison between two models and I don't talk about the Grün's programme as a panacea but it remains really performant especially in the case of the major part of your samples that exhibit p-values comprised between -0.64 and +3.7 (excepted for Heijang sample) !

For gamma measurement in the field I can imagine that there are variations for a same layer of 10-20 %. But in your calculations, the variations change from 330 to 784 microGy/y for Bapeng and from 245 to 796 microGy/y for Yanliang, and reaches a factor 2 in difference for Baixian !! I can not believe such differences in the results which change considerably the ages which become not reliable. So, the conclusions of the paper are affected by this consideration.

ESR ages on quartz

I do not understand the answer of the authors about a massive radiation in quartz which would not be re-emitted ?

The multi-grains quartz samples is the rule in ESR dating because of the relatively weak sensitivity of the method for this mineral, excepting some scarce analyses performed in Q-Band. So, how it could be different not to find heterogeneity in the dose response of at least 3000 quartz grains in a tube ? How can you compare luminescence and ESR on a quartz grain ? What method ? OSL, TL ?

I do not recommend the publication of this paper in this state

No responses required by Nature

Referee #5 (Remarks to the Author):

The only change I suggested in the last round was to remove the last four words in this passage within the SI:

1232 For most teeth a Diffusion-adsorption (D-A) model of U uptake for performed D-A is a computational

1233 and mathematical approach used to simulate and analyze the behavior of uranium diffusion in porous

1234 materials. This modeling technique describe how substances disperse and evolve over time in response

1235 to concentration gradients according to, absorption process¹⁹⁷.

The passage lines 1232 through 1235 are still there (I copied them from the existing manuscript version. Thus the changes have not been made in the SI.

This change needs to be made.

This change has been made now

I remind you also that I suggested that the reference 197 at the end of the passage be retained.

The reference has been retained